# Neural Networks for Principal Component Analysis: A New Loss Function Provably Yields Ordered Exact Eigenvectors

## Abstract

In this paper, we propose a new loss function for performing principal component analysis (PCA) using linear autoencoders (LAEs). Optimizing the standard $L_2$ loss results in a decoder matrix that spans the principal subspace of the sample covariance of the data, but fails to identify the exact eigenvectors. This downside originates from an invariance that cancels out in the global map. Here, we prove that our loss function eliminates this issue, i.e. the decoder converges to the exact ordered unnormalized eigenvectors of the sample covariance matrix. For this new loss, we establish that all local minima are global optima and also show that computing the new loss (and also its gradients) has the same order of complexity as the classical loss. We report numerical results on both synthetic simulations, and a real-data PCA experiment on MNIST (i.e., a $60,000 \times 784$ matrix), demonstrating our approach to be practically applicable and rectify previous LAEs' downsides.

## 1 Introduction

Ranking among the most widely-used and valuable statistical tools, Principal Component Analysis (PCA) represents a given set of data within a new orthogonal coordinate system in which the data are uncorrelated and the variance of the data along each orthogonal axis is successively ordered from the highest to lowest. The projection of data along each axis gives what are called principal components. Theoretically, eigendecomposition of the covariance matrix provides exactly such a transformation. For large data sets, however, classical decomposition techniques are infeasible and other numerical methods, such as least squares approximation schemes, are practically employed. An especially notable instance is the problem of dimensionality reduction, where only the largest principal components—as the best representative of the data—are desired. Linear autoencoders (LAEs) are one such scheme for dimensionality reduction that is applicable to large data sets.

An LAE with a single fully-connected and linear hidden layer, and Mean Squared Error (MSE) loss function can discover the linear subspace spanned by the principal components. This subspace is the same as the one spanned by the weights of the decoder. However, it failure to identify the exact principal directions. This is due to the fact that, when the encoder is transformed by some matrix, transforming the decoder by the inverse of that matrix will yield no change in the loss. In other words, the loss possesses a symmetry under the action of a group of invertible matrices, so that directions (and orderings/permutations thereto) will not be discriminated.

The early work of Bourlard & Kamp (1988) and Baldi & Hornik (1989) connected LAEs and PCA and demonstrated the lack of identifiability of principal components. Several methods for neural networks compute the exact eigenvectors (Rubner & Tavan, 1989; Xu, 1993; Kung & Diamantaras, 1990; Oja et al., 1992), but they depend on either particular network structures or special optimization methods. It was recently observed (Plaut, 2018; Kunin et al., 2019) that regularization causes the left singular vectors of the decoder to become the exact eigenvectors, but recovering them still requires an extra decomposition step. As Plaut (2018) point out, no existent method recovers the eigenvectors from an LAE in an optimization-independent way on a standard network — this work fills that void.

Moreover, analyzing the loss surface for various architectures of linear/non-linear neural networks is a highly active and prominent area of research (e.g. Baldi & Hornik (1989); Kunin et al. (2019); Pretorius et al. (2018); Frye et al. (2019)). Most of these works extend the results of Baldi & Hornik (1989) for shallow LAEs to more complex networks. However, most retain the original MSE loss, and they prove the same critical point characterization for their specific architecture of interest. Most notably Zhou & Liang (2018) extends the results of Baldi & Hornik (1989) to deep linear networks and shallow RELU networks. In contrast in this work we are going after a loss with better loss surface properties.

We propose a new loss function for performing PCA using LAEs. We show that with the proposed loss function, the decoder converges to the exact ordered unnormalized eigenvectors of the sample covariance matrix. The idea is simple: for identifying $p$ principal directions we build up a total loss function as a sum of $p$ squared error losses, where the $i^{\text{th}}$ loss function identifies only the first $i$ principal directions. This approach breaks the symmetry since minimizing the first loss results in the first principal direction, which forces the second loss to find the first and the second. This constraint is propagated through the rest of the losses, resulting in all $p$ principal components being identified. For the new loss we prove that all local minima are global minima.

Consequently, the proposed loss function has both theoretical and practical implications. Theoretically, it provides better understanding of the loss surface. Specifically, we show that any critical point of our loss $L$ is a critical point of the original MSE loss but not vice versa, and conclude that $L$ eliminates those undesirable global minima of the original loss (i.e., exactly those which suffer from the invariance). Given that the set of critical points of $L$ is a subset of critical points of MSE loss, many of the previous work on loss surfaces of more complex networks likely extend. In light of the removal of undesirable global minima through $L$, examining more complex networks is certainly a very promising direction.

As for practical consequences, we show that the loss and its gradients can be compactly vectorized so that their computational complexity is no different from the MSE loss. Therefore, the loss $L$ can be used to perform PCA/SVD on large datasets using any method of optimization such as Stochastic Gradient Descent (SGD). Chief among the compellingly reasons to perform PCA/SVD using this method is that, in recent years, there has been unprecedented gains in the performance of very large SGD optimizations, with autoencoders in particular successfully handling larger numbers of high-dimensional training data (e.g., images). The loss function we offer is attractive in terms of parallelizability and distributability, and does not prescribe any single specific algorithm or implementation, so stands to continue to benefit from the arms race between SGD and its competitors.

More importantly, this single loss function (without an additional post hoc processing step) fits seamlessly into optimization pipelines (where SGD is but one instance). The result is that the loss allows for PCA/SVD computation as single optimization layer, akin to an instance of a fully differentiable building block in a NN pipeline Amos & Kolter (2017), potentially as part of a much larger network.

## 2    THE PROPOSED LOSS FUNCTION AND REVIEW OF FINAL RESULTS

Let $\boldsymbol{X} \in \mathbb{R}^{n \times m}$ and $\boldsymbol{Y} \in \mathbb{R}^{n \times m}$ be the input and output matrices, where $m$ centered sample points, each $n$-dimensional, are stacked column-wise. Let $\boldsymbol{x}_j \in \mathbb{R}^n$ and $\boldsymbol{y}_j \in \mathbb{R}^n$ be the $j^{\text{th}}$ sample input and output (i.e. the $j^{\text{th}}$ column of $\boldsymbol{X}$ and $\boldsymbol{Y}$, respectively). Define the loss function $L(\boldsymbol{A}, \boldsymbol{B})$ as

$$L(\boldsymbol{A}, \boldsymbol{B}) \coloneqq \sum_{i=1}^{p} \sum_{j=1}^{m} \|\boldsymbol{y}_j - \boldsymbol{A}\boldsymbol{I}_{i;p}\boldsymbol{B}\boldsymbol{x}_j\|_2^2 = \sum_{i=1}^{p} \|\boldsymbol{Y} - \boldsymbol{A}\boldsymbol{I}_{i;p}\boldsymbol{B}\boldsymbol{X}\|_F^2 \tag{1}$$

where $\langle \cdot, \cdot \rangle_F$ and $\|\cdot\|_F$ are the Frobenius inner product and norm, $\boldsymbol{I}_{i;p}$ is a $p \times p$ matrix with all elements zero except the first $i$ diagonal elements being one. (Or, equivalently, the matrix obtained by setting the last $p - i$ diagonal elements of a $p \times p$ identity matrix to zero, e.g. $\boldsymbol{I}_{2;3} = \begin{bmatrix} 1 & 0 & 0 \\ 0 & 1 & 0 \\ 0 & 0 & 0 \end{bmatrix}$.) In what follows, we shall denote the transpose of matrix $\boldsymbol{M}$ by $\boldsymbol{M}'$. Moreover, the matrices $\boldsymbol{A} \in \mathbb{R}^{n \times p}$, and $\boldsymbol{B} \in \mathbb{R}^{p \times n}$ can be viewed as the weights of the decoder and encoder parts of an LAE.

The results are based on the following standard assumptions that hold generically:

**Assumption 1.** *For an input $\boldsymbol{X}$ and an output $\boldsymbol{Y}$, let $\boldsymbol{\Sigma}_{xx} \coloneqq \boldsymbol{X}\boldsymbol{X}'$, $\boldsymbol{\Sigma}_{xy} \coloneqq \boldsymbol{X}\boldsymbol{Y}'$, $\boldsymbol{\Sigma}_{yx} \coloneqq \boldsymbol{\Sigma}'_{xy}$ and $\boldsymbol{\Sigma}_{yy} = \boldsymbol{Y}\boldsymbol{Y}'$ be their sample covariance matrices. We assume*

- *The input and output data are centered (zero mean).*

- *$\boldsymbol{\Sigma}_{xx}$, $\boldsymbol{\Sigma}_{xy}$, $\boldsymbol{\Sigma}_{yx}$ and $\boldsymbol{\Sigma}_{yy}$ are positive definite (of full rank and invertible).*

- *The covariance matrix $\boldsymbol{\Sigma} \coloneqq \boldsymbol{\Sigma}_{yx}\boldsymbol{\Sigma}_{xx}^{-1}\boldsymbol{\Sigma}_{xy}$ is of full rank with $n$ distinct eigenvalues $\lambda_1 > \lambda_2 > \cdots > \lambda_n$.*

- *The decoder matrix $\boldsymbol{A}$ has no zero columns.*

*Claim.* The main result of this work proved in Theorem 2 is as follows:

---

If the above assumptions hold then all the local minima of $L(\boldsymbol{A}, \boldsymbol{B})$ are achieved iff $\boldsymbol{A}$ and $\boldsymbol{B}$ are of the form

$$\boldsymbol{A} = \boldsymbol{U}_{1:p}\boldsymbol{D}_p$$
$$\boldsymbol{B} = \boldsymbol{D}_p^{-1}\boldsymbol{U}'_{1:p}\boldsymbol{\Sigma}_{yx}\boldsymbol{\Sigma}_{xx}^{-1},$$

where the $i^{\text{th}}$ column of $\boldsymbol{U}_{1:p}$ is the unit eigenvector of $\boldsymbol{\Sigma} \coloneqq \boldsymbol{\Sigma}_{yx}\boldsymbol{\Sigma}_{xx}^{-1}\boldsymbol{\Sigma}_{xy}$ corresponding to the $i^{\text{th}}$ largest eigenvalue and $\boldsymbol{D}_p$ is a diagonal matrix with nonzero diagonal elements. In other words, $\boldsymbol{A}$ contains ordered unnormalized eigenvectors of $\boldsymbol{\Sigma}$ corresponding to the $p$ largest eigenvalues. Moreover, all the local minima are global minima with the value of the loss function at those global minima being

$$L(\boldsymbol{A}, \boldsymbol{B}) = p \, \text{Tr}(\boldsymbol{\Sigma}_{yy}) - \sum_{i=1}^{p} (p - i + 1)\,\lambda_i,$$

where $\lambda_i$ is the $i^{\text{th}}$ largest eigenvalue of $\boldsymbol{\Sigma} \coloneqq \boldsymbol{\Sigma}_{yx}\boldsymbol{\Sigma}_{xx}^{-1}\boldsymbol{\Sigma}_{xy}$. In the case of autoencoder ($\boldsymbol{Y} = \boldsymbol{X}$): $\boldsymbol{\Sigma} = \boldsymbol{\Sigma}_{xx}$. Finally, while $L(\boldsymbol{A}, \boldsymbol{B})$ in the given form contains $O(p)$ matrix products, we will show that it can be evaluated with constant (less than 7) matrix products independent of the value $p$.

---

## 3  NOTATION

In this paper, the underlying field is always $\mathbb{R}$, and positive semidefinite matrices are symmetric by definition. The following constant matrices are used extensively throughout. The matrices $\boldsymbol{T}_p \in \mathbb{R}^{p \times p}$ and $\boldsymbol{S}_p \in \mathbb{R}^{p \times p}$ are defined as

$$(\boldsymbol{T}_p)_{ij} = (p - i + 1)\,\delta_{ij}, \text{ i.e. } \boldsymbol{T}_p = \text{diag}\,(p, p-1, \cdots, 1), \tag{2}$$

$$(\boldsymbol{S}_p)_{ij} = p - \max(i, j) + 1, \text{ i.e. } \boldsymbol{S}_p = \begin{bmatrix} p & p-1 & \cdots & 2 & 1 \\ p-1 & p-1 & \cdots & 2 & 1 \\ \vdots & \vdots & \ddots & 2 & 1 \\ 2 & 2 & 2 & 2 & 1 \\ 1 & 1 & 1 & 1 & 1 \end{bmatrix}, \text{ e.g. } S_4 = \begin{bmatrix} 4 & 3 & 2 & 1 \\ 3 & 3 & 2 & 1 \\ 2 & 2 & 2 & 1 \\ 1 & 1 & 1 & 1 \end{bmatrix}. \tag{3}$$

Another matrix that will appear in the formulation is $\hat{\boldsymbol{S}}_p \coloneqq \boldsymbol{T}_p^{-1}\boldsymbol{S}_p\boldsymbol{T}_p^{-1}$. Clearly, the diagonal matrix $\boldsymbol{T}_p$ is positive definite. As shown in Lemma 2, $\boldsymbol{S}_p$ and $\hat{\boldsymbol{S}}_p$ are positive definite as well.

## 4  MAIN THEOREMS

The general strategy to prove the above claim is as follows. First the analytical gradients of the loss is derived in a matrix form in Propositions 1 and 2. We compare the gradients with that of the original Minimum Square Error (MSE) loss. Next, we analyze the loss surface by solving the gradient equations which yields the general structure of critical points based on the rank of the decoder matrix $\boldsymbol{A}$. Next, we delineate several interesting properties of the critical points, notably, any critical point of the loss is also a critical point for the MSE loss but not the other way around. Finally, by performing second order analysis on the loss in Theorem 2 the exact equations for local minima are derived which is shown to be as claimed.

Let $\tilde{L}(\boldsymbol{A}, \boldsymbol{B})$ and $L(\boldsymbol{A}, \boldsymbol{B})$ be the original loss, and the proposed loss function, respectively, i.e.,

$$
\begin{aligned}
\tilde{L}(\boldsymbol{A}, \boldsymbol{B}) &:= \sum_{j=1}^{m} \|y_j - \boldsymbol{A}\boldsymbol{B}x_j\|_2^2 \\
&= \|\boldsymbol{Y} - \boldsymbol{A}\boldsymbol{B}\boldsymbol{X}\|_F^2
\end{aligned}
\qquad
\begin{aligned}
L(\boldsymbol{A}, \boldsymbol{B}) &:= \sum_{i=1}^{p} \sum_{j=1}^{m} \|y_j - \boldsymbol{A}\boldsymbol{I}_{i;p}\boldsymbol{B}x_j\|_2^2 \\
&= \sum_{i=1}^{p} \|\boldsymbol{Y} - \boldsymbol{A}\boldsymbol{I}_{i;p}\boldsymbol{B}\boldsymbol{X}\|_F^2
\end{aligned}
$$

The first step is to calculate the gradients with respect to $\boldsymbol{A}$ and $\boldsymbol{B}$ and set them to zero to derive the implicit expressions for the critical points. In order to do so, first, in Lemma 5, for a fixed $\boldsymbol{A}$, we derive the directional (Gateaux) derivative of the loss with respect to $\boldsymbol{B}$ along an arbitrary direction $\boldsymbol{W} \in \mathbb{R}^{p \times n}$, denoted as $d_{\boldsymbol{B}}L(\boldsymbol{A}, \boldsymbol{B})\boldsymbol{W}$, i.e.

$$
d_{\boldsymbol{B}}L(\boldsymbol{A}, \boldsymbol{B})\boldsymbol{W} = \lim_{\|\boldsymbol{W}\|_F \to \boldsymbol{0}} \frac{L(\boldsymbol{A}, \boldsymbol{B} + \boldsymbol{W}) - L(\boldsymbol{A}, \boldsymbol{B})}{\|\boldsymbol{W}\|_F}.
$$

As shown in the proof of the lemma, $d_{\boldsymbol{B}}L(\boldsymbol{A}, \boldsymbol{B})\boldsymbol{W}$ is derived by writing the norm in the loss as an inner product, opening it up using linearity of inner product, dismiss second order terms in $\boldsymbol{W}$ (i.e. $O(\|\boldsymbol{W}\|^2)$) and rearrange the result as the inner product between the gradient with respect to $\boldsymbol{B}$, and the direction $\boldsymbol{W}$, which yields

$$
\begin{aligned}
d_{\boldsymbol{B}}L(\boldsymbol{A}, \boldsymbol{B})\boldsymbol{W} &= -2\operatorname{Tr}\left(\boldsymbol{W}'\left(\boldsymbol{T}_p\boldsymbol{A}'\boldsymbol{\Sigma}_{yx} - \left(\boldsymbol{S}_p \circ (\boldsymbol{A}'\boldsymbol{A})\right)\boldsymbol{B}\boldsymbol{\Sigma}_{xx}\right)\right) \\
&= -2\langle \boldsymbol{T}_p\boldsymbol{A}'\boldsymbol{\Sigma}_{yx} - \left(\boldsymbol{S}_p \circ (\boldsymbol{A}'\boldsymbol{A})\right)\boldsymbol{B}\boldsymbol{\Sigma}_{xx}, \boldsymbol{W}\rangle_F,
\end{aligned}
\tag{4}
$$

where, $\circ$ is the Hadamard product and the constant matrices $\boldsymbol{T}_p$ and $\boldsymbol{S}_p$, were defined in the beginning. Second, the same process is done in Lemma 6, to derive $d_{\boldsymbol{A}}L(\boldsymbol{A}, \boldsymbol{B})V$; the derivative of $L$ with respect to $\boldsymbol{A}$ in an arbitrary direction $\boldsymbol{V} \in \mathbb{R}^{n \times p}$, for a fixed $\boldsymbol{B}$, which is then set to zero to derive the implicit expressions for the critical points. The results are formally stated in the two following propositions.

**Proposition 1.** *For any fixed matrix $\boldsymbol{A} \in \mathbb{R}^{n \times p}$ the function $L(\boldsymbol{A}, \boldsymbol{B})$ is convex in the coefficients of $\boldsymbol{B}$ and attains its minimum for any $\boldsymbol{B}$ satisfying the equation*

$$
(\boldsymbol{S}_p \circ (\boldsymbol{A}'\boldsymbol{A}))\boldsymbol{B}\boldsymbol{\Sigma}_{xx} = \boldsymbol{T}_p\boldsymbol{A}'\boldsymbol{\Sigma}_{yx},
\tag{5}
$$

*where $\circ$ is the Hadamard (element-wise) product operator, and $\boldsymbol{S}_p$ and $\boldsymbol{T}_p$ are constant matrices defined in the previous section. Further, if $\boldsymbol{A}$ has no zero column, then $L(\boldsymbol{A}, \boldsymbol{B})$ is strictly convex in $\boldsymbol{B}$ and has a unique minimum when the critical $\boldsymbol{B}$ is*

$$
\boldsymbol{B} = \hat{\boldsymbol{B}}(\boldsymbol{A}) = (\boldsymbol{S}_p \circ (\boldsymbol{A}'\boldsymbol{A}))^{-1}\boldsymbol{T}_p\boldsymbol{A}'\boldsymbol{\Sigma}_{yx}\boldsymbol{\Sigma}_{xx}^{-1},
\tag{6}
$$

*and in the autoencoder case it becomes*

$$
\boldsymbol{B} = \hat{\boldsymbol{B}}(\boldsymbol{A}) = (\boldsymbol{S}_p \circ (\boldsymbol{A}'\boldsymbol{A}))^{-1}\boldsymbol{T}_p\boldsymbol{A}'.
\tag{6'}
$$

*The proof is given in appendix A.2.*

*Remark* 1. Note that as long as $\boldsymbol{A}$ has no zero column, $\boldsymbol{S}_p \circ (\boldsymbol{A}'\boldsymbol{A})$ is nonsingular (we will explain the reason soon). In practice, $\boldsymbol{A}$ with zero columns can always be avoided by nudging the zero columns of $\boldsymbol{A}$ during the gradient decent process.

**Proposition 2.** *For any fixed matrix $\boldsymbol{B} \in \mathbb{R}^{p \times n}$ the function $L(\boldsymbol{A}, \boldsymbol{B})$ is a convex function in $\boldsymbol{A}$. Moreover, for a fixed $\boldsymbol{B}$, the matrix $\boldsymbol{A}$ that satisfies*

$$
\boldsymbol{A}\left(\boldsymbol{S}_p \circ (\boldsymbol{B}\boldsymbol{\Sigma}_{xx}\boldsymbol{B}')\right) = \boldsymbol{\Sigma}_{yx}\boldsymbol{B}'\boldsymbol{T}_p
\tag{7}
$$

*is a critical point of $L(\boldsymbol{A}, \boldsymbol{B})$.*

*The proof is given in appendix A.3.*

The pair $(\boldsymbol{A}, \boldsymbol{B})$ is a critical point of $L$ if they make $d_{\boldsymbol{B}}L(\boldsymbol{A}, \boldsymbol{B})\boldsymbol{W}$ and $d_{\boldsymbol{A}}L(\boldsymbol{A}, \boldsymbol{B})\boldsymbol{V}$ zero for any pair of directions $(\boldsymbol{V}, \boldsymbol{W})$. Therefore, the implicit equations for critical points are given below, next to the ones derived by Baldi & Hornik (1989) for $\tilde{L}(\boldsymbol{A}, \boldsymbol{B})$.

For $\tilde{L}(\boldsymbol{A}, \boldsymbol{B})$:

$$\boldsymbol{A}'\boldsymbol{A}\boldsymbol{B}\boldsymbol{\Sigma}_{xx} = \boldsymbol{A}'\boldsymbol{\Sigma}_{yx},$$
$$\boldsymbol{A}\boldsymbol{B}\boldsymbol{\Sigma}_{xx}\boldsymbol{B}' = \boldsymbol{\Sigma}_{yx}\boldsymbol{B}'.$$

For $L(\boldsymbol{A}, \boldsymbol{B})$:

$$(\boldsymbol{S}_p \circ (\boldsymbol{A}'\boldsymbol{A}))\boldsymbol{B}\boldsymbol{\Sigma}_{xx} = \boldsymbol{T}_p\boldsymbol{A}'\boldsymbol{\Sigma}_{yx},$$
$$\boldsymbol{A}(\boldsymbol{S}_p \circ (\boldsymbol{B}\boldsymbol{\Sigma}_{xx}\boldsymbol{B}')) = \boldsymbol{\Sigma}_{yx}\boldsymbol{B}'\boldsymbol{T}_p.$$

*Remark* 2. Notice the similarity, and the difference only being the presence of Hadamard product by $\boldsymbol{S}_p$ in the left and by diagonal $\boldsymbol{T}_p$ in the right. Therefore, practically, the added computational cost of evaluating the gradients is negligible compare to that of MSE loss.

The next step is to determine the structure of $(\boldsymbol{A}, \boldsymbol{B})$ that satisfies the above equations, and find the subset of those solutions that account for local minima. For the original loss, the first expression $(\boldsymbol{A}'\boldsymbol{A}\boldsymbol{B}\boldsymbol{\Sigma}_{xx} = \boldsymbol{A}'\boldsymbol{\Sigma}_{yx})$ is used to solve for $\boldsymbol{B}$ and put it in the second to derive an expression solely based on $\boldsymbol{A}$. Obviously, in order to solve the first expression for $\boldsymbol{B}$, two cases are considered separately: the case where $\boldsymbol{A}$ is of full rank $p$, so $\boldsymbol{A}'\boldsymbol{A}$ is invertible, and the case of $\boldsymbol{A}$ being of rank $r < p$. Here we do the same but there is a twist; for us there is only one case. The reason is as long as (not necessarily full rank) $\boldsymbol{A}$ has no zero column, $\boldsymbol{S}_p \circ (\boldsymbol{A}'\boldsymbol{A})$ is positive definite and hence, invertible. This is discussed in detail in Lemma 2 and we briefly explain it here. As shown in the lemma, $\boldsymbol{S}_p$ is positive definite and by Shur product theorem for any $\boldsymbol{A}$ (of any rank), $\boldsymbol{S}_p \circ (\boldsymbol{A}'\boldsymbol{A})$ is positive semidefinite. However, as a result of Oppenheim inequality (Horn & Johnson (2012), Thm 7.8.16), that in our case translates to $\det(\boldsymbol{S}_p)\prod_i(\boldsymbol{A}'\boldsymbol{A})_{ii} \le \det(\boldsymbol{S}_p \circ (\boldsymbol{A}'\boldsymbol{A}))$, as long as $\boldsymbol{A}$ has no zero column, $\prod_i(\boldsymbol{A}'\boldsymbol{A})_{ii} > 0$ and therefore, $\det(\boldsymbol{S}_p \circ (\boldsymbol{A}'\boldsymbol{A})) > 0$. Here, we assume $\boldsymbol{A}$ of any rank $r \le p$ has no zero column (since this can be easily avoided in practice) and consider $\boldsymbol{S}_p \circ (\boldsymbol{A}'\boldsymbol{A})$ to be always invertible. Therefore, $(\boldsymbol{A}, \boldsymbol{B})$ define a critical point of losses $\tilde{L}$ and $L$ if

For $\tilde{L}(\boldsymbol{A}, \boldsymbol{B})$ and full rank $\boldsymbol{A}$:

$$\boldsymbol{B} = \hat{\boldsymbol{B}}(\boldsymbol{A}) = (\boldsymbol{A}'\boldsymbol{A})^{-1}\boldsymbol{A}'\boldsymbol{\Sigma}_{yx}\boldsymbol{\Sigma}_{xx}^{-1},$$
$$\boldsymbol{A}\boldsymbol{B}\boldsymbol{\Sigma}_{xx}\boldsymbol{B}' = \boldsymbol{\Sigma}_{yx}\boldsymbol{B}'.$$

For $L(\boldsymbol{A}, \boldsymbol{B})$ and no zero column $\boldsymbol{A}$:

$$\boldsymbol{B} = \hat{\boldsymbol{B}}(\boldsymbol{A}) = (\boldsymbol{S}_p \circ (\boldsymbol{A}'\boldsymbol{A}))^{-1}\boldsymbol{T}_p\boldsymbol{A}'\boldsymbol{\Sigma}_{yx}\boldsymbol{\Sigma}_{xx}^{-1},$$
$$\boldsymbol{A}(\boldsymbol{S}_p \circ (\boldsymbol{B}\boldsymbol{\Sigma}_{xx}\boldsymbol{B}')) = \boldsymbol{\Sigma}_{yx}\boldsymbol{B}'\boldsymbol{T}_p.$$

Before, we state the main theorem we need the following definitions. First, a rectangular permutation matrix $\boldsymbol{\Pi}_r \in \mathbb{R}^{r \times p}$ is a matrix that each column consists of at most one nonzero element with the value 1. If the rank of $\boldsymbol{\Pi}_r$ is $r$ with $r < p$ then clearly, $\boldsymbol{\Pi}_r$ has $p - r$ zero columns. Also, by taking away those zero columns the resultant $r \times r$ submatrix of $\boldsymbol{\Pi}_r$ is a standard square permutation matrix.

Second, under the conditions provided in Assumption 1, the matrix $\boldsymbol{\Sigma} := \boldsymbol{\Sigma}_{yx}\boldsymbol{\Sigma}_{xx}^{-1}\boldsymbol{\Sigma}_{xy}$ has an eigenvalue decomposition $\boldsymbol{\Sigma} = \boldsymbol{U}\boldsymbol{\Lambda}\boldsymbol{U}'$, where the $i^{th}$ column of $\boldsymbol{U}$, denoted as $\boldsymbol{u}_i$, is an eigenvector of $\boldsymbol{\Sigma}$ corresponding to the $i^{th}$ largest eigenvalue of $\boldsymbol{\Sigma}$, denoted as $\lambda_i$. Also, $\boldsymbol{\Lambda} = \text{diag}(\lambda_1, \cdots, \lambda_n)$ is the diagonal vector of ordered eigenvalues of $\boldsymbol{\Sigma}$, with $\lambda_1 > \lambda_2 > \cdots > \lambda_n > 0$. We use the following notation to organize a subset of eigenvectors of $\boldsymbol{\Sigma}$ into a rectangular matrix. Let for any $r \le p$, $\mathbb{I}_r = \{i_1, \cdots, i_r\}(1 \le i_1 < \cdots < i_r < n)$ be any *ordered* $r-$index set. Define $\boldsymbol{U}_{\mathbb{I}_r} \in \mathbb{R}^{n \times p}$ as $\boldsymbol{U}_{\mathbb{I}_r} = [\boldsymbol{u}_{i_1}, \cdots, \boldsymbol{u}_{i_r}]$. That is the columns of $\boldsymbol{U}_{\mathbb{I}_r}$ are the ordered orthonormal eigenvectors of $\boldsymbol{\Sigma}$ associated with eigenvalues $\lambda_{i_1} < \cdots < \lambda_{i_r}$. Clearly, when $r = p$, we have $\boldsymbol{U}_{\mathbb{I}_r} = [\boldsymbol{u}_{i_1}, \cdots, \boldsymbol{u}_{i_p}]$ corresponding to an $p-$index set $\mathbb{I}_p = \{i_1, \cdots, i_p\}(1 \le i_1 < \cdots < i_p < n)$. Similarly, we define $\boldsymbol{\Lambda}_{\mathbb{I}_r} \in \mathbb{R}^{p \times p}$ as $\boldsymbol{\Lambda}_{\mathbb{I}_r} = \text{diag}(\lambda_{i_1}, \cdots, \lambda_{i_r})$.

**Theorem 1.** *Let $\boldsymbol{A} \in \mathbb{R}^{n \times p}$ and $\boldsymbol{B} \in \mathbb{R}^{p \times n}$ such that $\boldsymbol{A}$ is of rank $r \le p$. Under the conditions provided in Assumption 1 and the above notation, The matrices $\boldsymbol{A}$ and $\boldsymbol{B}$ define a critical point of $L(\boldsymbol{A}, \boldsymbol{B})$ if and only if for any given $r-$index set $\mathbb{I}_r$, and a nonsingular diagonal matrix $\boldsymbol{D} \in \mathbb{R}^{r \times r}$, $\boldsymbol{A}$ and $\boldsymbol{B}$ are of the form*

$$\boldsymbol{A} = \boldsymbol{U}_{\mathbb{I}_r}\boldsymbol{C}\boldsymbol{D}, \tag{8}$$

$$\boldsymbol{B} = \hat{\boldsymbol{B}}(\boldsymbol{A}) = \boldsymbol{D}^{-1}\boldsymbol{\Pi}_{\boldsymbol{C}}\boldsymbol{U}_{\mathbb{I}_r}'\boldsymbol{\Sigma}_{yx}\boldsymbol{\Sigma}_{xx}^{-1}, \tag{9}$$

*where, $\boldsymbol{C} \in \mathbb{R}^{r \times p}$ is of of full rank $r$ with nonzero and normalized columns such that $\boldsymbol{\Pi}_{\boldsymbol{C}} := (\boldsymbol{S}_p \circ (\boldsymbol{C}'\boldsymbol{C}))^{-1}\boldsymbol{T}_p\boldsymbol{C}'$ is a rectangular permutation matrix of rank $r$ and $\boldsymbol{C}\boldsymbol{\Pi}_{\boldsymbol{C}} = \boldsymbol{I}_r$. For all $1 \le r \le p$, such $\boldsymbol{C}$ always exists. In particular, if matrix $\boldsymbol{A}$ is of full rank $p$, i.e. $r = p$, the two given conditions on $\boldsymbol{\Pi}_{\boldsymbol{C}}$ are satisfied iff the invertible matrix $\boldsymbol{C}$ is any squared $p \times p$ permutation matrix $\boldsymbol{\Pi}$. In this case $(\boldsymbol{A}, \boldsymbol{B})$ define a critical point of $L(\boldsymbol{A}, \boldsymbol{B})$ iff they are of the form*

$$\boldsymbol{A} = \boldsymbol{U}_{\mathbb{I}_p}\boldsymbol{\Pi}\boldsymbol{D}, \tag{10}$$

$$\boldsymbol{B} = \hat{\boldsymbol{B}}(\boldsymbol{A}) = \boldsymbol{D}^{-1}\boldsymbol{\Pi}'\boldsymbol{U}'_{\mathbb{I}_p}\boldsymbol{\Sigma}_{yx}\boldsymbol{\Sigma}^{-1}_{xx}. \tag{11}$$

*The proof is given in appendix A.4.*

*Remark* 3. The above theorem provides explicit equations for the critical points of the loss surface in terms of the rank of the decoder matrix $A$ and the eigenvectors of $\boldsymbol{\Sigma}$. This explicit structure allows us to further analyze the loss surface and its local/global minima.

Here, we provide a proof sketch for the above theorem to make the claims more clear. Again as a reminder, the EVD of $\boldsymbol{\Sigma} := \boldsymbol{\Sigma}_{yx}\boldsymbol{\Sigma}^{-1}_{xx}\boldsymbol{\Sigma}_{xy}$ is $\boldsymbol{\Sigma} = \boldsymbol{U}\boldsymbol{\Lambda}\boldsymbol{U}'$. For both $\tilde{L}$ and $L$, the corresponding $\hat{\boldsymbol{B}}(\boldsymbol{A})$ is replaced by $\boldsymbol{B}$ on the RHS of critical point equations. For the loss $L(A,B)$, as shown in the proof of the theorem, results in the following identity

$$\boldsymbol{U}'\boldsymbol{A}\left(\boldsymbol{S}_p \circ \left(\hat{\boldsymbol{B}}\boldsymbol{\Sigma}_{xx}\hat{\boldsymbol{B}}'\right)\right)\boldsymbol{A}'\boldsymbol{U} = \boldsymbol{\Lambda}\boldsymbol{\Delta}, \tag{12}$$

where $\boldsymbol{\Delta} := \boldsymbol{U}'\boldsymbol{A}\boldsymbol{T}_p(\boldsymbol{S}_p \circ (\boldsymbol{A}'\boldsymbol{A}))^{-1}\boldsymbol{T}_p\boldsymbol{A}'\boldsymbol{U}$ is symmetric and positive semidefinite. The LHS of eq. (12) is symmetric so the RHS is symmetric too, so $\boldsymbol{\Lambda}\boldsymbol{\Delta} = (\boldsymbol{\Lambda}\boldsymbol{\Delta})' = \boldsymbol{\Delta}'\boldsymbol{\Lambda}' = \boldsymbol{\Delta}\boldsymbol{\Lambda}$. Therefore, $\boldsymbol{\Delta}$ commutes with the diagonal matrix of eigenvalues $\boldsymbol{\Lambda}$. Since eigenvalues are assumed to be distinct, $\boldsymbol{\Delta}$ has to be diagonal as well. By Lemma 2 $\boldsymbol{T}_p(\boldsymbol{S}_p \circ (\boldsymbol{A}'\boldsymbol{A}))^{-1}\boldsymbol{T}_p$ is positive definite and $\boldsymbol{U}$ is an orthogonal matrix. Therefore, $r = \text{rank}(\boldsymbol{A}) = \text{rank}(\boldsymbol{\Delta}) = \text{rank}(\boldsymbol{U}'\boldsymbol{\Delta}\boldsymbol{U})$, which implies that the diagonal matrix $\boldsymbol{\Delta}$, has $r$ nonzero and *positive* diagonal entries. There exists an $r-$index set $\mathbb{I}_r$ corresponding to the nonzero diagonal elements of $\boldsymbol{\Delta}$. Forming a diagonal matrix $\boldsymbol{\Delta}_{\mathbb{I}_r} \in \mathbb{R}^{r\times r}$ by filling its diagonal entries (in order) by the nonzero diagonal elements of $\boldsymbol{\Delta}$, we have

$$\boldsymbol{U}\boldsymbol{\Delta}\boldsymbol{U}' = \boldsymbol{U}_{\mathbb{I}_r}\boldsymbol{\Delta}_{\mathbb{I}_r}\boldsymbol{U}'_{\mathbb{I}_r} \xLongrightarrow{\text{Def of } \boldsymbol{\Delta}}$$
$$\boldsymbol{A}\boldsymbol{T}_p(\boldsymbol{S}_p \circ (\boldsymbol{A}'\boldsymbol{A}))^{-1}\boldsymbol{T}_p\boldsymbol{A}' = \boldsymbol{U}_{\mathbb{I}_r}\boldsymbol{\Delta}_{\mathbb{I}_r}\boldsymbol{U}'_{\mathbb{I}_r}, \tag{13}$$

which indicates that the matrix $\boldsymbol{A}$ has the same column space as $\boldsymbol{U}_{\mathbb{I}_r}$. Therefore, there exists a full rank matrix $\bar{\boldsymbol{C}} \in \mathbb{R}^{r\times p}$ such that $\boldsymbol{A} = \boldsymbol{U}_{\mathbb{I}_r}\bar{\boldsymbol{C}}$. Since $\boldsymbol{A}$ has no zero column, $\bar{\boldsymbol{C}}$ has no zero column. Further, by normalizing the columns of $\bar{\boldsymbol{C}}$ we can write $\boldsymbol{A} = \boldsymbol{U}_{\mathbb{I}_r}\boldsymbol{C}\boldsymbol{D}$, where $\boldsymbol{D} \in \mathbb{R}^{p\times p}$ is diagonal that contains the norms of columns of $\bar{\boldsymbol{C}}$.

Baldi & Hornik (1989) did something similar for full rank $\boldsymbol{A}$ for the loss $\tilde{L}$ to derive $(\boldsymbol{A}_{\tilde{L}} = \boldsymbol{U}_{\mathbb{I}_p}\tilde{\boldsymbol{C}})$. But their $\tilde{\boldsymbol{C}}$ can be any invertible $p \times p$ matrix. However, in our case, the matrix $\boldsymbol{C} \in \mathbb{R}^{r\times p}$ corresponding to rank $r \le p$ matrix $\boldsymbol{A}$, has to satisfy eq. (13) by replacing $\boldsymbol{A}$ by $\boldsymbol{U}_{\mathbb{I}_r}\boldsymbol{C}\boldsymbol{D}$ and eq. (12) by replacing $\hat{\boldsymbol{B}}(\boldsymbol{A})$ by $\hat{\boldsymbol{B}}(\boldsymbol{U}_{\mathbb{I}_r}\boldsymbol{C}\boldsymbol{D})$. In the case of Baldi & Hornik (1989), for the original loss $\tilde{L}$, equations similar to eq. (13) and eq. (12) appear but they are are satisfied trivially by any invertible matrix $\tilde{\boldsymbol{C}}$. Simplifying those equations by using $\boldsymbol{A} = \boldsymbol{U}_{\mathbb{I}_r}\boldsymbol{C}\boldsymbol{D}$ after some algebraic manipulation results in the following two conditions for $\boldsymbol{C}$:

$$\boldsymbol{C}\boldsymbol{T}_p\left(\boldsymbol{S}_p \circ (\boldsymbol{C}'\boldsymbol{C})\right)^{-1}\boldsymbol{T}_p\boldsymbol{C}' = \boldsymbol{\Delta}_{\mathbb{I}_r} \text{ and} \tag{14}$$
$$\boldsymbol{C}\left(\boldsymbol{S}_p \circ \left((\boldsymbol{S}_p \circ (\boldsymbol{C}'\boldsymbol{C}))^{-1}\boldsymbol{T}_p\boldsymbol{C}'\boldsymbol{\Lambda}_{\mathbb{I}_r}\boldsymbol{C}\boldsymbol{T}_p(\boldsymbol{S}_p \circ (\boldsymbol{C}'\boldsymbol{C}))^{-1}\right)\right)\boldsymbol{C}' = \boldsymbol{\Lambda}_{\mathbb{I}_r}\boldsymbol{\Delta}_{\mathbb{I}_r}. \tag{15}$$

As detailed in proof of Theorem 1, solving for $\boldsymbol{C}$ leads to its specific structure as laid out in the theorem.

*Remark* 4. Note that when $\boldsymbol{A}$ is of rank $r < p$ with no zero columns then the invariant matrix $\boldsymbol{C}$ is not necessarily a rectangular permutation matrix but $\boldsymbol{\Pi}_C := (\boldsymbol{S}_p \circ (\boldsymbol{C}'\boldsymbol{C}))^{-1}\boldsymbol{T}_p\boldsymbol{C}'$ is a rectangular permutation matrix with $\boldsymbol{C}\boldsymbol{\Pi}_C = \boldsymbol{I}_r$. It is only when $r = p$ that the invariant matrix $\boldsymbol{C}$ becomes a permutation matrix. Nevertheless, as we show in the following corollary, the global map is always $\forall r \le p : \boldsymbol{G} = \boldsymbol{A}\boldsymbol{B} = \boldsymbol{U}_{\mathbb{I}_r}\boldsymbol{U}'_{\mathbb{I}_r}\boldsymbol{\Sigma}_{yx}\boldsymbol{\Sigma}^{-1}_{xx}$. It is possible to find further structure (in terms of block matrices) for the invariant matrix $\boldsymbol{C}$ when $r < p$. However, this is not necessary as we soon show that all rank deficient matrix $\boldsymbol{A}$s are saddle points for the loss and ideally should be passed by during the gradient decent process. Based on some numerical results our conjecture is that when $r < p$ the matrix $\boldsymbol{C}$ can only start with a $r \times k$ rectangular permutation matrix of rank $r$ with $r \le k \le p$ and the rest of $p - k$ columns of $\boldsymbol{C}$ is arbitrary as long as none of the columns are identically zero.

**Corollary 1.** *Let* $(\boldsymbol{A}, \boldsymbol{B})$ *be a critical point of* $L(\boldsymbol{A}, \boldsymbol{B})$ *under the conditions provided in Assumption 1 and* $\text{rank}\boldsymbol{A} = r \le p$. *Then the following are true*

    *1. The matrix* $\boldsymbol{B}\boldsymbol{\Sigma}_{xx}\boldsymbol{B}'$ *is a* $p \times p$ *diagonal matrix of rank* $r$.

2. *For all $1 \leq r \leq p$, for any critical pair $(\boldsymbol{A}, \boldsymbol{B})$, the global map $\boldsymbol{G} := \boldsymbol{A}\boldsymbol{B}$ becomes*

$$\boldsymbol{G} = \boldsymbol{U}_{\mathbb{I}_r} \boldsymbol{U}'_{\mathbb{I}_r} \boldsymbol{\Sigma}_{yx} \boldsymbol{\Sigma}_{xx}^{-1}. \tag{16}$$

*For the autoencoder case $(\boldsymbol{Y} = \boldsymbol{X})$ the global map is simply $\boldsymbol{G} = \boldsymbol{U}_{\mathbb{I}_r} \boldsymbol{U}'_{\mathbb{I}_r}$.*

3. $(\boldsymbol{A}, \boldsymbol{B})$ *is also the critical point of the classical loss $\tilde{L}(\boldsymbol{A}, \boldsymbol{B}) = \sum_{i=1}^{p} \|\boldsymbol{Y} - \boldsymbol{A}\boldsymbol{B}\boldsymbol{X}\|_F^2$.*

*The proof is given in appendix A.5.*

*Remark* 5. The above corollary implies that $L(\boldsymbol{A}, \boldsymbol{B})$ not only does not add any extra critical point compare to the original loss $\tilde{L}(\boldsymbol{A}, \boldsymbol{B})$, it provides the same global map $\boldsymbol{G} := \boldsymbol{A}\boldsymbol{B}$. It only limits the structure of the invariance matrix $\boldsymbol{C}$ as described in Theorem 1 so that the decoder matrix $\boldsymbol{A}$ can recover the exact eigenvectors of $\boldsymbol{\Sigma}$.

**Lemma 1.** *The loss function $L(\boldsymbol{A}, \boldsymbol{B})$ can be written as*

$$L(\boldsymbol{A}, \boldsymbol{B}) = p \operatorname{Tr}(\boldsymbol{\Sigma}_{yy}) - 2 \operatorname{Tr}\left(\boldsymbol{A}\boldsymbol{T}_p \boldsymbol{B}\boldsymbol{\Sigma}_{xy}\right) + \operatorname{Tr}\left(\boldsymbol{B}'\left(\boldsymbol{S}_p \circ (\boldsymbol{A}'\boldsymbol{A})\right)\boldsymbol{B}\boldsymbol{\Sigma}_{xx}\right). \tag{17}$$

*The above identity shows that the number of matrix operations required for computing the loss $L(\boldsymbol{A}, \boldsymbol{B})$ is constant and thereby independent of the value of $p$.*

*The proof is given in appendix A.6.*

**Theorem 2.** *Let $\boldsymbol{A}^* \in \mathbb{R}^{n \times p}$ and $\boldsymbol{B}^* \in \mathbb{R}^{p \times n}$ such that $\boldsymbol{A}^*$ is of rank $r \leq p$. Under the conditions provided in Assumption 1, $(\boldsymbol{A}^*, \boldsymbol{B}^*)$ define a local minima of the proposed loss function iff they are of the form*

$$\boldsymbol{A}^* = \boldsymbol{U}_{1:p} \boldsymbol{D}_p \tag{18}$$

$$\boldsymbol{B}^* = \boldsymbol{D}_p^{-1} \boldsymbol{U}'_{1:p} \boldsymbol{\Sigma}_{yx} \boldsymbol{\Sigma}_{xx}^{-1}, \tag{19}$$

*where the $i^{th}$ column of $\boldsymbol{U}_{1:p}$ is a unit eigenvector of $\boldsymbol{\Sigma} := \boldsymbol{\Sigma}_{yx} \boldsymbol{\Sigma}_{xx}^{-1} \boldsymbol{\Sigma}_{xy}$ corresponding the $i^{th}$ largest eigenvalue and $\boldsymbol{D}_p$ is a diagonal matrix with nonzero diagonal elements. In other words, $\boldsymbol{A}^*$ contains ordered unnormalized eigenvectors of $\boldsymbol{\Sigma}$ corresponding to the $p$ largest eigenvalues. Moreover, all the local minima are global minima with the value of the loss function at those global minima being*

$$L(\boldsymbol{A}^*, \boldsymbol{B}^*) = p \operatorname{Tr}(\boldsymbol{\Sigma}_{yy}) - \sum_{i=1}^{p} (p - i + 1) \lambda_i, \tag{20}$$

*where $\lambda_i$ is the $i^{th}$ largest eigenvalue of $\boldsymbol{\Sigma}$.*

*The proof is given in appendix A.7.*

*Remark* 6. Finally, the second and third assumptions we made in the beginning in Assumption 1 can be relaxed by requiring only $\boldsymbol{\Sigma}_{xx}$ to be full rank. The output data can have a different dimension than the input. That is $\boldsymbol{Y} \in \mathbb{R}^{n \times m}$ and $\boldsymbol{X} \in \mathbb{R}^{n' \times m}$, where $n \neq n'$. The reason is that the given loss function structurally is very similar to MSE loss and can be represented as a Frobenius norm on the space of $n \times m$ matrices. In this case the covariance matrix $\boldsymbol{\Sigma} := \boldsymbol{\Sigma}_{yx} \boldsymbol{\Sigma}_{xx}^{-1} \boldsymbol{\Sigma}_{xy}$ is still $n \times n$. Clearly, for under-constrained systems with $n < n'$ the full rank assumption of $\boldsymbol{\Sigma}$ holds. For the overdetermined case, where $n' > n$ the second and third assumptions in Assumption 1 can be relaxed: we only require $\boldsymbol{\Sigma}_{xx}$ to be full rank since this is the only matrix that is inverted in the theorems. Note that if $p > \min(n', n)$ then $\boldsymbol{\Lambda}_{\mathbb{I}_p}$: the $p \times p$ diagonal matrix of eigenvalues of $\boldsymbol{\Sigma}$ for a $p$-index-set $\mathbb{I}_p$ bounds to have some zeros and will be say rank $r < p$, which in turn, results in the encoder $\boldsymbol{A}$ with rank $r$. However, the Theorem 1 is proved for encoder of any rank $r \leq p$. Finally, following theorem 2 then the first $r$ columns of the encoder $\boldsymbol{A}$ converges to ordered eigenvectors of $\boldsymbol{\Sigma}$ while the $p - r$ remaining columns span the kernel space of $\boldsymbol{\Sigma}$. Moreover, $\boldsymbol{\Sigma}$ need not to have distinct eigenvectors. In this case $\boldsymbol{\Delta}_{\mathbb{I}_r}$ becomes a block diagonal matrix, where the blocks correspond to identical eigenvalues $\boldsymbol{\Sigma}_{\mathbb{I}_r}$. In this case, the corresponding eigenvectors in $\boldsymbol{A}^*$ are not unique but they span the respective eigenspace.

## 5 EXPERIMENTS

### 5.1 EXPERIMENTAL SETUP

**LAEs with Two Loss functions** We will verify the loss function $L(\boldsymbol{A}, \boldsymbol{B})$ defined in eq. (1) by setting the input matrix $\boldsymbol{X} \in \mathbb{R}^{n \times m}$ equal to the output matrix $\boldsymbol{Y} \in \mathbb{R}^{n \times m}$ ($\boldsymbol{Y} = \boldsymbol{X}$), where the

linear autodecoder (LAE) becomes a solution to PCA. In order for comparison, we train another LAE using the MSE loss $\tilde{L}(\tilde{A}, \tilde{B})$ defined as $\tilde{L}(\tilde{A}, \tilde{B}) = \left\| Y - \tilde{A}\tilde{B}X \right\|_F^2$, where $Y = X$ is also applied in our experiments.

The weights of networks are initialized to random numbers with a small enough standard deviation ($10^{-7}$ in our case). We choose to use the Adam optimizer with a scheduled learning rate (starting from $10^{-3}$ and ending with $10^{-6}$ in our case), which empirically benefits the optimization process. The two training processes are stopped at the same iteration at which one of the models firstly finds all of the principal directions. As a side note, we feed all data samples to the network at one time with batch size equal to $m$, although mini-batch implementations are apparently amendable.

**Evaluation Metrics**   We use the classical PCA approach to get the ground truth principal direction matrix $A^* \in \mathbb{R}^{n \times p}$, by conducting Eigen Value Decomposition (EVD) to $XX' \in \mathbb{R}^{n \times n}$ or Singular Value Decomposition (SVD) to $X \in \mathbb{R}^{n \times m}$. As a reminder, $A \in \mathbb{R}^{n \times p}$ stands for the decoder weight matrix of an trained LAE given a loss function $L$. To measure the distance between $A^*$ and $A$, we propose an absolute cosine similarity (**ACS**) matrix inspired by mutual coherence (Donoho et al., 2005), which is defined as:

$$\mathbf{ACS}_{ij} = \frac{|\langle A_i^*, A_j \rangle|}{\|A_i^*\| \cdot \|A_j\|}, \tag{21}$$

where $A_i^* \in \mathbb{R}^{n \times 1}$ denotes the $i^{th}$ ground truth principal direction, and $A_j \in \mathbb{R}^{n \times 1}$ denotes the $j^{th}$ column of the decoder $A$, $i, j = 1, 2, \ldots, p$. The elements of $\mathbf{ACS} \in \mathbb{R}^{p \times p}$ in eq. (21) take values between [0,1], measuring pair-wise similarity across two sets of vectors. The absolute value absorbs the sign ambiguity of principal directions.

The performances of LAEs are evaluated by defining the following metrics:

$$\mathbf{Ratio}_{TP} = \sum_{i=1}^{p} I[\mathbf{ACS}_{ii} > 1 - \epsilon]/p \tag{22}$$

$$\mathbf{Ratio}_{FP} = \sum_{\substack{i,j=1 \\ i \neq j}}^{p} I[\mathbf{ACS}_{ij} > 1 - \epsilon]/p, \text{ and} \tag{23}$$

$$\mathbf{Ratio}_{Total} = \mathbf{Ratio}_{TP} + \mathbf{Ratio}_{FP}, \tag{24}$$

where $I$ is the indicator function and $\epsilon$ is a manual tolerance threshold ($\epsilon = 0.01$ in our case). If two vectors have absolute cosine similarity over $1 - \epsilon$, they are deemed equal. Considering some columns of decoder may be correct principal directions but not in the right order, we introduce $Ratio_{TP}$ and $Ratio_{FP}$ in eqs. (22) and (23) to check the ratio of correct in-place and out-of-place principal directions respectively. Then $Ratio_{Total}$ in eq. (24) measures the total ratio of the correctly obtained principal directions by the LAE regardless of the order.

**Datasets**   As a proof-of-concept, both synthetic data and real data are used. For the synthetic data, 2000 zero-centered data samples are generated from a 1000-dimension zero mean multivariate normal distribution with the covariance matrix being $\mathrm{diag}(\mathbb{N}_p)$. For the real data, we choose to use MNIST dataset (LeCun et al., 1998), which includes 60,000 grayscale handwritten digits images, each of dimension $28 \times 28 = 784$.

## 5.2   Evaluation and Analysis

**Synthetic Data Experiments**   In our experiment, $p$, the number of desired principal components (PCs), is set to 100, i.e. the dimension is to be reduced from 1000 to 100. Figures 1 and 2 demonstrate a few conclusions. First, during the training process, the *loss ratio* of both losses continuously decreases to 1, i.e. they both converge to the optimal loss value. However, when both get close enough, $L$ require more iterations since the optimizer is forced to find the right directions: it fully converges only after it has found all the principal directions in the right order.

Second, using the loss $L$ results in finding more and more correct principal directions, with $Ratio_{TP}$ continuously rising; and ultimately affords all correct and ordered principal directions,

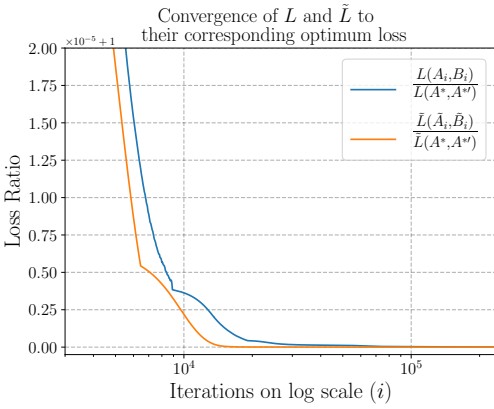
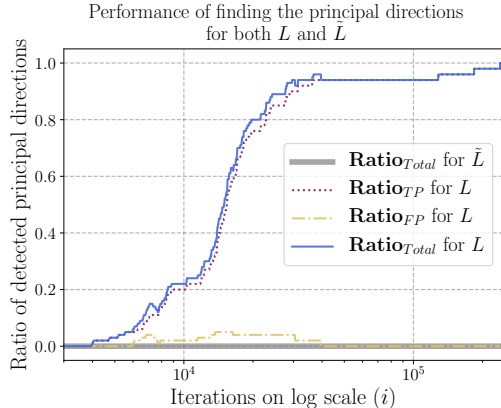

Figure 1: Convergence of losses to their corresponding optimal loss. Note that the correct shift and scaling of the $y$-axis tick values is printed at the top left corner of the figure.

Figure 2: Performance of both losses $L$ and $\tilde{L}$ in finding the principal directions at the columns of their respective decoders.

with $Ratio_{TP}$ ending with 100%. Notice that occasionally and temporarily, some of the principal directions is found but not at their correct position, which is indicated by the rise of $Ratio_{FP}$ in the figure. However, as optimization continues they are shifted to the right column, which results in $Ratio_{FP}$ going back to zero, and $Ratio_{TP}$ reaching one. As for $\tilde{L}$, it fails to identify any principal directions; both $Ratio_{TP}$ and $Ratio_{FP}$ for $\tilde{L}$ stay at 0, which indicates that none of the columns of the decoder $\tilde{A}$, aligns with any principal direction.

Third, as shown in the figure, while the optimizer finds almost all the principal directions rather quickly, it requires much more iterations to find some final ones. This is because some eigenvalues in the empirical covariance matrix of the finite 2000 samples become very close (the difference becomes less than 1). Therefore, the loss has to get very close to the optimal loss, making the gradient of the loss hard to distinguish between the two.

**Real Data: MNIST Experiments** We set the number of principal components (PCs) as 100, i.e., the dimension is to be reduced from 784 to 100. We also try to reconstruct with the top-10 columns found in this case. As in Fig. 3, the reconstruction performance of $L$ is consistently better than $\tilde{L}$. That also reflects that $\tilde{L}$ does not identify PCs, while $L$ is directly applicable to performing PCA without bells and whistles.

## 6 CONCLUSION

In this paper, we have introduced a loss function for performing principal component analysis and linear regression using linear autoencoders. We have proved that the optimizing with the given loss results in the decoder matrix converges to the exact ordered unnormalized eigenvectors of the sample covariance matrix. We have also demonstrated the claims on a synthetic data set of random samples drawn from a multivariate normal distribution and on

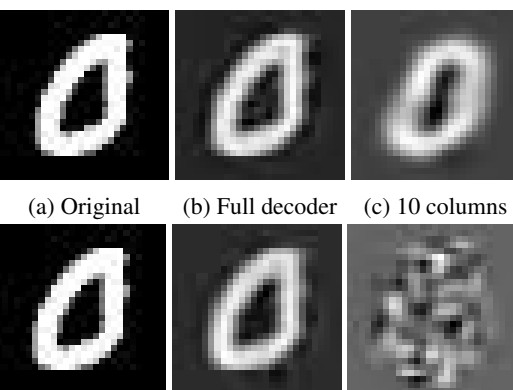

(a) Original    (b) Full decoder    (c) 10 columns

(d) Original    (e) Full decoder    (f) 10 columns

Figure 3: Real data experimental comparison in the reconstruction performance of MNIST images. First column: original image. Second column: reconstructed image using full columns of the decoder. Third column: reconstructed image using the first 10 columns of the decoder. Top row: using $L$. Bottom row: using $\tilde{L}$ .

MNIST data set. There are several possible generalizations of this approach we are currently working on. One is improving performance when the corresponding eigenvalues of two principal directions are very close and another is generalization of the loss for tensor decomposition.

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

## APPENDIX

## A    PROOFS

### A.1    PRELIMINARIES

Before we present the proof for the main theorems, the following two lemmas introduce some notations and basic relations that are required for the proofs.

**Lemma 2.** *The constant matrices $\boldsymbol{T}_p \in \mathbb{R}^{p \times p}$ and $\boldsymbol{S}_p \in \mathbb{R}^{p \times p}$ are defined as*

$$(\boldsymbol{T}_p)_{ij} = (p - i + 1)\,\delta_{ij}, \ i.e. \ \boldsymbol{T}_p = \mathrm{diag}\,(p, p - 1, \cdots, 1)\,,$$

$$(\boldsymbol{S}_p)_{ij} = p - \max(i, j) + 1, \ i.e. \ \boldsymbol{S}_p = \begin{bmatrix} p & p-1 & \cdots & 2 & 1 \\ p-1 & p-1 & \cdots & 2 & 1 \\ \vdots & \vdots & \ddots & 2 & 1 \\ 2 & 2 & 2 & 2 & 1 \\ 1 & 1 & 1 & 1 & 1 \end{bmatrix}, \ e.g. \ S_4 = \begin{bmatrix} 4 & 3 & 2 & 1 \\ 3 & 3 & 2 & 1 \\ 2 & 2 & 2 & 1 \\ 1 & 1 & 1 & 1 \end{bmatrix}.$$

*Clearly, the diagonal matrix $\boldsymbol{T}_p$ is positive definite. Another matrix that will appear in the formulation is $\hat{\boldsymbol{S}}_p := \boldsymbol{T}_p^{-1} \boldsymbol{S}_p \boldsymbol{T}_p^{-1}$*

$$\left(\hat{\boldsymbol{S}}_p\right)_{ij} = \left(\boldsymbol{T}_p^{-1} \boldsymbol{S}_p \boldsymbol{T}_p^{-1}\right)_{ij} = \frac{1}{p - \min(i, j) + 1} \ i.e. \ \boldsymbol{T}_p^{-1} \boldsymbol{S}_p \boldsymbol{T}_p^{-1} = \begin{bmatrix} \frac{1}{p} & \frac{1}{p} & \cdots & \frac{1}{p} & \frac{1}{p} \\ \frac{1}{p} & \frac{1}{p-1} & \cdots & \frac{1}{p-1} & \frac{1}{p-1} \\ \vdots & \vdots & \ddots & \vdots & \vdots \\ \frac{1}{p} & \frac{1}{p-1} & \cdots & \frac{1}{2} & \frac{1}{2} \\ \frac{1}{p} & \frac{1}{p-1} & \cdots & \frac{1}{2} & 1 \end{bmatrix},$$

$$e.g. \ \hat{\boldsymbol{S}}_4 = \begin{bmatrix} \frac{1}{4} & \frac{1}{4} & \frac{1}{4} & \frac{1}{4} \\ \frac{1}{4} & \frac{1}{3} & \frac{1}{3} & \frac{1}{3} \\ \frac{1}{4} & \frac{1}{3} & \frac{1}{2} & \frac{1}{2} \\ \frac{1}{4} & \frac{1}{3} & \frac{1}{2} & 1 \end{bmatrix}.$$

*The following properties of Hadamard product and matrices $\boldsymbol{T}_p$ and $\boldsymbol{S}_p$ are used throughout:*

1. *For any arbitrary matrix $\boldsymbol{A} \in \mathbb{R}^{n \times p}$,*

$$\sum_{i=1}^{p} \boldsymbol{I}_{i;p} = \boldsymbol{T}_p, \ and \tag{25}$$

$$\sum_{i=1}^{p} \boldsymbol{I}_{i;p} \boldsymbol{A}' \boldsymbol{A} \boldsymbol{I}_{i;p} = \boldsymbol{S}_p \circ (\boldsymbol{A}' \boldsymbol{A})\,, \tag{26}$$

   *where, $\circ$ is the Hadamard (element-wise) product.*

2. *For any matrices $\boldsymbol{M}_1, \boldsymbol{M}_2 \in \mathbb{R}^{p \times p}$ and diagonal matrices $\boldsymbol{\mathscr{D}}, \boldsymbol{\mathscr{E}} \in \mathbb{R}^{p \times p}$,*

$$\boldsymbol{\mathscr{D}}\,(\boldsymbol{M}_1 \circ \boldsymbol{M}_2)\,\boldsymbol{\mathscr{E}} = (\boldsymbol{\mathscr{D}} \boldsymbol{M}_1 \boldsymbol{\mathscr{E}}) \circ \boldsymbol{M}_2 = \boldsymbol{M}_1 \circ (\boldsymbol{\mathscr{D}} \boldsymbol{M}_2 \boldsymbol{\mathscr{E}})\,.$$

   *Moreover, if $\boldsymbol{\Pi}_1, \boldsymbol{\Pi}_2 \in \mathbb{R}^{p \times p}$ are permutation matrices then*

$$\boldsymbol{\Pi}_1\,(\boldsymbol{M}_1 \circ \boldsymbol{M}_2)\,\boldsymbol{\Pi}_2 = (\boldsymbol{\Pi}_1 \boldsymbol{M}_1 \boldsymbol{\Pi}_2) \circ (\boldsymbol{\Pi}_1 \boldsymbol{M}_2 \boldsymbol{\Pi}_2)\,.$$

3. *$\boldsymbol{S}_p$ is invertible and its inverse is a symmetric tridiagonal matrix*

$$(\boldsymbol{S}_p^{-1})_{ij} = \begin{cases} 1 & i = j = 1 \\ 2 & i = j \neq 1 \\ -1 & |i - j| = 1 \\ 0 & otherwise \end{cases}, \ i.e. \ \boldsymbol{S}_p^{-1} = \begin{bmatrix} 1 & -1 & \cdots & 0 & 0 \\ -1 & 2 & -1 & 0 & 0 \\ \vdots & \vdots & \ddots & \vdots & \vdots \\ 0 & 0 & -1 & 2 & -1 \\ 0 & 0 & 0 & -1 & 2 \end{bmatrix}.$$

4. $\boldsymbol{S}_p$ is positive definite.

5. For any matrix $\boldsymbol{A} \in \mathbb{R}^{n \times p}$, $\boldsymbol{S}_p \circ (\boldsymbol{A}'\boldsymbol{A})$ is positive semidefinite. If (not necessarily full rank) $\boldsymbol{A}$ has no zero column then $\boldsymbol{S}_p \circ (\boldsymbol{A}'\boldsymbol{A})$ is positive definite.

6. For any diagonal matrix $\boldsymbol{\mathscr{D}} \in \mathbb{R}^{p \times p}$

$$\boldsymbol{S}_p \circ \boldsymbol{\mathscr{D}} = \boldsymbol{T}_p \boldsymbol{\mathscr{D}}, \ \text{ and} \tag{27}$$

$$\hat{\boldsymbol{S}}_p \circ \boldsymbol{\mathscr{D}} = \boldsymbol{T}_p^{-1} \boldsymbol{\mathscr{D}}. \tag{28}$$

7. Let $\boldsymbol{\mathscr{D}}, \boldsymbol{\mathscr{E}} \in \mathbb{R}^{p \times p}$ be positive semidefinite matrices, where $\boldsymbol{\mathscr{E}}$ has no zero diagonal element, and $\boldsymbol{\mathscr{D}}$ is of rank $r \leq p$. Also, let for any $r \leq p$, $\mathbb{J}_r = \{i_1, \cdots, i_r\}(1 \leq i_1 < \cdots < i_r < n)$ be any ordered $r-$index set. Then $\boldsymbol{\mathscr{D}}$ and $\boldsymbol{\mathscr{E}}$ satisfy

$$\boldsymbol{\mathscr{E}}\left(\hat{\boldsymbol{S}}_p \circ \boldsymbol{\mathscr{D}}\right) = \left(\hat{\boldsymbol{S}}_p \circ \boldsymbol{\mathscr{E}}\right)\boldsymbol{\mathscr{D}},$$

if and only if, the following two conditions are satisfied:

(a) The matrix $\boldsymbol{\mathscr{D}}$ is diagonal with $p - r$ zero diagonal elements and $r$ positive diagonal elements indexed by the set $\mathbb{J}_r$. That is for any $i \in \mathbb{J}_r : (\boldsymbol{\mathscr{D}})_{ii} > 0$ and the rest of elements of $\boldsymbol{\mathscr{D}}$ are zero.

(b) For any $i, j \in \mathbb{J}_r$ and $i \neq j$ we have $(\boldsymbol{\mathscr{E}})_{i,j} = 0$.

Clearly, if $\boldsymbol{\mathscr{D}}$ is positive definite then $\mathbb{J}_r = \mathbb{N}_p$ and hence, both $\boldsymbol{\mathscr{D}}$ and $\boldsymbol{\mathscr{E}}$ are diagonal.

*Proof.* . The proof of the properties are as follows.

1. eq. (25) is trivial. For eq. (26) note that $\boldsymbol{A}\boldsymbol{I}_{i;p}$ selects the first $i$ columns of $\boldsymbol{A}$ (zeros out the rest), and similarly, $\boldsymbol{I}_{i;p}\boldsymbol{A}'$ selects the first $i$ rows of $\boldsymbol{A}$ (zeros out the rest). Therefore, $\boldsymbol{I}_{i;p}\boldsymbol{A}'\boldsymbol{A}\boldsymbol{I}_{i;p}$ is a $p \times p$ matrix that its Leading Principal Submatrix of order $i$ (LPS$_i$) [1] is the same as the LPS$_i$ of $\boldsymbol{A}'\boldsymbol{A}$ (and the rest of the elements are zero). Hence, $\sum_{i=1}^{p} \boldsymbol{I}_{i;p}\boldsymbol{A}'\boldsymbol{A}\boldsymbol{I}_{i;p}$ (counting backwards) adds LPS$_p$ of $\boldsymbol{A}'\boldsymbol{A}$ (i.e. $\boldsymbol{A}'\boldsymbol{A}$ itself) with LPS$_{p-1}$ that doubles LPS$_{p-1}$ part of the result and then adds LPS$_{p-2}$ that triples the LPS$_{p-2}$ part of result, the process continues until by the last addition LPS$_1$is added to the result for the $p^{\text{th}}$times. This is exactly the same as evaluating $\boldsymbol{S}_p \circ (\boldsymbol{A}'\boldsymbol{A})$.

2. This is a standard result (Horn & Johnson, 2012), and no proof is needed.

3. Directly compute $\boldsymbol{S}_p\boldsymbol{S}_p^{-1}$:

$$\left(\boldsymbol{S}_p\boldsymbol{S}_p^{-1}\right)_{ij} = \sum_{k=1}^{p}(\boldsymbol{S}_p)_{ik}(\boldsymbol{S}_p^{-1})_{kj} \xrightarrow{\forall |k-j|>1 : (\boldsymbol{S}_p^{-1})_{kj}=0}$$

$$= \begin{cases} (\boldsymbol{S}_p)_{i,j-1}(\boldsymbol{S}_p^{-1})_{j-1,j} + (\boldsymbol{S}_p)_{i,j}(\boldsymbol{S}_p^{-1})_{j,j} + (\boldsymbol{S}_p)_{i,j+1}(\boldsymbol{S}_p^{-1})_{j+1,j} & 2 \leq j \ \& \\ & j \leq p-1 \\ (\boldsymbol{S}_p)_{i,p-1}(\boldsymbol{S}_p^{-1})_{p-1,p} + (\boldsymbol{S}_p)_{i,p}(\boldsymbol{S}_p^{-1})_{p,p} & j = p \\ (\boldsymbol{S}_p)_{i,1}(\boldsymbol{S}_p^{-1})_{1,1} + (\boldsymbol{S}_p)_{i,2}(\boldsymbol{S}_p^{-1})_{2,1} & j = 1 \end{cases}$$

$$= \begin{cases} -(\boldsymbol{S}_p)_{i,j-1} + 2(\boldsymbol{S}_p)_{i,j} - (\boldsymbol{S}_p)_{i,j+1} & 2 \leq j \leq p-1 \\ -(\boldsymbol{S}_p)_{i,p-1} + 2(\boldsymbol{S}_p)_{i,p} & j = p \\ (\boldsymbol{S}_p)_{i,1} - (\boldsymbol{S}_p)_{i,2} & j = 1 \end{cases}$$

$$= \begin{cases} \max(i, j-1) - 2\max(i,j) + \max(i, j+1) & 2 \leq j \leq p-1 \\ -(p - \max(i, p-1) + 1) + 2(p - \max(i,p) + 1) & j = p \\ -\max(i, 1) + \max(i, 2) & j = 1 \end{cases}$$

---

[1]For a $p \times p$ matrix, the leading principal submatrix of order $i$ is an $i \times i$ matrix derived by removing the last $p - i$ rows and columns of the original matrix (Horn & Johnson (2012), P17)

$$= \begin{cases} \max(i, j-1) - 2\max(i,j) + \max(i, j+1) & 2 \le j \le p-1 \\ 1 - p + \max(i, p-1) & j = p \\ \max(i, 2) - \max(i, 1) & j = 1 \end{cases}$$

$$= \begin{cases} \begin{cases} 1 & i = j \\ 0 & i \neq j \end{cases} & 1 < j < p \\ \begin{cases} 1 & i = p \\ 0 & i \neq p \end{cases} & j = p \\ \begin{cases} 1 & i = 1 \\ 0 & i \ge 2 \end{cases} & j = 1 \end{cases} = (\boldsymbol{I}_p)_{ij}.$$

4. Firstly, note that $\boldsymbol{S}_p^{-1}$ is symmetric and nonsingular so all the eigenvalues are real and nonzero. It is also a diagonally dominant matrix (Horn & Johnson (2012), Def 6.1.9) since

$$\forall i \in \{1, \cdots, p\} : C_i := |(\boldsymbol{S}_p^{-1})_{ii}| \ge \sum_{j=1, j \neq i} |(\boldsymbol{S}_p^{-1})_{ij}| =: R_i,$$

where the inequality is strict for the first and the last row and it is equal for the rows in the middle. Moreover, by Gersgorin circle theorem (Horn & Johnson (2012), Thm 6.1.1) for every eigenvalue $l_i$ of $\boldsymbol{S}_p^{-1}$ there exists $i$ such that $l_i \in [C_i - R_i, C_i + R_i]$. Since $\forall i : C_i \ge R_i$ we have all the eigenvalues are non-negative. They are also nonzero, hence, $\boldsymbol{S}_p^{-1}$ is positive definite, which implies $\boldsymbol{S}_p$ is also positive definite.

5. For any matrix $\boldsymbol{A} \in \mathbb{R}^{n \times p}$, $\boldsymbol{A}'\boldsymbol{A}$ is positive semidefinite. Also, $\boldsymbol{S}_p$ is positive definite so by Schur product theorem (Horn & Johnson (2012), Thm 7.5.3(a)), $\boldsymbol{S}_p \circ (\boldsymbol{A}'\boldsymbol{A})$ is positive semidefinite. Moreover, if all diagonal elements of $\boldsymbol{A}'\boldsymbol{A}$ are positive (i.e. $\boldsymbol{A}$ has no zero column) by the extension of Schur product theorem (Horn & Johnson (2012), Thm 7.5.3(b)) it is positive definite. This can also be easily deduced using the Oppenheim inequality (Horn & Johnson (2012), Thm 7.8.16); that is for positive semidefinite matrices $\boldsymbol{S}_p$ and $\boldsymbol{A}'\boldsymbol{A}$: $\det(\boldsymbol{S}_p) \prod_i (\boldsymbol{A}'\boldsymbol{A})_{ii} \le \det(\boldsymbol{S}_p \circ (\boldsymbol{A}'\boldsymbol{A}))$. Since, $\boldsymbol{S}_p$ is positive definite, $\det(\boldsymbol{S}_p) > 0$ (in fact it is 1 for any $p$) and if $\boldsymbol{A}'\boldsymbol{A}$ has no zero diagonal then $\det(\boldsymbol{S}_p \circ (\boldsymbol{A}'\boldsymbol{A})) > 0$ and therefore, $\boldsymbol{S}_p \circ (\boldsymbol{A}'\boldsymbol{A})$ is positive definite.

6. Clearly, the matrix $\boldsymbol{T}_p$ is achieved by setting the off-diagonal elements of $\boldsymbol{S}_p$ to zero. Hence, for any diagonal matrix $\boldsymbol{\mathscr{D}} \in \mathbb{R}^{p \times p}$: $\boldsymbol{S}_p \circ \boldsymbol{\mathscr{D}} = \boldsymbol{T}_p \circ \boldsymbol{\mathscr{D}}$. For the diagonal matrices Hadamard product and matrix product are interchangeable so the latter may also be written as $\boldsymbol{T}_p \boldsymbol{\mathscr{D}}$. The same argument applies for the second identity.

7. This property can easily be proved by induction on $p$ and careful bookkeeping of indices.

$\square$

**Lemma 3** (Simultaneous diagonalization by congruence). *Let $\boldsymbol{M}_1, \boldsymbol{M}_2 \in \mathbb{R}^{p \times p}$, where $\boldsymbol{M}_1$ is positive definite and $\boldsymbol{M}_2$ is positive semidefinite. Also, let $\boldsymbol{\mathscr{D}}, \boldsymbol{\mathscr{E}} \in \mathbb{R}^{r \times r}$ be positive definite diagonal matrices with $r \le p$. Further, assume there is a $\boldsymbol{C} \in \mathbb{R}^{r \times p}$ of rank $r \le p$ such that*

$$\boldsymbol{C}\boldsymbol{M}_1\boldsymbol{C}' = \boldsymbol{\mathscr{D}} \text{ and}$$
$$\boldsymbol{C}\boldsymbol{M}_2\boldsymbol{C}' = \boldsymbol{\mathscr{D}}\boldsymbol{\mathscr{E}}.$$

*Then there exists a nonsingular $\bar{\boldsymbol{C}} \in \mathbb{R}^{p \times p}$ that its first $r$ rows are the matrix $\boldsymbol{C}$ and*

$$\bar{\boldsymbol{C}}\boldsymbol{M}_1\bar{\boldsymbol{C}}' = \bar{\boldsymbol{\mathscr{D}}} \text{ and}$$
$$\bar{\boldsymbol{C}}\boldsymbol{M}_2\bar{\boldsymbol{C}}' = \bar{\boldsymbol{\mathscr{D}}}\bar{\boldsymbol{\mathscr{E}}},$$

*where, $\bar{\boldsymbol{\mathscr{D}}} = \bar{\boldsymbol{\mathscr{D}}} \oplus \boldsymbol{I}_{r-p}$ is a $p \times p$ diagonal matrix and $\bar{\boldsymbol{\mathscr{E}}} = \boldsymbol{\mathscr{E}} \oplus \underline{\boldsymbol{\mathscr{E}}}$ is another $p \times p$ diagonal matrix, in which $\underline{\boldsymbol{\mathscr{E}}} \in \mathbb{R}^{p-r \times p-r}$ is a nonnegative diagonal matrix. Clearly, the rank of $\boldsymbol{M}_2$ is $r$ plus the number of nonzero diagonal elements of $\underline{\boldsymbol{\mathscr{E}}}$.*

*Proof.* The proof is rather straightforward since this lemma is the direct consequence of Theorem 7.6.4 in Horn & Johnson (2012). The theorem basically states that if $\boldsymbol{M}_1, \boldsymbol{M}_2 \in \mathbb{R}^{p \times p}$ is symmetric

and $M_1$ is positive definite then there exists an invertible $S \in \mathbb{R}^{p \times p}$ such that $SM_1S' = I_p$ and $SM_2S'$ is a diagonal matrix with the same inertia as $M_2$. Here, we have $M_2$ that is positive semidefinite and $C \in \mathbb{R}^{r \times p}$ of rank $r \leq p$ such that

$$\left(\mathscr{D}^{\frac{-1}{2}}C\right) M_1 \left(\mathscr{D}^{\frac{-1}{2}}C\right)' = I_r \text{ and}$$

$$\left(\mathscr{D}^{\frac{-1}{2}}C\right) M_2 \left(\mathscr{D}^{\frac{-1}{2}}C\right)' = \mathscr{E}.$$

Therefore, since $S$ is of full rank $p$ and $\mathscr{D}^{\frac{-1}{2}}C$ is of rank $r \leq p$, there exists $p - r$ rows in $S$ that are linearly independent of rows of $\mathscr{D}^{\frac{-1}{2}}C$. Establish $\bar{C} \in \mathbb{R}^{p \times p}$ by adding those $p - r$ rows to $C$. Then $\bar{C}$ has $p$ linearly independent rows so it is nonsingular, and fulfills the lemma's proposition that is

$$\bar{C}M_1\bar{C}' = \bar{\mathscr{D}} \text{ and}$$

$$\bar{C}M_2\bar{C}' = \bar{\mathscr{D}}\bar{\mathscr{E}},$$

where, $\bar{\mathscr{D}} = \bar{\mathscr{D}} \oplus I_{r-p}$ is a $p \times p$ diagonal matrix and $\bar{\mathscr{E}} = \mathscr{E} \oplus \underline{\mathscr{E}}$ is another $p \times p$ diagonal matrix, in which $\underline{\mathscr{E}} \in \mathbb{R}^{p-r \times p-r}$ is a nonnegative diagonal matrix. $\square$

**Lemma 4.** *Let $A$ and $B$ define a critical point of L. Further, let $V \in \mathbb{R}^{n \times p}$ and $W \in \mathbb{R}^{p \times n}$ are such that $\|V\|_F, \|W\|_F = O(\varepsilon)$ for some $\varepsilon > 0$. Then*

$$
\begin{aligned}
L(A + V, B + W) - L(A, B) =& \langle VT_p B\Sigma_{xx}B', V\rangle_F \\
&- 2\langle \Sigma_{yx}W'T_p - A\left(S_p \circ \left(B\Sigma_{xx}W' + W\Sigma_{xx}B'\right)\right), V\rangle_F \\
&+ \langle (S_p \circ (A'A)) W\Sigma_{xx}, W\rangle_F + O(\varepsilon^3).
\end{aligned}
\tag{29}
$$

*Further, for $W = \bar{W} := (S_p \circ (A'A))^{-1} T_p V' \Sigma_{yx}\Sigma_{xx}^{-1}$, the above equation becomes*

$$
\begin{aligned}
L(A + V, B + \bar{W}) - L(A, B) =& \mathrm{Tr}\left(V'VT_pB\Sigma_{xx}B'\right) - \mathrm{Tr}\left(V'\Sigma VT_p (S_p \circ (A'A))^{-1}T_p\right) \\
&+ 2\,\mathrm{Tr}\left(V'A\left(S_p \circ \left(B\Sigma_{xy}VT_p (S_p \circ (A'A))^{-1}\right.\right.\right. \\
&\quad\left.\left.\left. + (S_p \circ (A'A))^{-1}T_pV'\Sigma_{yx}B'\right)\right)\right) + O(\varepsilon^3).
\end{aligned}
\tag{30}
$$

*Finally, in case the critical $A$ is of full rank $p$ and so, $(A, B) = (U_{\mathbb{I}_p}\Pi D, \hat{B}(U_{\mathbb{I}_p}\Pi D))$, for the encoder direction $V$ with $\|V\|_F = O(\varepsilon)$ and $W = \bar{W}$ we have,*

$$
\begin{aligned}
L(A + V, B + W) - L(A, B) =& \mathrm{Tr}\left(V'V\Pi'\Lambda_{\mathbb{I}_p}\Pi T_p D^{-2}\right) - \mathrm{Tr}\left(V'\Sigma VT_pD^{-2}\right) \\
&+ 2\,\mathrm{Tr}\left(V'U_{\mathbb{I}_p}\Pi D\left(S_p \circ \left(D^{-1}\Pi'U_{\mathbb{I}_p}'\Sigma VD^{-2}\right)\right)\right) \\
&+ 2\,\mathrm{Tr}\left(V'U_{\mathbb{I}_p}\Pi D\left(S_p \circ \left(D^{-2}V'\Sigma U_{\mathbb{I}_p}\Pi D^{-1}\right)\right)\right) \\
&+ O(\varepsilon^3).
\end{aligned}
\tag{31}
$$

*Proof.* As described in appendix B.1, the second order Taylor expansion for the loss $L(A, B)$ is then given by eq. (63), i.e.

$$
\begin{aligned}
L(A + V, B + W) - L(A, B) =& d_A L(A, B)V + d_B L(A, B)W + \frac{1}{2}d_A^2 L(A, B)V^2 \\
&+ d_{AB}L(A, B)VW + \frac{1}{2}d_B^2 L(A, B)W^2 + R_{V,W}(A, B).
\end{aligned}
$$

If $\|V\|_F, \|W\|_F = O(\varepsilon)$ then $\|R(V, W)\| = O(\varepsilon^3)$. Moreover, when $A$ and $B$ define a critical point of $L$ we have $d_A L(A, B)V = d_B L(A, B)W = 0$. By setting the derivatives $d_A^2 L(A, B)V^2$, $d_{AB}L(A, B)VW$, $d_B^2 L(A, B)W^2$ that are given by eq. (69), eq. (68), and eq. (66) respectively, the above equation simplifies to

$$
\begin{aligned}
L(\boldsymbol{A}+\boldsymbol{V},\boldsymbol{B}+\boldsymbol{W})-L(\boldsymbol{A},\boldsymbol{B})=&\langle\boldsymbol{V}\left(\boldsymbol{S}_p\circ(\boldsymbol{B}\boldsymbol{\Sigma}_{xx}\boldsymbol{B}')\right),\boldsymbol{V}\rangle_F\\
&-2\langle\boldsymbol{\Sigma}_{yx}\boldsymbol{W}'\boldsymbol{T}_p-\boldsymbol{A}\left(\boldsymbol{S}_p\circ(\boldsymbol{B}\boldsymbol{\Sigma}_{xx}\boldsymbol{W}'+\boldsymbol{W}\boldsymbol{\Sigma}_{xx}\boldsymbol{B}')\right),\boldsymbol{V}\rangle_F\\
&+\langle(\boldsymbol{S}_p\circ(\boldsymbol{A}'\boldsymbol{A}))\boldsymbol{W}\boldsymbol{\Sigma}_{xx},\boldsymbol{W}\rangle_F+O(\varepsilon^3).
\end{aligned}
$$

Now, based on the first item in Corollary 1, $\boldsymbol{B}\boldsymbol{\Sigma}_{xx}\boldsymbol{B}'$ is a $p\times p$ diagonal matrix, so based on eq. (27): $\boldsymbol{S}_p\circ(\boldsymbol{B}\boldsymbol{\Sigma}_{xx}\boldsymbol{B}')=\boldsymbol{T}_p\boldsymbol{B}\boldsymbol{\Sigma}_{xx}\boldsymbol{B}'$. The substitution then yields eq. (29). Finally, in the above equation replace $\boldsymbol{W}$ with $\bar{\boldsymbol{W}}=\left(\boldsymbol{S}_p\circ(\boldsymbol{A}'\boldsymbol{A})\right)^{-1}\boldsymbol{T}_p\boldsymbol{V}'\boldsymbol{\Sigma}_{yx}\boldsymbol{\Sigma}_{xx}^{-1}$. We have

$$
\begin{aligned}
&L(\boldsymbol{A}+\boldsymbol{V},\boldsymbol{B}+\bar{\boldsymbol{W}})-L(\boldsymbol{A},\boldsymbol{B})=\\
&=\langle\boldsymbol{V}\boldsymbol{T}_p\boldsymbol{B}\boldsymbol{\Sigma}_{xx}\boldsymbol{B}',\boldsymbol{V}\rangle_F-2\langle\boldsymbol{\Sigma}_{yx}\boldsymbol{\Sigma}_{xx}^{-1}\boldsymbol{\Sigma}_{xy}\boldsymbol{V}\boldsymbol{T}_p\left(\boldsymbol{S}_p\circ(\boldsymbol{A}'\boldsymbol{A})\right)^{-1}\boldsymbol{T}_p,\boldsymbol{V}\rangle_F\\
&+2\langle\boldsymbol{A}\Big(\boldsymbol{S}_p\circ\Big(\boldsymbol{B}\boldsymbol{\Sigma}_{xx}\boldsymbol{\Sigma}_{xx}^{-1}\boldsymbol{\Sigma}_{xy}\boldsymbol{V}\boldsymbol{T}_p(\boldsymbol{S}_p\circ(\boldsymbol{A}'\boldsymbol{A}))^{-1}+(\boldsymbol{S}_p\circ(\boldsymbol{A}'\boldsymbol{A}))^{-1}\boldsymbol{T}_p\boldsymbol{V}'\boldsymbol{\Sigma}_{yx}\boldsymbol{\Sigma}_{xx}^{-1}\boldsymbol{\Sigma}_{xx}\boldsymbol{B}'\Big)\Big),\boldsymbol{V}\rangle_F\\
&+\langle(\boldsymbol{S}_p\circ(\boldsymbol{A}'\boldsymbol{A}))\left(\boldsymbol{S}_p\circ(\boldsymbol{A}'\boldsymbol{A})\right)^{-1}\boldsymbol{T}_p\boldsymbol{V}'\boldsymbol{\Sigma}_{yx}\boldsymbol{\Sigma}_{xx}^{-1}\boldsymbol{\Sigma}_{xx},\left(\boldsymbol{S}_p\circ(\boldsymbol{A}'\boldsymbol{A})\right)^{-1}\boldsymbol{T}_p\boldsymbol{V}'\boldsymbol{\Sigma}_{yx}\boldsymbol{\Sigma}_{xx}^{-1}\rangle_F+O(\varepsilon^3)\\
&=\operatorname{Tr}\left(\boldsymbol{V}'\boldsymbol{V}\boldsymbol{T}_p\boldsymbol{B}\boldsymbol{\Sigma}_{xx}\boldsymbol{B}'\right)-\operatorname{Tr}\left(\boldsymbol{V}'\boldsymbol{\Sigma}\boldsymbol{V}\boldsymbol{T}_p\left(\boldsymbol{S}_p\circ(\boldsymbol{A}'\boldsymbol{A})\right)^{-1}\boldsymbol{T}_p\right)\\
&+2\operatorname{Tr}\left(\boldsymbol{V}'\boldsymbol{A}\left(\boldsymbol{S}_p\circ\left(\boldsymbol{B}\boldsymbol{\Sigma}_{xy}\boldsymbol{V}\boldsymbol{T}_p\left(\boldsymbol{S}_p\circ(\boldsymbol{A}'\boldsymbol{A})\right)^{-1}+\left(\boldsymbol{S}_p\circ(\boldsymbol{A}'\boldsymbol{A})\right)^{-1}\boldsymbol{T}_p\boldsymbol{V}'\boldsymbol{\Sigma}_{yx}\boldsymbol{B}'\right)\right)\right)+O(\varepsilon^3),
\end{aligned}
$$

which is eq. (30). For the final equation, we have

$$
\begin{aligned}
\boldsymbol{T}_p\boldsymbol{B}\boldsymbol{\Sigma}_{xx}\boldsymbol{B}'=&\boldsymbol{T}_p\boldsymbol{D}^{-1}\boldsymbol{\Pi}'\boldsymbol{U}_{\mathbb{I}_p}'\underbrace{\boldsymbol{\Sigma}_{yx}\boldsymbol{\Sigma}_{xx}^{-1}\boldsymbol{\Sigma}_{xx}\boldsymbol{\Sigma}_{xx}^{-1}\boldsymbol{\Sigma}_{xy}}\boldsymbol{U}_{\mathbb{I}_p}\boldsymbol{\Pi}\boldsymbol{D}^{-1}\\
=&\boldsymbol{T}_p\boldsymbol{D}^{-1}\boldsymbol{\Pi}'\underbrace{\boldsymbol{U}_{\mathbb{I}_p}'\boldsymbol{\Sigma}\boldsymbol{U}_{\mathbb{I}_p}}\boldsymbol{\Pi}\boldsymbol{D}^{-1}=\boldsymbol{T}_p\boldsymbol{D}^{-1}\underbrace{\boldsymbol{\Pi}'\boldsymbol{\Lambda}_{\mathbb{I}_p}\boldsymbol{\Pi}}\boldsymbol{D}^{-1}\\
=&\boldsymbol{\Pi}'\boldsymbol{\Lambda}_{\mathbb{I}_p}\boldsymbol{\Pi}\boldsymbol{T}_p\boldsymbol{D}^{-2},\text{ and}
\end{aligned}
\tag{32}
$$

$$
\begin{aligned}
\boldsymbol{T}_p\left(\boldsymbol{S}_p\circ(\boldsymbol{A}'\boldsymbol{A})\right)^{-1}\boldsymbol{T}_p=&\boldsymbol{T}_p\left(\boldsymbol{S}_p\circ\left(\boldsymbol{D}\underbrace{\boldsymbol{\Pi}'\boldsymbol{U}_{\mathbb{I}_p}'\boldsymbol{U}_{\mathbb{I}_p}\boldsymbol{\Pi}}\boldsymbol{D}\right)\right)^{-1}\boldsymbol{T}_p\\
=&\boldsymbol{T}_p\left(\boldsymbol{S}_p\circ\boldsymbol{D}^2\right)^{-1}\boldsymbol{T}_p=\boldsymbol{T}_p\boldsymbol{T}_p^{-1}\boldsymbol{D}^{-2}\boldsymbol{T}_p=\boldsymbol{T}_p\boldsymbol{D}^{-2}.
\end{aligned}
\tag{33}
$$

Replace the above in eq. (30) and simplify:

$$
\begin{aligned}
L(\boldsymbol{A}+\boldsymbol{V},\boldsymbol{B}+\boldsymbol{W})-L(\boldsymbol{A},\boldsymbol{B})=&\operatorname{Tr}\left(\boldsymbol{V}'\boldsymbol{V}\boldsymbol{T}_p\boldsymbol{B}\boldsymbol{\Sigma}_{xx}\boldsymbol{B}'\right)-\operatorname{Tr}\left(\boldsymbol{V}'\boldsymbol{\Sigma}\boldsymbol{V}\boldsymbol{T}_p\left(\boldsymbol{S}_p\circ(\boldsymbol{A}'\boldsymbol{A})\right)^{-1}\boldsymbol{T}_p\right)\\
&+2\operatorname{Tr}\left(\boldsymbol{V}'\boldsymbol{A}\left(\boldsymbol{S}_p\circ\left(\boldsymbol{B}\boldsymbol{\Sigma}_{xy}\boldsymbol{V}\boldsymbol{T}_p\left(\boldsymbol{S}_p\circ(\boldsymbol{A}'\boldsymbol{A})\right)^{-1}\right.\right.\right.\\
&\left.\left.\left.+\ \left(\boldsymbol{S}_p\circ(\boldsymbol{A}'\boldsymbol{A})\right)^{-1}\boldsymbol{T}_p\boldsymbol{V}'\boldsymbol{\Sigma}_{yx}\boldsymbol{B}'\right)\right)\right)+O(\varepsilon^3)\xRightarrow[\text{eq. (33)}]{\text{eq. (32)}}\\
L(\boldsymbol{A}+\boldsymbol{V},\boldsymbol{B}+\boldsymbol{W})-L(\boldsymbol{A},\boldsymbol{B})=&\operatorname{Tr}\left(\boldsymbol{V}'\boldsymbol{V}\boldsymbol{\Pi}'\boldsymbol{\Lambda}_{\mathbb{I}_p}\boldsymbol{\Pi}\boldsymbol{T}_p\boldsymbol{D}^{-2}\right)-\operatorname{Tr}\left(\boldsymbol{V}'\boldsymbol{\Sigma}\boldsymbol{V}\boldsymbol{T}_p\boldsymbol{D}^{-2}\right)\\
&+2\operatorname{Tr}\left(\boldsymbol{V}'\boldsymbol{A}\left(\boldsymbol{S}_p\circ\left(\boldsymbol{B}\boldsymbol{\Sigma}_{xy}\boldsymbol{V}\boldsymbol{D}^{-2}+\boldsymbol{D}^{-2}\boldsymbol{V}'\boldsymbol{\Sigma}_{yx}\boldsymbol{B}'\right)\right)\right)\\
&+O(\varepsilon^3)\xRightarrow[\boldsymbol{B}=\hat{\boldsymbol{B}}(\boldsymbol{U}_{\mathbb{I}_p}\boldsymbol{\Pi}\boldsymbol{D})]{\boldsymbol{A}=\boldsymbol{U}_{\mathbb{I}_p}\boldsymbol{\Pi}\boldsymbol{D}}\\
L(\boldsymbol{A}+\boldsymbol{V},\boldsymbol{B}+\boldsymbol{W})-L(\boldsymbol{A},\boldsymbol{B})=&\operatorname{Tr}\left(\boldsymbol{V}'\boldsymbol{V}\boldsymbol{\Pi}'\boldsymbol{\Lambda}_{\mathbb{I}_p}\boldsymbol{\Pi}\boldsymbol{T}_p\boldsymbol{D}^{-2}\right)-\operatorname{Tr}\left(\boldsymbol{V}'\boldsymbol{\Sigma}\boldsymbol{V}\boldsymbol{T}_p\boldsymbol{D}^{-2}\right)\\
&+2\operatorname{Tr}\left(\boldsymbol{V}'\boldsymbol{U}_{\mathbb{I}_p}\boldsymbol{\Pi}\boldsymbol{D}\left(\boldsymbol{S}_p\circ\left(\boldsymbol{D}^{-1}\boldsymbol{\Pi}'\boldsymbol{U}_{\mathbb{I}_p}'\boldsymbol{\Sigma}\boldsymbol{V}\boldsymbol{D}^{-2}\right)\right)\right)\\
&+2\operatorname{Tr}\left(\boldsymbol{V}'\boldsymbol{U}_{\mathbb{I}_p}\boldsymbol{\Pi}\boldsymbol{D}\left(\boldsymbol{S}_p\circ\left(\boldsymbol{D}^{-2}\boldsymbol{V}'\boldsymbol{\Sigma}\boldsymbol{U}_{\mathbb{I}_p}\boldsymbol{\Pi}\boldsymbol{D}^{-1}\right)\right)\right)\\
&+O(\varepsilon^3),
\end{aligned}
$$

which finalizes the proof. $\qquad\square$

## A.2 Proof of Proposition 1

For this proof we use the first and second order derivatives for $L(\boldsymbol{A},\boldsymbol{B})$ wrt $\boldsymbol{B}$ derived in Lemma 5. From eq. (66), we have that for a given $\boldsymbol{A}$ the second derivative wrt to $\boldsymbol{B}$ of the cost $L(\boldsymbol{A},\boldsymbol{B})$ at $\boldsymbol{B}$,

and in the direction $\boldsymbol{W}$ is the quadratic form

$$d_{\boldsymbol{B}^2}^2 L(\boldsymbol{A}, \boldsymbol{B}) \boldsymbol{W}^2 = 2\operatorname{Tr}\left(\boldsymbol{W}'\left(\boldsymbol{S}_p \circ \boldsymbol{A}'\boldsymbol{A}\right)\boldsymbol{W}\boldsymbol{\Sigma}_{xx}\right).$$

The matrix $\boldsymbol{\Sigma}_{xx}$ is positive-definite and by Lemma 2, $\boldsymbol{S}_p \circ \boldsymbol{A}'\boldsymbol{A}$ is positive-semidefinite. Hence, $d_{\boldsymbol{B}^2}^2 L(\boldsymbol{A}, \boldsymbol{B}) \boldsymbol{W}^2$ is clearly non-negative for all $\boldsymbol{W} \in \mathbb{R}^{p \times n}$. Therefore, $L(\boldsymbol{A}, \boldsymbol{B})$ is convex in coefficients of $\boldsymbol{B}$ for a fixed matrix $\boldsymbol{A}$. Also the critical points of $L(\boldsymbol{A}, \boldsymbol{B})$ for a fixed $\boldsymbol{A}$ is a matrix $\boldsymbol{B}$ that satisfies $\forall \boldsymbol{W} \in \mathbb{R}^{p \times n} : d_{\boldsymbol{B}} L(\boldsymbol{A}, \boldsymbol{B}) \boldsymbol{W} = 0$ and hence, from eq. (64) we have

$$-2\langle \boldsymbol{T}_p \boldsymbol{A}'\boldsymbol{\Sigma}_{yx} - (\boldsymbol{S}_p \circ (\boldsymbol{A}'\boldsymbol{A}))\boldsymbol{B}\boldsymbol{\Sigma}_{xx}, \boldsymbol{W}\rangle_F = 0.$$

Setting $\boldsymbol{W} = \boldsymbol{T}_p \boldsymbol{A}'\boldsymbol{\Sigma}_{yx} - (\boldsymbol{S}_p \circ (\boldsymbol{A}'\boldsymbol{A}))\boldsymbol{B}\boldsymbol{\Sigma}_{xx}$ we have

$$\boldsymbol{T}_p \boldsymbol{A}'\boldsymbol{\Sigma}_{yx} - (\boldsymbol{S}_p \circ (\boldsymbol{A}'\boldsymbol{A}))\boldsymbol{B}\boldsymbol{\Sigma}_{xx} = 0.$$

For a fixed $\boldsymbol{A}$, the cost $L(\boldsymbol{A}, \boldsymbol{B})$ is convex in $\boldsymbol{B}$, so any matrix $\boldsymbol{B}$ that satisfies the above equation corresponds to a minimum of $L(\boldsymbol{A}, \boldsymbol{B})$. Further, if $\boldsymbol{A}$ has no zero column then by Lemma 2, $\boldsymbol{S}_p \circ \boldsymbol{A}'\boldsymbol{A}$ is positive definite. Hence, $\forall \boldsymbol{W} \in \mathbb{R}^{p \times n} : d_{\boldsymbol{B}^2}^2 L(\boldsymbol{A}, \boldsymbol{B}) \boldsymbol{W}^2 = 2\operatorname{Tr}\left(\boldsymbol{W}'\left(\boldsymbol{S}_p \circ \boldsymbol{A}'\boldsymbol{A}\right)\boldsymbol{W}\boldsymbol{\Sigma}_{xx}\right)$ is positive. Therefore, the cost $L(\boldsymbol{A}, \boldsymbol{B})$ becomes strictly convex and the unique global minimum is achieved at $\boldsymbol{B} = \hat{\boldsymbol{B}}(\boldsymbol{A})$ as defined in eq. (6).

### A.3 PROOF OF PROPOSITION 2

For this proof we use the first and second order derivatives for $L(\boldsymbol{A}, \boldsymbol{B})$ wrt $\boldsymbol{A}$ derived in Lemma 6. For a fixed $\boldsymbol{B}$, based on eq. (69) the second derivative wrt to $\boldsymbol{A}$ of $L(\boldsymbol{A}, \boldsymbol{B})$ at $\boldsymbol{A}$, and in the direction $\boldsymbol{V}$ is the quadratic form

$$d_{\boldsymbol{A}^2}^2 L(\boldsymbol{A}, \boldsymbol{B}) \boldsymbol{V}^2 = 2\langle \boldsymbol{V}\left(\boldsymbol{S}_p \circ (\boldsymbol{B}\boldsymbol{\Sigma}_{xx}\boldsymbol{B}')\right), \boldsymbol{V}\rangle_F = 2\operatorname{Tr}\left(\boldsymbol{V}\left(\boldsymbol{S}_p \circ (\boldsymbol{B}\boldsymbol{\Sigma}_{xx}\boldsymbol{B}')\right)\boldsymbol{V}'\right).$$

The matrix $\boldsymbol{\Sigma}_{xx}$ is positive-definite and by Lemma 2, $\boldsymbol{S}_p \circ (\boldsymbol{B}\boldsymbol{\Sigma}_{xx}\boldsymbol{B}')$ is positive-semidefinite. Hence, $d_{\boldsymbol{A}^2}^2 L_1(\boldsymbol{A}, \boldsymbol{B}) \boldsymbol{V}^2$ is non-negative for all $\boldsymbol{V} \in \mathbb{R}^{n \times p}$. Therefore, $L(\boldsymbol{A}, \boldsymbol{B})$ is convex in coefficients of $\boldsymbol{A}$ for a fixed matrix $\boldsymbol{B}$. Based on eq. (67) the critical point of $L(\boldsymbol{A}, \boldsymbol{B})$ for a fixed $\boldsymbol{B}$ is a matrix $\boldsymbol{A}$ that satisfies for all $\boldsymbol{V} \in \mathbb{R}^{n \times p}$

$$d_{\boldsymbol{A}} L(\boldsymbol{A}, \boldsymbol{B}) \boldsymbol{V} = \langle -2\left(\boldsymbol{\Sigma}_{yx}\boldsymbol{B}'\boldsymbol{T}_p - \boldsymbol{A}\left(\boldsymbol{S}_p \circ (\boldsymbol{B}\boldsymbol{\Sigma}_{xx}\boldsymbol{B}')\right)\right), \boldsymbol{V}\rangle_F = 0 \implies$$
$$\boldsymbol{\Sigma}_{yx}\boldsymbol{B}'\boldsymbol{T}_p = \boldsymbol{A}\left(\boldsymbol{S}_p \circ (\boldsymbol{B}\boldsymbol{\Sigma}_{xx}\boldsymbol{B}')\right),$$

which is eq. (7).

### A.4 PROOF OF THEOREM 1

Before we start, a reminder on notation and some useful identities that are used throughout the proof. The matrix $\boldsymbol{\Sigma} := \boldsymbol{\Sigma}_{yx}\boldsymbol{\Sigma}_{xx}^{-1}\boldsymbol{\Sigma}_{xy}$ has an eigenvalue decomposition $\boldsymbol{\Sigma} = \boldsymbol{U}\boldsymbol{\Lambda}\boldsymbol{U}'$, where the $i^{th}$ column of $\boldsymbol{U}$, denoted as $\boldsymbol{u}_i$, is an eigenvector of $\boldsymbol{\Sigma}$ corresponding to the $i^{\text{th}}$ largest eigenvalue of $\boldsymbol{\Sigma}$, denoted as $\lambda_i$. Also, $\boldsymbol{\Lambda} = \operatorname{diag}(\lambda_1, \cdots, \lambda_n)$ is the diagonal vector of ordered eigenvalues of $\boldsymbol{\Sigma}$, with $\lambda_1 > \lambda_2 > \cdots > \lambda_n > 0$. We use the following notation to organize a subset of eigenvectors of $\boldsymbol{\Sigma}$ into a rectangular matrix. Let for any $r \leq p$, $\mathbb{I}_r = \{i_1, \cdots, i_r\}(1 \leq i_1 < \cdots < i_r < n)$ be any *ordered* $r-$index set. Define $\boldsymbol{U}_{\mathbb{I}_r} \in \mathbb{R}^{n \times p}$ as $\boldsymbol{U}_{\mathbb{I}_r} = [\boldsymbol{u}_{i_1}, \cdots, \boldsymbol{u}_{i_r}]$. That is the columns of $\boldsymbol{U}_{\mathbb{I}_r}$ are the ordered orthonormal eigenvectors of $\boldsymbol{\Sigma}$ associated with eigenvalues $\lambda_{i_1} < \cdots < \lambda_{i_r}$. The following identities are then easy to verify:

$$\boldsymbol{U}_{\mathbb{I}_r}'\boldsymbol{U}_{\mathbb{I}_r} = \boldsymbol{I}_r,$$
$$\boldsymbol{\Sigma}\boldsymbol{U}_{\mathbb{I}_r} = \boldsymbol{U}_{\mathbb{I}_r}\boldsymbol{\Lambda}_{\mathbb{I}_r}, \tag{34}$$
$$\boldsymbol{U}_{\mathbb{I}_r}'\boldsymbol{\Sigma}\boldsymbol{U}_{\mathbb{I}_r} = \boldsymbol{\Lambda}_{\mathbb{I}_r}. \tag{35}$$

**The sufficient condition:**

Let $\boldsymbol{A} \in \mathbb{R}^{n \times p}$ of rank $r \leq p$ and no zero column be given by eq. (8), $\boldsymbol{B} \in \mathbb{R}^{p \times n}$ given by eq. (9), and the accompanying conditions are met. Notice that $\boldsymbol{U}_{\mathbb{I}_r}'\boldsymbol{U}_{\mathbb{I}_r} = \boldsymbol{I}_r$ implies that $\boldsymbol{D}\boldsymbol{C}'\boldsymbol{C}\boldsymbol{D} = \boldsymbol{D}\boldsymbol{C}'\boldsymbol{U}_{\mathbb{I}_r}'\boldsymbol{U}_{\mathbb{I}_r}\boldsymbol{C}\boldsymbol{D} = \boldsymbol{A}'\boldsymbol{A}$, so

$$\boldsymbol{B} = \boldsymbol{D}^{-1}\boldsymbol{\Pi}_C\boldsymbol{U}_{\mathbb{I}_r}'\boldsymbol{\Sigma}_{yx}\boldsymbol{\Sigma}_{xx}^{-1} \xrightarrow[\boldsymbol{D}^{-1}\boldsymbol{D}=\boldsymbol{I}_p]{\boldsymbol{\Pi}_C := (\boldsymbol{S}_p \circ (\boldsymbol{C}'\boldsymbol{C}))^{-1}\boldsymbol{T}_p\boldsymbol{C}'}$$

$$B = D^{-1} \left( S_p \circ (C'C) \right)^{-1} D^{-1} D T_p C' U'_{\mathbb{I}_r} \Sigma_{yx} \Sigma_{xx}^{-1} \xrightarrow[D T_p = T_p D]{\text{Lemma 2-2}}$$

$$B = \left( S_p \circ (\underbrace{DC'CD}) \right)^{-1} T_p \underbrace{DC'U'_{\mathbb{I}_r}} \Sigma_{yx} \Sigma_{xx}^{-1} \xrightarrow[DC'CD=A'A]{A'=D'C'U'_{\mathbb{I}_r}}$$

$$B = \left( S_p \circ (A'A) \right)^{-1} T_p A' \Sigma_{yx} \Sigma_{xx}^{-1} = \hat{B}(A),$$

which is eq. (6). Therefore, based on Proposition 1, for the given $A$, the matrix $B$ defines a critical point of $L(A, B)$. For the gradient wrt to $A$, first note that with $B$ given by eq. (9) we have

$$B\Sigma_{xx}B' = D^{-1}\Pi_C U'_{\mathbb{I}_r} \Sigma_{yx} \Sigma_{xx}^{-1} \Sigma_{xx} \Sigma_{xx}^{-1} \Sigma_{xy} U_{\mathbb{I}_r} \Pi'_C D^{-1}$$

$$= D^{-1}\Pi_C \underbrace{U'_{\mathbb{I}_r} \Sigma_{yx} \Sigma_{xx}^{-1} \Sigma_{xy} U_{\mathbb{I}_r}} \Pi'_C D^{-1} \xrightarrow{\text{eq. (35)}}$$

$$B\Sigma_{xx}B' = D^{-1}\Pi_C \Lambda_{\mathbb{I}_r} \Pi'_C D^{-1}. \tag{36}$$

The matrix $\Pi_C$ is a rectangular permutation matrix so $\Pi_C \Lambda_{\mathbb{I}_r} \Pi'_C$ is diagonal so as $D^{-1}\Pi_C \Lambda_{\mathbb{I}_r} \Pi'_C D^{-1}$. Therefore, $B\Sigma_{xx}B'$ is diagonal and by eq. (27) in Lemma 2-6 we have

$$S_p \circ (B\Sigma_{xx}B') = T_p B\Sigma_{xx}B' = B\Sigma_{xx}B'T_p$$

$$= D^{-1}\Pi_C \Lambda_{\mathbb{I}_r} \Pi'_C D^{-1} T_p \xrightarrow{A\times}$$

$$A \left( S_p \circ (B\Sigma_{xx}B') \right) = AD^{-1}\Pi_C \Lambda_{\mathbb{I}_r} \Pi'_C D^{-1} T_p \xrightarrow{A=U_{\mathbb{I}_r}CD}$$

$$A \left( S_p \circ (B\Sigma_{xx}B') \right) = U_{\mathbb{I}_r}CDD^{-1}\Pi_C \Lambda_{\mathbb{I}_r} \Pi'_C D^{-1} T_p \xrightarrow{A=U_{\mathbb{I}_r}CD}$$

$$= U_{\mathbb{I}_r} \underbrace{C\Pi_C} \Lambda_{\mathbb{I}_r} \Pi'_C D^{-1} T_p \xrightarrow{C\Pi_C=I_r}$$

$$A \left( S_p \circ (B\Sigma_{xx}B') \right) = \underbrace{U_{\mathbb{I}_r} \Lambda_{\mathbb{I}_r}} \Pi'_C D^{-1} T_p \xrightarrow{\text{eq. (34)}}$$

$$= \Sigma U_{\mathbb{I}_r} \Pi'_C D^{-1} T_p$$

$$= \Sigma_{yx} \Sigma_{xx}^{-1} \Sigma_{xy} U_{\mathbb{I}_r} \Pi'_C D^{-1} T_p$$

$$= \Sigma_{yx} \underbrace{\left( D^{-1}\Pi_C U'_{\mathbb{I}_r} \Sigma_{yx} \Sigma_{xx}^{-1} \right)'} T_p$$

$$= \Sigma_{yx} B' T_p,$$

which is eq. (7). Therefore, based on Proposition Proposition 2, for the given $B$, the matrix $A$ define a critical point of $L(A, B)$. Hence, $A$ and $B$ together define a critical point of $L(A, B)$.

**The necessary condition:**

Based on Proposition 1 and Proposition 2, for $A$ (with no zero column) and $B$, to define a critical point of $L(A, B)$, $B$ has to be $\hat{B}(A)$ given by eq. (6), and $A$ has to satisfy eq. (7). That is

$$A \left( S_p \circ \left( \hat{B}\Sigma_{xx}\hat{B}' \right) \right) = \Sigma_{yx} \hat{B}' T_p \xrightarrow{\hat{B}(A) \text{ on RHS}}$$

$$A \left( S_p \circ \left( \hat{B}\Sigma_{xx}\hat{B}' \right) \right) = \Sigma_{xy} \Sigma_{xx}^{-1} \Sigma_{yx} A T_p (S_p \circ (A'A))^{-1} T_p \xrightarrow[\Sigma=\Sigma_{xy}\Sigma_{xx}^{-1}\Sigma_{yx}]{\times A'}$$

$$A \left( S_p \circ \left( \hat{B}\Sigma_{xx}\hat{B}' \right) \right) A' = \Sigma A T_p (S_p \circ (A'A))^{-1} T_p A' \xrightarrow[\times U, U'\times]{\Sigma=U\Lambda U''}$$

$$U'A \left( S_p \circ \left( \hat{B}\Sigma_{xx}\hat{B}' \right) \right) A'U = U'U\Lambda U'A T_p (S_p \circ (A'A))^{-1} T_p A'U \xrightarrow{U'U=I_n}$$

$$U'A \left( S_p \circ \left( \hat{B}\Sigma_{xx}\hat{B}' \right) \right) A'U = \Lambda\Delta, \tag{37}$$

where, $\Delta := U'A T_p (S_p \circ (A'A))^{-1} T_p A'U$ is symmetric and positive semidefinite. The LHS of the above equation is symmetric so the RHS is symmetric too, so $\Lambda\Delta = (\Lambda\Delta)' = \Delta'\Lambda' = \Delta\Lambda$. Therefore, $\Delta$ commutes with the diagonal matrix of eigenvalues $\Lambda$. Since, eigenvalues are assumed to be distinct, $\Delta$ has to be diagonal as well. By Lemma 2 $T_p(S_p \circ (A'A))^{-1} T_p$ is positive definite and $U$ is an orthogonal matrix. Therefore, $r = \text{rank}(A) = \text{rank}(\Delta) = \text{rank}(U'\Delta U)$, which implies that the diagonal matrix $\Delta$, has $r$ nonzero and *positive* diagonal entries. There exists an

$r-$index set $\mathbb{I}_r$ corresponding to the nonzero diagonal elements of $\mathbf{\Delta}$. Forming a diagonal matrix $\mathbf{\Delta}_{\mathbb{I}_r} \in \mathbb{R}^{r \times r}$ by filling its diagonal entries (in order) by the nonzero diagonal elements of $\mathbf{\Delta}$ we have

$$\boldsymbol{U}\boldsymbol{\Delta}\boldsymbol{U}' = \boldsymbol{U}_{\mathbb{I}_r}\boldsymbol{\Delta}_{\mathbb{I}_r}\boldsymbol{U}'_{\mathbb{I}_r} \xLeftrightarrow{\text{Def of } \boldsymbol{\Delta}}$$

$$\boldsymbol{U}\boldsymbol{U}'\boldsymbol{A}\boldsymbol{T}_p(\boldsymbol{S}_p \circ (\boldsymbol{A}'\boldsymbol{A}))^{-1}\boldsymbol{T}_p\boldsymbol{A}'\boldsymbol{U}\boldsymbol{U}' = \boldsymbol{U}_{\mathbb{I}_r}\boldsymbol{\Delta}_{\mathbb{I}_r}\boldsymbol{U}'_{\mathbb{I}_r} \xLeftrightarrow{\boldsymbol{U}\boldsymbol{U}'=\boldsymbol{I}_n}$$

$$\boldsymbol{A}\boldsymbol{T}_p(\boldsymbol{S}_p \circ (\boldsymbol{A}'\boldsymbol{A}))^{-1}\boldsymbol{T}_p\boldsymbol{A}' = \boldsymbol{U}_{\mathbb{I}_r}\boldsymbol{\Delta}_{\mathbb{I}_r}\boldsymbol{U}'_{\mathbb{I}_r}, \tag{38}$$

which indicates that the matrix $\boldsymbol{A}$ has the same column space as $\boldsymbol{U}_{\mathbb{I}_r}$. Therefore, there exists a full rank matrix $\tilde{\boldsymbol{C}} \in \mathbb{R}^{r \times p}$ such that $\boldsymbol{A} = \boldsymbol{U}_{\mathbb{I}_r}\tilde{\boldsymbol{C}}$. Since $\boldsymbol{A}$ has no zero column, $\tilde{\boldsymbol{C}}$ has no zero column. Further, by normalizing the columns of $\tilde{\boldsymbol{C}}$ we can write $\boldsymbol{A} = \boldsymbol{U}_{\mathbb{I}_r}\boldsymbol{C}\boldsymbol{D}$, where $\boldsymbol{D} \in \mathbb{R}^{p \times p}$ is diagonal that contains the norms of columns of $\tilde{\boldsymbol{C}}$. Therefore, $\boldsymbol{A}$ is exactly in the form given by eq. (8). The matrix $\boldsymbol{C}$ has to satisfy eq. (38) that is

$$\boldsymbol{A}\boldsymbol{T}_p(\boldsymbol{S}_p \circ (\boldsymbol{A}'\boldsymbol{A}))^{-1}\boldsymbol{T}_p\boldsymbol{A}' = \boldsymbol{U}_{\mathbb{I}_r}\boldsymbol{\Delta}_{\mathbb{I}_r}\boldsymbol{U}'_{\mathbb{I}_r} \xLeftrightarrow{\boldsymbol{A}=\boldsymbol{U}_{\mathbb{I}_r}\boldsymbol{C}}$$

$$\boldsymbol{U}_{\mathbb{I}_r}\boldsymbol{C}\boldsymbol{D}\boldsymbol{T}_p(\boldsymbol{S}_p \circ (\boldsymbol{A}'\boldsymbol{A}))^{-1}\boldsymbol{T}_p\boldsymbol{D}\boldsymbol{C}'\boldsymbol{U}'_{\mathbb{I}_r} = \boldsymbol{U}_{\mathbb{I}_r}\boldsymbol{\Delta}_{\mathbb{I}_r}\boldsymbol{U}'_{\mathbb{I}_r} \xLeftrightarrow[\boldsymbol{A}'\boldsymbol{A}=\boldsymbol{D}\boldsymbol{C}'\boldsymbol{C}\boldsymbol{D}]{\times \boldsymbol{U}_{\mathbb{I}_r}, \boldsymbol{U}_{\mathbb{I}_r} \times}$$

$$\boldsymbol{C}\boldsymbol{D}\boldsymbol{T}_p(\boldsymbol{S}_p \circ (\boldsymbol{D}\boldsymbol{C}'\boldsymbol{C}\boldsymbol{D}))^{-1}\boldsymbol{T}_p\boldsymbol{C}'\boldsymbol{D} = \boldsymbol{\Delta}_{\mathbb{I}_r} \xLeftrightarrow{\text{Lemma 2-2}}$$

$$\boldsymbol{C}\boldsymbol{T}_p\boldsymbol{D}\boldsymbol{D}^{-1}(\boldsymbol{S}_p \circ (\boldsymbol{C}'\boldsymbol{C}))^{-1}\boldsymbol{D}^{-1}\boldsymbol{D}\boldsymbol{T}_p\boldsymbol{C}' = \boldsymbol{\Delta}_{\mathbb{I}_r} \implies$$

$$\boldsymbol{C}\boldsymbol{T}_p(\boldsymbol{S}_p \circ (\boldsymbol{C}'\boldsymbol{C}))^{-1}\boldsymbol{T}_p\boldsymbol{C}' = \boldsymbol{\Delta}_{\mathbb{I}_r}. \tag{39}$$

Now that the structure of $\boldsymbol{A}$ has been identified, evaluate $\hat{\boldsymbol{B}}(\boldsymbol{A})$ of eq. (6) by setting $\boldsymbol{A} = \boldsymbol{U}_{\mathbb{I}_r}\boldsymbol{C}\boldsymbol{D}$, that is

$$\boldsymbol{B} = \hat{\boldsymbol{B}}(\boldsymbol{A}) = (\boldsymbol{S}_p \circ (\boldsymbol{A}'\boldsymbol{A}))^{-1}\boldsymbol{T}_p\boldsymbol{A}'\boldsymbol{\Sigma}_{yx}\boldsymbol{\Sigma}_{xx}^{-1}$$

$$= (\boldsymbol{S}_p \circ (\boldsymbol{D}\boldsymbol{C}'\boldsymbol{C}\boldsymbol{D}))^{-1}\boldsymbol{T}_p\boldsymbol{D}\boldsymbol{C}'\boldsymbol{U}'_{\mathbb{I}_r}\boldsymbol{\Sigma}_{yx}\boldsymbol{\Sigma}_{xx}^{-1} \xLeftrightarrow{\text{Lemma 2-2}}$$

$$\boldsymbol{B} = \boldsymbol{D}^{-1}(\boldsymbol{S}_p \circ (\boldsymbol{C}'\boldsymbol{C}))^{-1}\boldsymbol{T}_p\boldsymbol{C}'\boldsymbol{U}'_{\mathbb{I}_r}\boldsymbol{\Sigma}_{yx}\boldsymbol{\Sigma}_{xx}^{-1},$$

which by defining $\boldsymbol{\Pi}_C := (\boldsymbol{S}_p \circ (\boldsymbol{C}'\boldsymbol{C}))^{-1}\boldsymbol{T}_p\boldsymbol{C}'$ gives eq. (34) for $\boldsymbol{B}$ as claimed. While $\boldsymbol{C}$ has to satisfy eq. (39), $\boldsymbol{A}$ and $\boldsymbol{B}$ in the given form have to satisfy eq. (37) that provides another condition for $\boldsymbol{C}$ as follows. First, note that

$$\boldsymbol{S}_p \circ \left(\hat{\boldsymbol{B}}\boldsymbol{\Sigma}_{xx}\hat{\boldsymbol{B}}'\right) = \boldsymbol{S}_p \circ \left(\boldsymbol{D}^{-1}(\boldsymbol{S}_p \circ (\boldsymbol{C}'\boldsymbol{C}))^{-1}\boldsymbol{T}_p\boldsymbol{C}'\boldsymbol{U}'_{\mathbb{I}_r}\boldsymbol{\Sigma}\boldsymbol{U}_{\mathbb{I}_r}\boldsymbol{C}\boldsymbol{T}_p(\boldsymbol{S}_p \circ (\boldsymbol{C}'\boldsymbol{C}))^{-1}\boldsymbol{D}^{-1}\right)$$

$$= \boldsymbol{S}_p \circ \left(\boldsymbol{D}^{-1}(\boldsymbol{S}_p \circ (\boldsymbol{C}'\boldsymbol{C}))^{-1}\boldsymbol{T}_p\boldsymbol{C}'\boldsymbol{\Lambda}_{\mathbb{I}_r}\boldsymbol{C}\boldsymbol{T}_p(\boldsymbol{S}_p \circ (\boldsymbol{C}'\boldsymbol{C}))^{-1}\boldsymbol{D}^{-1}\right) \xLeftrightarrow{\text{Lemma 2-2}}$$

$$= \boldsymbol{D}^{-1}\left(\boldsymbol{S}_p \circ ((\boldsymbol{S}_p \circ (\boldsymbol{C}'\boldsymbol{C}))^{-1}\boldsymbol{T}_p\boldsymbol{C}'\boldsymbol{\Lambda}_{\mathbb{I}_r}\boldsymbol{C}\boldsymbol{T}_p(\boldsymbol{S}_p \circ (\boldsymbol{C}'\boldsymbol{C}))^{-1})\right)\boldsymbol{D}^{-1}$$

Now, replace $\boldsymbol{A}$ and $\boldsymbol{B}$ in eq. (37) by their respective identities that we just derived. Performing the same process for eq. (37) we have

$$\boldsymbol{U}'\boldsymbol{A}\left(\boldsymbol{S}_p \circ \left(\hat{\boldsymbol{B}}\boldsymbol{\Sigma}_{xx}\hat{\boldsymbol{B}}'\right)\right)\boldsymbol{A}'\boldsymbol{U} = \boldsymbol{\Lambda}\boldsymbol{\Delta} \xLeftrightarrow[\times \boldsymbol{U}', \boldsymbol{U} \times]{\boldsymbol{A}=\boldsymbol{U}_{\mathbb{I}_r}\boldsymbol{C}\boldsymbol{D}}$$

$$\boldsymbol{U}_{\mathbb{I}_r}\boldsymbol{C}\left(\boldsymbol{S}_p \circ ((\boldsymbol{S}_p \circ (\boldsymbol{C}'\boldsymbol{C}))^{-1}\boldsymbol{T}_p\boldsymbol{C}'\boldsymbol{\Lambda}_{\mathbb{I}_r}\boldsymbol{C}\boldsymbol{T}_p(\boldsymbol{S}_p \circ (\boldsymbol{C}'\boldsymbol{C}))^{-1})\right)\boldsymbol{C}'\boldsymbol{U}'_{\mathbb{I}_r} = \boldsymbol{U}\boldsymbol{\Lambda}\boldsymbol{\Delta}\boldsymbol{U}' \xLeftrightarrow[\boldsymbol{U}'_{\mathbb{I}_r} \times]{\times \boldsymbol{U}_{\mathbb{I}_r}}$$

$$\boldsymbol{C}\left(\boldsymbol{S}_p \circ ((\boldsymbol{S}_p \circ (\boldsymbol{C}'\boldsymbol{C}))^{-1}\boldsymbol{T}_p\boldsymbol{C}'\boldsymbol{\Lambda}_{\mathbb{I}_r}\boldsymbol{C}\boldsymbol{T}_p(\boldsymbol{S}_p \circ (\boldsymbol{C}'\boldsymbol{C}))^{-1})\right)\boldsymbol{C}' = \boldsymbol{U}'_{\mathbb{I}_r}\boldsymbol{U}\boldsymbol{\Lambda}\boldsymbol{\Delta}\boldsymbol{U}'\boldsymbol{U}_{\mathbb{I}_r} \implies$$

$$\boldsymbol{C}\left(\boldsymbol{S}_p \circ ((\boldsymbol{S}_p \circ (\boldsymbol{C}'\boldsymbol{C}))^{-1}\boldsymbol{T}_p\boldsymbol{C}'\boldsymbol{\Lambda}_{\mathbb{I}_r}\boldsymbol{C}\boldsymbol{T}_p(\boldsymbol{S}_p \circ (\boldsymbol{C}'\boldsymbol{C}))^{-1})\right)\boldsymbol{C}' = \boldsymbol{\Lambda}_{\mathbb{I}_r}\boldsymbol{\Delta}_{\mathbb{I}_r}. \tag{40}$$

Now we have to find $\boldsymbol{C}$ such that it satisfies eq. (39) and eq. (40). To make the process easier to follow, lets have them in one place. The matrix $\boldsymbol{C} \in \mathbb{R}^{r \times p}$ have to satisfy

$$\boldsymbol{C}\boldsymbol{T}_p\left(\boldsymbol{S}_p \circ (\boldsymbol{C}'\boldsymbol{C})\right)^{-1}\boldsymbol{T}_p\boldsymbol{C}' = \boldsymbol{\Delta}_{\mathbb{I}_r} \text{ and} \tag{41}$$

$$\boldsymbol{C}\left(\boldsymbol{S}_p \circ ((\boldsymbol{S}_p \circ (\boldsymbol{C}'\boldsymbol{C}))^{-1}\boldsymbol{T}_p\boldsymbol{C}'\boldsymbol{\Lambda}_{\mathbb{I}_r}\boldsymbol{C}\boldsymbol{T}_p(\boldsymbol{S}_p \circ (\boldsymbol{C}'\boldsymbol{C}))^{-1})\right)\boldsymbol{C}' = \boldsymbol{\Lambda}_{\mathbb{I}_r}\boldsymbol{\Delta}_{\mathbb{I}_r}. \tag{42}$$

Since $\boldsymbol{C}$ is a rectangular matrix, solving above equations for $\boldsymbol{C}$ in this form seems intractable. We use a trick to temporarily extend $\boldsymbol{C}$ into an invertible square matrix as follows.

- Temporarily, let $M_1 = T_p\left(S_p \circ (C'C)\right)^{-1} T_p$, and $M_2 = S_p \circ \left((S_p \circ (C'C))^{-1} T_p C' \Lambda_{\mathbb{I}_r} C T_p (S_p \circ (C'C))^{-1}\right)$. Then $M_1$ is positive definite and $M_2$ is positive semidefinite, so they are simultaneously diagonalizable by congruence that is based on Lemma 3 and eq. (41) and eq. (42), there exists a nonsingular $\bar{C} \in \mathbb{R}^{p \times p}$ such that $C$ consists of the first $r$ rows of $\bar{C}$ and

$$\bar{C} T_p \left(S_p \circ (C'C)\right)^{-1} T_p \bar{C}' = \bar{\Delta}_{\mathbb{I}_r}, \qquad (43)$$

$$\bar{C} \left(S_p \circ \left((S_p \circ (C'C))^{-1} T_p C' \Lambda_{\mathbb{I}_r} C T_p (S_p \circ (C'C))^{-1}\right)\right) \bar{C}' = \bar{\Lambda}_{\mathbb{I}_r} \bar{\Delta}_{\mathbb{I}_r}, \qquad (44)$$

where, $\bar{\Delta}_{\mathbb{I}_r} = \Delta_{\mathbb{I}_r} \oplus I_{r-p}$ is a $p \times p$ diagonal matrix and $\bar{\Lambda}_{\mathbb{I}_r} = \Lambda_{\mathbb{I}_r} \oplus \underline{\Lambda}$ is another $p \times p$ diagonal matrix, in which $\underline{\Lambda} \in \mathbb{R}^{r-p \times r-p}$ is a nonnegative diagonal matrix.

- Substitute $\bar{\Delta}_{\mathbb{I}_r}$ from eq. (43) in eq. (44), then left multiply by $\bar{C}'^{-1}$, and right multiply by $\bar{C}' I_{r;p}$:

$$\bar{C} \left(S_p \circ \left((S_p \circ (C'C))^{-1} T_p C' \Lambda_{\mathbb{I}_r} C T_p (S_p \circ (C'C))^{-1}\right)\right) \bar{C}' =$$

$$\bar{\Lambda}_{\mathbb{I}_r} \bar{C} T_p \left(S_p \circ (C'C)\right)^{-1} T_p \bar{C}' \xrightarrow[\times \bar{C}'^{-1}]{\bar{C}' I_{r;p} \times}$$

$$\bar{C}' I_{r;p} \bar{C} \left(S_p \circ \left((S_p \circ (C'C))^{-1} T_p C' \Lambda_{\mathbb{I}_r} C T_p (S_p \circ (C'C))^{-1}\right)\right) =$$

$$\bar{C}' I_{r;p} \bar{\Lambda}_{\mathbb{I}_r} \bar{C} T_p \left(S_p \circ (C'C)\right)^{-1} T_p.$$

- Now we can revert back everything to $C$ again. Since $C$ consists of the first $r$ rows of $\bar{C}$ we have $\bar{C}' I_{r;p} \bar{C} = C'C$, and $\bar{C}' I_{r;p} \bar{\Lambda}_{\mathbb{I}_r} \bar{C} = C' \Lambda_{\mathbb{I}_r} C$, which turns the above equation into

$$C'C \left(S_p \circ \left(I_p (S_p \circ (C'C))^{-1} T_p C' \Lambda_{\mathbb{I}_r} C T_p (S_p \circ (C'C))^{-1} I_p\right)\right) =$$

$$I_p C' \Lambda_{\mathbb{I}_r} C T_p \left(S_p \circ (C'C)\right)^{-1} T_p.$$

- In the above equation, replace $I_p$ by $T_p^{-1} T_p$ in LHS and by $T_p^{-1} \left(S_p \circ (C'C)\right) T_p^{-1} T_p (S_p \circ (C'C))^{-1} T_p$ in the RHS. Use $\Pi_C := (S_p \circ (C'C))^{-1} T_p C'$ to shrink it into :

$$C'C \left(S_p \circ \left(T_p^{-1} T_p \Pi_C \Lambda_{\mathbb{I}_r} \Pi_C' T_p T_p^{-1}\right)\right) = T_p^{-1} \left(S_p \circ (C'C)\right) T_p^{-1} T_p \Pi_C \Lambda_{\mathbb{I}_r} \Pi_C' T_p.$$

- By the second property of Lemma 2 we can collect diagonal matrices $T_p^{-1}$'s around $S_p$ to arrive at

$$(C'C) \left(\hat{S}_p \circ (T_p \Pi_C \Lambda_{\mathbb{I}_r} \Pi_C' T_p)\right) = \left(\hat{S}_p \circ (C'C)\right) (T_p \Pi_C \Lambda_{\mathbb{I}_r} \Pi_C' T_p),$$

where, $\hat{S}_p := T_p^{-1} S_p T_p^{-1}$.

- Define $p \times p$ matrices $\mathscr{E}_r := C'C$ and $\mathscr{D}_r := T_p \Pi_C \Lambda_{\mathbb{I}_r} \Pi_C' T_p$. Substitute in the above to arrive at:

$$\mathscr{E}_r \left(\hat{S}_p \circ \mathscr{D}_r\right) = \left(\hat{S}_p \circ \mathscr{E}_r\right) \mathscr{D}_r.$$

Both $\mathscr{D}_r$ and $\mathscr{E}_r$ in the above identity are positive semidefinite. Moreover, since by assumption $C$ has no zero columns, $\mathscr{E}_r$ has no zero diagonal element. Then the 7th property of Lemma 2 implies the following two conclusions:

  1. The matrix $\mathscr{D}_r$ is diagonal. The rank of $\mathscr{D}_r$ is $r$ so it has exactly $r$ positive diagonal elements and the rest is zero. This argument is true for $T_p^{-1} \mathscr{D}_r T_p^{-1} = \Pi_C \Lambda_{\mathbb{I}_r} \Pi_C'$. Since $\Lambda_{\mathbb{I}_r}$ is a diagonal positive definite matrix, the $p \times r$ matrix $\Pi_C := (S_p \circ (C'C))^{-1} T_p C'$ of rank $r$ should have $p - r$ zero rows. Let $\mathbb{J}_r$ be an $r-$index set corresponding to nonzero diagonal elements of $\Pi_C \Lambda_{\mathbb{I}_r} \Pi_C'$. Then the matrix $\Pi_C[\mathbb{J}_r, \mathbb{N}_r]$ ($r \times r$ submatrix of $\Pi_C$ consist of its $\mathbb{J}_r$ rows) is nonsingular.
  2. For every $i, j \in \mathbb{J}_r$ and $i \neq j$, $(\mathscr{E}_r)_{i,j} = 0$. Since $\mathscr{E}_r := C'C$ and so $(\mathscr{E}_r)_{i,j}$ is the inner product of $i^{\text{th}}$ and $j^{\text{th}}$ columns of $C$, we conclude that the columns of $C[\mathbb{N}_r, \mathbb{J}_r]$ ($r \times r$ submatrix of $C$ consist of its $\mathbb{J}_r$ columns) are orthogonal or in other words $C[\mathbb{N}_r, \mathbb{J}_r]' C[\mathbb{N}_r, \mathbb{J}_r]$ is diagonal. The columns of $C$ are normalized. Therefore, $C[\mathbb{N}_r, \mathbb{J}_r]' C[\mathbb{N}_r, \mathbb{J}_r] = I_r$ and hence, $C[\mathbb{N}_r, \mathbb{J}_r]$ is an orthogonal matrix.

- We use the two conclusions to solve the original eq. (41) and eq. (42). First use $\boldsymbol{\Pi}_C := \left(\boldsymbol{S}_p \circ (\boldsymbol{C}'\boldsymbol{C})\right)^{-1} \boldsymbol{T}_p \boldsymbol{C}'$ to shrink them into :

$$\boldsymbol{C}\boldsymbol{T}_p\boldsymbol{\Pi}_C = \boldsymbol{\Delta}_{\mathbb{I}_r}, \tag{45}$$

$$\boldsymbol{C}\left(\boldsymbol{S}_p \circ (\boldsymbol{\Pi}_C\boldsymbol{\Lambda}_{\mathbb{I}_r}\boldsymbol{\Pi}'_C)\right)\boldsymbol{C}' = \boldsymbol{\Lambda}_{\mathbb{I}_r}\boldsymbol{\Delta}_{\mathbb{I}_r}. \tag{46}$$

Next, by the first conclusion, the matrix $\boldsymbol{T}_p^{-1}\boldsymbol{\mathscr{D}}_r\boldsymbol{T}_p^{-1} = \boldsymbol{\Pi}_C\boldsymbol{\Lambda}_{\mathbb{I}_r}\boldsymbol{\Pi}'_C$ is diagonal and so eq. (46) becomes

$$\underbrace{\boldsymbol{C}\boldsymbol{T}_p\boldsymbol{\Pi}_C}\,\boldsymbol{\Lambda}_{\mathbb{I}_r}\boldsymbol{\Pi}'_C\boldsymbol{C}' = \boldsymbol{\Lambda}_{\mathbb{I}_r}\boldsymbol{\Delta}_{\mathbb{I}_r} \xrightarrow{\text{eq. (45)}}$$

$$\boldsymbol{\Delta}_{\mathbb{I}_r}\boldsymbol{\Lambda}_{\mathbb{I}_r}\boldsymbol{\Pi}'_C\boldsymbol{C}' = \boldsymbol{\Lambda}_{\mathbb{I}_r}\boldsymbol{\Delta}_{\mathbb{I}_r} \implies$$

$$\boldsymbol{\Pi}'_C\boldsymbol{C}' = \boldsymbol{C}\boldsymbol{\Pi}_C = \boldsymbol{I}_r, \tag{47}$$

which is one of the two claimed conditions. What is left is to show that $\boldsymbol{\Pi}_C$ is a rectangular permutation matrix. From the first conclusion we also have $\boldsymbol{\Pi}_C$ has exactly $r$ nonzero columns indexed by $\mathbb{J}_r$ so

$$\boldsymbol{C}[\mathbb{N}_r, \mathbb{J}_r]\boldsymbol{\Pi}_C[\mathbb{J}_r, \mathbb{N}_r] = \boldsymbol{I}_r.$$

By the second conclusion $\boldsymbol{C}[\mathbb{N}_r, \mathbb{J}_r]$ is an orthogonal matrix therefore, $\boldsymbol{\Pi}_C[\mathbb{J}_r, \mathbb{N}_r]$ is the orthogonal matrix $\boldsymbol{C}[\mathbb{N}_r, \mathbb{J}_r]'$. Moreover, we had $\boldsymbol{T}_p^{-1}\boldsymbol{\mathscr{D}}_r\boldsymbol{T}_p^{-1} = \boldsymbol{\Pi}_C\boldsymbol{\Lambda}_{\mathbb{I}_r}\boldsymbol{\Pi}'_C$ is a $p \times p$ diagonal matrix with exactly $r$ nonzero diagonal elements. Hence, $\boldsymbol{\Pi}_C[\mathbb{N}_r, \mathbb{J}_r]\boldsymbol{\Lambda}_{\mathbb{I}_r}\boldsymbol{\Pi}'_C[\mathbb{N}_r, \mathbb{J}_r]$ is an $r \times r$ positive definite diagonal matrix with $\boldsymbol{\Lambda}_{\mathbb{I}_r}$ having distinct diagonal elements, and $\boldsymbol{\Pi}_C[\mathbb{N}_r, \mathbb{J}_r]$ being orthogonal. Therefore, $\boldsymbol{\Pi}_C[\mathbb{J}_r, \mathbb{N}_r]$ (as well as $\boldsymbol{C}[\mathbb{N}_r, \mathbb{J}_r]$) should be a square permutation matrix. Putting back the zero columns, we conclude that $\boldsymbol{C}$ should be such that $\boldsymbol{\Pi}_C := \left(\boldsymbol{S}_p \circ (\boldsymbol{C}'\boldsymbol{C})\right)^{-1} \boldsymbol{T}_p \boldsymbol{C}'$ is a rectangular permutation matrix and $\boldsymbol{C}\boldsymbol{\Pi}_C = \boldsymbol{I}_r$. Note that it is possible to further analyze these conditions and determine the exact structure of $\boldsymbol{C}$. However, this is not needed in general for the critical point analysis of the next theorem except for the case where $r = p$ and $\boldsymbol{C}$ is a square invertible matrix. In this case, square matrix $\boldsymbol{\Pi}_C$ is of full rank $p$, $\mathbb{J}_r = \mathbb{N}_p$ and therefore, $\boldsymbol{C}[\mathbb{N}_r, \mathbb{J}_r] = \boldsymbol{C}[\mathbb{N}_p, \mathbb{N}_p] = \boldsymbol{C}$. Hence, $\boldsymbol{C}$ is any square permutation matrix $\boldsymbol{\Pi}$, $\boldsymbol{C}'\boldsymbol{C} = \boldsymbol{\Pi}'\boldsymbol{\Pi} = \boldsymbol{I}_p$ and $\boldsymbol{\Pi}_C := \left(\boldsymbol{S}_p \circ (\boldsymbol{C}'\boldsymbol{C})\right)^{-1} \boldsymbol{T}_p \boldsymbol{C}' = \boldsymbol{T}_p^{-1}\boldsymbol{T}_p\boldsymbol{\Pi}' = \boldsymbol{\Pi}'$, which verifies eq. (10) and eq. (11) for $\boldsymbol{A}$ and $\boldsymbol{B}$ when $\boldsymbol{A}$ is of full rank $p$.

## A.5 PROOF OF COROLLARY 1

1. We already show in the proof Theorem 1 that for critical $(\boldsymbol{A}, \boldsymbol{B})$ the matrix $\boldsymbol{B}\boldsymbol{\Sigma}_{xx}\boldsymbol{B}'$ is given by eq. (36) that is

$$\boldsymbol{B}\boldsymbol{\Sigma}_{xx}\boldsymbol{B}' = \boldsymbol{D}^{-1}\boldsymbol{\Pi}_C\boldsymbol{\Lambda}_{\mathbb{I}_r}\boldsymbol{\Pi}'_C\boldsymbol{D}^{-1}.$$

The matrix $\boldsymbol{\Pi}_C$ is a $p \times r$ rectangular permutation matrix so $\boldsymbol{\Pi}_C\boldsymbol{\Lambda}_{\mathbb{I}_r}\boldsymbol{\Pi}'_C$ is diagonal as well as $\boldsymbol{D}^{-1}\boldsymbol{\Pi}_C\boldsymbol{\Lambda}_{\mathbb{I}_r}\boldsymbol{\Pi}'_C\boldsymbol{D}^{-1}$. Therefore, $\boldsymbol{B}\boldsymbol{\Sigma}_{xx}\boldsymbol{B}'$ is diagonal. The diagonal matrix $\boldsymbol{\Lambda}_{\mathbb{I}_r}$ is of rank $r$ therefore, $\boldsymbol{B}\boldsymbol{\Sigma}_{xx}\boldsymbol{B}'$ is of rank $r$.

2. Again by Theorem 1 critical $(\boldsymbol{A}, \boldsymbol{B})$ is of the form given by eq. (8) and eq. (9) with the proceeding conditions on the invariance $\boldsymbol{C}$. Therefore, the global map is

$$\boldsymbol{G} = \boldsymbol{A}\boldsymbol{B} = \boldsymbol{U}_{\mathbb{I}_r}\boldsymbol{C}\boldsymbol{D}\boldsymbol{D}^{-1}\boldsymbol{\Pi}_C\boldsymbol{U}'_{\mathbb{I}_r}\boldsymbol{\Sigma}_{yx}\boldsymbol{\Sigma}_{xx}^{-1}$$

$$= \boldsymbol{U}_{\mathbb{I}_r}\boldsymbol{C}\boldsymbol{\Pi}_C\boldsymbol{U}'_{\mathbb{I}_r}\boldsymbol{\Sigma}_{yx}\boldsymbol{\Sigma}_{xx}^{-1} \xrightarrow{\boldsymbol{C}\boldsymbol{\Pi}_C = \boldsymbol{I}_r}$$

$$\boldsymbol{G} = \boldsymbol{U}_{\mathbb{I}_r}\boldsymbol{U}'_{\mathbb{I}_r}\boldsymbol{\Sigma}_{yx}\boldsymbol{\Sigma}_{xx}^{-1}.$$

3. Based on Baldi & Hornik (1989) $(\boldsymbol{A}, \boldsymbol{B})$ define a critical point of $\tilde{L}(\boldsymbol{A}, \boldsymbol{B}) = \sum_{i=1}^p \|\boldsymbol{Y} - \boldsymbol{A}\boldsymbol{B}\boldsymbol{X}\|_F^2$ iff they satisfy

$$\boldsymbol{A}'\boldsymbol{A}\boldsymbol{B}\boldsymbol{\Sigma}_{xx} = \boldsymbol{A}'\boldsymbol{\Sigma}_{yx} \text{ and} \tag{48}$$

$$\boldsymbol{A}\boldsymbol{B}\boldsymbol{\Sigma}_{xx}\boldsymbol{B}' = \boldsymbol{\Sigma}_{yx}\boldsymbol{B}'. \tag{49}$$

Again by assumption $(\boldsymbol{A}, \boldsymbol{B})$ define a critical point of $L(\boldsymbol{A}, \boldsymbol{B})$ so by Theorem 1 they are of the form given by eq. (8) and eq. (9) with the proceeding conditions on the invariance $\boldsymbol{C}$. Hence,

$$\boldsymbol{A}'\boldsymbol{A}\boldsymbol{B}\boldsymbol{\Sigma}_{xx} = \boldsymbol{D}\boldsymbol{C}'\underbrace{\boldsymbol{U}'_{\mathbb{I}_r}\boldsymbol{U}_{\mathbb{I}_r}}\boldsymbol{C}\underbrace{\boldsymbol{D}\boldsymbol{D}^{-1}}\boldsymbol{\Pi}_C\boldsymbol{U}'_{\mathbb{I}_r}\boldsymbol{\Sigma}_{yx}\underbrace{\boldsymbol{\Sigma}_{xx}^{-1}\boldsymbol{\Sigma}_{xx}}$$

$$= DC' \underbrace{C\Pi_C} U'_{\mathbb{I}_r} \Sigma_{yx} \xrightarrow{C\Pi_C = I_r}$$

$$A'AB\Sigma_{xx} = DC'U'_{\mathbb{I}_r}\Sigma_{yx} = A'\Sigma_{yx}.$$

Hence, eq. (48) is satisfied. For the second equation we use the first property of this corollary that is $B\Sigma_{xx}B'$ is diagonal and satisfy eq. (7) of Proposition 2 that is

$$A\left(S_p \circ (B\Sigma_{xx}B')\right) = \Sigma_{yx}B'T_p \xrightarrow{B\Sigma_{xx}B' \text{ is diagonal}}$$

$$AT_pB\Sigma_{xx}B' = \Sigma_{yx}B'T_p \xrightarrow{B\Sigma_{xx}B' \text{ is diagonal}}$$

$$AB\Sigma_{xx}B'T_p = \Sigma_{yx}B'T_p \implies$$

$$AB\Sigma_{xx}B' = \Sigma_{yx}B'.$$

Hence, the second condition, eq. (49) is also satisfied. Therefore, any critical point of $L(A, B)$ is a critical point of $\tilde{L}(A, B)$.

## A.6 PROOF OF LEMMA 1

*Proof.* We have

$$
\begin{aligned}
L(A, B) &= \sum_{i=1}^{p} \|Y - AI_{i;p}BX\|_F^2 = \sum_{i=1}^{p} \langle Y - AI_{i;p}BX, Y - AI_{i;p}BX \rangle_F \\
&= \sum_{i=1}^{p} \left( \langle Y, Y \rangle_F + \langle Y, -AI_{i;p}BX \rangle_F + \langle -AI_{i;p}BX, Y \rangle_F \right. \\
&\quad + \langle -AI_{i;p}BX, -AI_{i;p}BX \rangle_F ) \\
&= p\langle Y, Y \rangle_F - 2\langle Y, A\left(\sum_{i=1}^{p} I_{i;p}\right) BX \rangle_F + \sum_{i=1}^{p} \langle AI_{i;p}BX, AI_{i;p}BX \rangle_F \xrightarrow{\text{eq. (25)}} \\
&= p\operatorname{Tr}(YY') - 2\operatorname{Tr}(AT_pBXY') + \sum_{i=1}^{p} \operatorname{Tr}(X'B'I_{i;p}A'AI_{i;p}BX) \\
&= p\operatorname{Tr}(\Sigma_{yy}) - 2\operatorname{Tr}(AT_pB\Sigma_{xy}) + \operatorname{Tr}\left(XX'B'\sum_{i=1}^{p}(I_{i;p}A'AI_{i;p})B\right) \xrightarrow{\text{eq. (26)}} \\
&= p\operatorname{Tr}(\Sigma_{yy}) - 2\operatorname{Tr}(AT_pB\Sigma_{xy}) + \operatorname{Tr}(B'(S_p \circ (A'A))B\Sigma_{xx}),
\end{aligned}
$$

which is eq. (17). $\qquad \square$

## A.7 PROOF OF THEOREM 2

*Proof.* The full rank matrices $A^*$ and $B^*$ given by eq. (18) and eq. (19) are clearly of the form given by Theorem 1 with $\mathbb{I}_p = \mathbb{N}_p \coloneqq \{1, 2, \cdots, p\}$, and $\Pi_p = I_p$. Hence, they define a critical point of $L(A, B)$. We want to show that these are the only local minima, that is any other critical $(A, B)$ is a saddle points. The proof is similar to the second partial derivative test. However, in this case the Hessian is a forth order tensor. Therefore, the second order Taylor approximation of the loss, derived in Lemma 4, is used directly. To prove the necessary condition, we show that at any other critical point $(A, B)$, where the first order derivatives are zero, there exists infinitesimal direction along which the second derivative of loss is negative. Next, for the sufficient condition we show that the any critical point of the form $(A^*, B^*)$ is a local and global minima.

**The necessary condition:**

Recall that $U_{\mathbb{I}_p}$ is the matrix of eigenvectors indexed by the $p-$index set $\mathbb{I}_p$ and $\Pi$ is a $p \times p$ permutation matrix. Since all the index sets $\mathbb{I}_r$, $r \le p$ are assumed to be ordered, the only way to have $U_{\mathbb{N}_p} = U_{\mathbb{I}_p}\Pi$ is by having $\mathbb{I}_p = \mathbb{N}_p$ and $\Pi = I_p$. Let $A$ (with no zero column) and $B$ define an arbitrary critical point of $L(A, B)$. Then Based on the previous theorem, either $A = U_{\mathbb{I}_r}C$ with $r < p$ or $A = U_{\mathbb{I}_p}\Pi D$ while in both cases $B = \hat{B}(A)$ given by eq. (6). If $(A, B)$ is not of the form of $(A^*, B^*)$ then there are three possibilities either 1) $A = U_{\mathbb{I}_r}CD$ with $r < p$, or 2)

$\boldsymbol{A} = \boldsymbol{U}_{\mathbb{I}_p} \boldsymbol{\Pi} \boldsymbol{D}$ with $\mathbb{I}_p \neq \mathbb{N}_p$ or 2) $\boldsymbol{A} = \boldsymbol{U}_{\mathbb{N}_p} \boldsymbol{\Pi} \boldsymbol{D}$ but $\boldsymbol{\Pi} \neq \boldsymbol{I}_p$. The first two cases corresponds to not having the "right" and/or "enough" eigenvectors, and the third corresponds to not having the "right" ordering. We introduce the following notation and investigate each case separately. Let $\varepsilon > 0$ and $\boldsymbol{U}_{i;j} \in \mathbb{R}^{n \times p}$ be a matrix of all zeros except the $i^{\text{th}}$ column, which contains $\boldsymbol{u}_j$; the eigenvector of $\boldsymbol{\Sigma}$ corresponding to the $j^{\text{th}}$ largest eigenvalue. Therefore,

$$\boldsymbol{U}'_{i;j} \boldsymbol{\Sigma} \boldsymbol{U}_{i;j} = \boldsymbol{U}'_{i;j} \boldsymbol{U} \boldsymbol{\Lambda} \boldsymbol{U}' \boldsymbol{U}_{i;j} = \lambda_j \boldsymbol{E}_i, \tag{50}$$

where, $\boldsymbol{E}_i \in \mathbb{R}^{p \times p}$ is matrix of zeros except the $i^{\text{th}}$ diagonal element that contains 1. In what follows, for each case we define a encoder direction $\boldsymbol{V} \in \mathbb{R}^{n \times p}$ with $\|\boldsymbol{V}\|_F = O(\varepsilon)$, and set the decoder direction $\boldsymbol{W} \in \mathbb{R}^{p \times n}$ as $\boldsymbol{W} = \bar{\boldsymbol{W}} := (\boldsymbol{S}_p \circ (\boldsymbol{A}'\boldsymbol{A}))^{-1} \boldsymbol{T}_p \boldsymbol{V}' \boldsymbol{\Sigma}_{yx} \boldsymbol{\Sigma}_{xx}^{-1}$. Then we use eq. (30) and eq. (31) of Lemma 4, to show that the given direction $(\boldsymbol{V}, \boldsymbol{W})$ infinitesimally reduces the loss and hence, in every case the corresponding critical $(\boldsymbol{A}, \boldsymbol{B})$ is a saddle point.

1. For the case $\boldsymbol{A} = \boldsymbol{U}_{\mathbb{I}_r} \boldsymbol{C} \boldsymbol{D}$, with $r < p$, note that based on the first item in Corollary 1, $\boldsymbol{B} \boldsymbol{\Sigma}_{xx} \boldsymbol{B}'$ is a $p \times p$ diagonal matrix of rank $r$ so it has $p - r$ zero diagonal elements. Pick an $i \in \mathbb{N}_p$ such that $(\boldsymbol{B} \boldsymbol{\Sigma}_{xx} \boldsymbol{B}')_{ii}$ is zero and a $j \in \mathbb{N}_p \setminus \mathbb{I}_r$. Set $\boldsymbol{V} = \varepsilon \boldsymbol{U}_{i;j} \boldsymbol{D}$ and $\boldsymbol{W} = \bar{\boldsymbol{W}}$. Clearly,

$$\boldsymbol{V}'\boldsymbol{A} = \varepsilon \boldsymbol{D} \boldsymbol{U}'_{i;j} \boldsymbol{U}_{\mathbb{I}_r} \boldsymbol{C} \boldsymbol{D} = 0, \tag{51}$$

$$\boldsymbol{V}'\boldsymbol{V} \boldsymbol{T}_p \boldsymbol{B} \boldsymbol{\Sigma}_{xx} \boldsymbol{B}' = \varepsilon^2 \boldsymbol{D} \underbrace{\boldsymbol{U}'_{i;j} \boldsymbol{U}_{i;j}}_{} \boldsymbol{D} \boldsymbol{T}_p \boldsymbol{B} \boldsymbol{\Sigma}_{xx} \boldsymbol{B}',$$

$$= \varepsilon^2 \boldsymbol{D} \boldsymbol{E}_i \boldsymbol{D} \boldsymbol{T}_p \boldsymbol{B} \boldsymbol{\Sigma}_{xx} \boldsymbol{B}' = \varepsilon^2 \boldsymbol{D}^2 \boldsymbol{T}_p \boldsymbol{E}_i (\boldsymbol{B} \boldsymbol{\Sigma}_{xx} \boldsymbol{B}') = 0 \text{ and} \tag{52}$$

$$\boldsymbol{V}'\boldsymbol{\Sigma}\boldsymbol{V} = \varepsilon^2 \boldsymbol{D} \boldsymbol{U}'_{i;j} \boldsymbol{U} \boldsymbol{\Lambda} \boldsymbol{U}' \boldsymbol{U}_{i;j} \boldsymbol{D} = \varepsilon^2 \lambda_j \boldsymbol{D}^2 \boldsymbol{E}_i. \tag{53}$$

Notice, $\|\boldsymbol{V}\|_F, \|\boldsymbol{W}\|_F = O(\varepsilon)$, so based on eq. (30) of Lemma 4, we have

$$L(\boldsymbol{A} + \boldsymbol{V}, \boldsymbol{B} + \boldsymbol{W}) - L(\boldsymbol{A}, \boldsymbol{B}) =$$

$$\text{Tr}\left(\boldsymbol{V}'\boldsymbol{V} \boldsymbol{T}_p \boldsymbol{B} \boldsymbol{\Sigma}_{xx} \boldsymbol{B}'\right) - \text{Tr}\left(\boldsymbol{V}'\boldsymbol{\Sigma}\boldsymbol{V} \boldsymbol{T}_p (\boldsymbol{S}_p \circ (\boldsymbol{A}'\boldsymbol{A}))^{-1} \boldsymbol{T}_p\right)$$

$$+ 2 \text{Tr}\left(\boldsymbol{V}'\boldsymbol{A} \left(\boldsymbol{S}_p \circ \left(\boldsymbol{B} \boldsymbol{\Sigma}_{xy} \boldsymbol{V} \boldsymbol{T}_p (\boldsymbol{S}_p \circ (\boldsymbol{A}'\boldsymbol{A}))^{-1} + (\boldsymbol{S}_p \circ (\boldsymbol{A}'\boldsymbol{A}))^{-1} \boldsymbol{T}_p \boldsymbol{V}' \boldsymbol{\Sigma}_{yx} \boldsymbol{B}'\right)\right)\right)$$

$$+ O(\varepsilon^3) \xrightarrow[\text{eq. (52)}]{\text{eq. (51)}}$$

$$L(\boldsymbol{A} + \boldsymbol{V}, \boldsymbol{B} + \boldsymbol{W}) - L(\boldsymbol{A}, \boldsymbol{B}) =$$

$$- \text{Tr}\left(\boldsymbol{V}'\boldsymbol{\Sigma}\boldsymbol{V} \boldsymbol{T}_p (\boldsymbol{S}_p \circ (\boldsymbol{A}'\boldsymbol{A}))^{-1} \boldsymbol{T}_p\right) + O(\varepsilon^3) \xrightarrow[\boldsymbol{A}'\boldsymbol{A} = \boldsymbol{D}\boldsymbol{C}'\boldsymbol{C}\boldsymbol{D}]{\text{eq. (53)}}$$

$$L(\boldsymbol{A} + \boldsymbol{V}, \boldsymbol{B} + \boldsymbol{W}) - L(\boldsymbol{A}, \boldsymbol{B}) =$$

$$- \varepsilon^2 \lambda_j \text{Tr}\left(\boldsymbol{D}^2 \boldsymbol{E}_i \boldsymbol{D}^{-1} \left(\left(\underbrace{\boldsymbol{T}_p^{-1} \boldsymbol{S}_p \boldsymbol{T}_p^{-1}}_{}\right) \circ (\boldsymbol{C}'\boldsymbol{C})\right)^{-1} \boldsymbol{D}^{-1}\right) + O(\varepsilon^3) =$$

$$- \varepsilon^2 \lambda_j \left(\left(\hat{\boldsymbol{S}}_p \circ (\boldsymbol{C}'\boldsymbol{C})\right)^{-1}\right)_{ii} + O(\varepsilon^3).$$

Therefore, since $\left(\hat{\boldsymbol{S}}_p \circ (\boldsymbol{C}'\boldsymbol{C})\right)^{-1}$ is a positive definite matrix, as $\varepsilon \to 0$, we have $L(\boldsymbol{A} + \boldsymbol{V}, \boldsymbol{B} + \boldsymbol{W}) \leq L(\boldsymbol{A}, \boldsymbol{B})$. Hence, any $(\boldsymbol{A}, \boldsymbol{B}) = (\boldsymbol{U}_{\mathbb{I}_r} \boldsymbol{C} \boldsymbol{D}, \hat{\boldsymbol{B}}(\boldsymbol{U}_{\mathbb{I}_r} \boldsymbol{C} \boldsymbol{D}))$ with $r < p$ is a saddle point.

2. Next, consider the case where $\boldsymbol{A} = \boldsymbol{U}_{\mathbb{I}_p} \boldsymbol{\Pi} \boldsymbol{D}$ with $\mathbb{I}_p \neq \mathbb{N}_p$. Then there exists at least one $j \in \mathbb{I}_p \setminus \mathbb{N}_p$ and $i \in \mathbb{N}_p \setminus \mathbb{I}_p$ such that $i < j$ (so $\lambda_i > \lambda_j$). Let $\sigma$ be the permutation corresponding to the permutation matrix $\boldsymbol{\Pi}$. Also, let $\varepsilon > 0$ and $\boldsymbol{U}_{\sigma(j);i} \in \mathbb{R}^{n \times p}$ be a matrix of all zeros except the $\sigma(j)^{\text{th}}$ column, which contains $\boldsymbol{u}_i$; the eigenvector of $\boldsymbol{\Sigma}$ corresponding to the $i^{\text{th}}$ largest eigenvalue. Set $\boldsymbol{V} = \varepsilon \boldsymbol{U}_{\sigma(j);i} \boldsymbol{D}$ and $\boldsymbol{W} = \bar{\boldsymbol{W}}$. Then, since $i \notin \mathbb{I}_p$ we have

$$\boldsymbol{V}'\boldsymbol{U}_{\mathbb{I}_p} = \varepsilon \boldsymbol{D} \boldsymbol{U}'_{\sigma(j);i} \boldsymbol{U}_{\mathbb{I}_p} = 0, \tag{54}$$

$$\boldsymbol{V}'\boldsymbol{V} = \varepsilon^2 \boldsymbol{D} \boldsymbol{U}'_{\sigma(j);i} \boldsymbol{U}_{\sigma(j);i} \boldsymbol{D} = \varepsilon^2 \boldsymbol{D}^2 \boldsymbol{E}_{\sigma(j)}, \text{ and} \tag{55}$$

$$\boldsymbol{V}'\boldsymbol{\Sigma}\boldsymbol{V} =\varepsilon^2 \boldsymbol{D}\boldsymbol{U}'_{\sigma(j);i}\boldsymbol{U}\boldsymbol{\Lambda}\boldsymbol{U}'\boldsymbol{U}_{\sigma(j);i}\boldsymbol{D} = \varepsilon^2 \lambda_i \boldsymbol{D}^2 \boldsymbol{E}_{\sigma(j)}. \tag{56}$$

Since $\|\boldsymbol{V}\|_F$, $\|\boldsymbol{W}\|_F = O(\varepsilon)$, based on eq. (31) of Lemma 4, we have

$$
\begin{aligned}
L(\boldsymbol{A}+\boldsymbol{V},\boldsymbol{B}+\boldsymbol{W}) - L(\boldsymbol{A},\boldsymbol{B}) = {}& \mathrm{Tr}\left(\boldsymbol{V}'\boldsymbol{V}\boldsymbol{\Pi}'\boldsymbol{\Lambda}_{\mathbb{I}_p}\boldsymbol{\Pi}\boldsymbol{T}_p\boldsymbol{D}^{-2}\right) - \mathrm{Tr}\left(\boldsymbol{V}'\boldsymbol{\Sigma}\boldsymbol{V}\boldsymbol{T}_p\boldsymbol{D}^{-2}\right) \\
& +2\,\mathrm{Tr}\left(\boldsymbol{V}'\boldsymbol{U}_{\mathbb{I}_p}\boldsymbol{\Pi}\boldsymbol{D}\left(\boldsymbol{S}_p \circ \left(\boldsymbol{D}^{-1}\boldsymbol{\Pi}'\boldsymbol{U}'_{\mathbb{I}_p}\boldsymbol{\Sigma}\boldsymbol{V}\boldsymbol{D}^{-2}\right)\right)\right) \\
& +2\,\mathrm{Tr}\left(\boldsymbol{V}'\boldsymbol{U}_{\mathbb{I}_p}\boldsymbol{\Pi}\boldsymbol{D}\left(\boldsymbol{S}_p \circ \left(\boldsymbol{D}^{-2}\boldsymbol{V}'\boldsymbol{\Sigma}\boldsymbol{U}_{\mathbb{I}_p}\boldsymbol{\Pi}\boldsymbol{D}^{-1}\right)\right)\right) \\
& +O(\varepsilon^3) \xRightarrow[\text{eq. (55),eq. (56)}]{\text{eq. (54)}} \\
L(\boldsymbol{A}+\boldsymbol{V},\boldsymbol{B}+\boldsymbol{W}) - L(\boldsymbol{A},\boldsymbol{B}) = {}& \mathrm{Tr}\left(\varepsilon^2 \boldsymbol{D}^2 \boldsymbol{E}_{\sigma(j)}\boldsymbol{\Pi}'\boldsymbol{\Lambda}_{\mathbb{I}_p}\boldsymbol{\Pi}\boldsymbol{T}_p\boldsymbol{D}^{-2}\right) \\
& - \mathrm{Tr}\left(\varepsilon^2 \lambda_i \boldsymbol{D}^2 \boldsymbol{E}_{\sigma(j)}\boldsymbol{T}_p\boldsymbol{D}^{-2}\right) + O(\varepsilon^3) \\
= {}& \varepsilon^2 \,\mathrm{Tr}\left(\underbrace{\boldsymbol{E}_{\sigma(j)}\boldsymbol{\Pi}'\boldsymbol{\Lambda}_{\mathbb{I}_p}\boldsymbol{\Pi}}\,\boldsymbol{T}_p\right) - \varepsilon^2 \lambda_i \,\mathrm{Tr}\left(\boldsymbol{E}_{\sigma(j)}\boldsymbol{T}_p\right) + O(\varepsilon^3) \\
= {}& \varepsilon^2 \lambda_j \,\mathrm{Tr}\left(\boldsymbol{E}_{\sigma(j)}\boldsymbol{T}_p\right) - \varepsilon^2 \lambda_i \,\mathrm{Tr}\left(\boldsymbol{E}_{\sigma(j)}\boldsymbol{T}_p\right) + O(\varepsilon^3) \\
= {}& -\varepsilon^2 (p - \sigma(j) + 1)(\lambda_i - \lambda_j) + O(\varepsilon^3).
\end{aligned}
$$

Note that in the above, the diagonal matrix $\boldsymbol{\Pi}'\boldsymbol{\Lambda}_{\mathbb{I}_p}\boldsymbol{\Pi}$ has the same diagonal elements as $\boldsymbol{\Lambda}_{\mathbb{I}_p}$ but they are permuted by $\sigma$. So $\boldsymbol{E}_{\sigma(j)}\boldsymbol{\Pi}'\boldsymbol{\Lambda}_{\mathbb{I}_p}\boldsymbol{\Pi}$ selects $\sigma(j)^{\text{th}}$ diagonal element of $\boldsymbol{\Pi}'\boldsymbol{\Lambda}_{\mathbb{I}_p}\boldsymbol{\Pi}$ that is the $j^{\text{th}}$diagonal element of $\boldsymbol{\Lambda}_{\mathbb{I}_p}$, which is nothing but $\lambda_j$. Now, since $i < j$ so $\lambda_i > \lambda_j$ and $\sigma(j) \le p$, as $\varepsilon \to 0$, we have $L(\boldsymbol{A}+\boldsymbol{V},\boldsymbol{B}+\boldsymbol{W}) \le L(\boldsymbol{A},\boldsymbol{B})$. Hence, any $(\boldsymbol{A},\boldsymbol{B}) = (\boldsymbol{U}_{\mathbb{I}_p}\boldsymbol{\Pi}\boldsymbol{D}, \hat{\boldsymbol{B}}(\boldsymbol{U}_{\mathbb{I}_p}\boldsymbol{\Pi}\boldsymbol{D}))$ is a saddle point.

3. Finally consider the case where $\boldsymbol{A} = \boldsymbol{U}_{\mathbb{N}_p}\boldsymbol{\Pi}\boldsymbol{D}$ with $\boldsymbol{\Pi} \ne \boldsymbol{I}_p$. Since $\boldsymbol{\Pi} \ne \boldsymbol{I}_p$, the permutation $\sigma$ of the set $\mathbb{N}_p$, corresponding to the permutation matrix $\boldsymbol{\Pi}$, has at least a cycle $(i_1 i_2 \cdots i_k)$, where $1 < i_1 < i_2 \cdots < i_k < p$ and $2 \le k \le p$. Hence, $\boldsymbol{\Pi}$ can be decomposed as $\boldsymbol{\Pi} = \boldsymbol{\Pi}_{(i_1 i_2 \cdots i_k)}\hat{\boldsymbol{\Pi}}$, where $\hat{\boldsymbol{\Pi}}$ is the permutation matrix corresponding to other cycles of $\sigma$. The cycle $(i_1 i_2 \cdots i_k)$ can be decomposed into transpositions as $(i_1 i_2 \cdots i_k) = (i_k i_{k-1}) \cdots (i_k i_1)$, which in matrix form is $\boldsymbol{\Pi}_{(i_1 i_2 \cdots i_k)} = \boldsymbol{\Pi}_{(i_k i_1)}\boldsymbol{\Pi}_{(i_k i_2)} \cdots \boldsymbol{\Pi}_{(i_k i_{k-1})}$. Therefore, $\boldsymbol{\Pi}$ can be decomposed as $\boldsymbol{\Pi} = \boldsymbol{\Pi}_{(i_k i_1)}\tilde{\boldsymbol{\Pi}}$, where $\tilde{\boldsymbol{\Pi}} = \boldsymbol{\Pi}_{(i_k i_2)} \cdots \boldsymbol{\Pi}_{(i_k i_{k-1})}\hat{\boldsymbol{\Pi}}$. Note that $\boldsymbol{\Pi}_{(i_k i_1)}$, the permutation matrix corresponding to transposition $(i_k i_1)$ is a symmetric involutory matrix, i.e. $\boldsymbol{\Pi}^2_{(i_k i_1)} = \boldsymbol{I}_p$. Set $\boldsymbol{V} = \varepsilon(\boldsymbol{U}_{i_1;i_1} - \boldsymbol{U}_{i_k;i_k})\tilde{\boldsymbol{\Pi}}\boldsymbol{D}$ and $\boldsymbol{W} = \bar{\boldsymbol{W}}$. Again we replace $\boldsymbol{V}$ and $\boldsymbol{W}$ in eq. (31) of Lemma 4. There are some tedious steps to simplify the equation, which is given in appendix A.7.1. The final result is as follows. With the given $\boldsymbol{V}$ and $\boldsymbol{W}$, the third and forth terms of the RHS of eq. (31) are canceled and the first two terms are simplified to

$$
\mathrm{Tr}\left(\boldsymbol{V}'\boldsymbol{V}\boldsymbol{\Pi}'\boldsymbol{\Lambda}_{\mathbb{N}_p}\boldsymbol{\Pi}\boldsymbol{T}_p\boldsymbol{D}^{-2}\right) = \varepsilon^2 \lambda_{i_k}(p - i_1 + 1) + \varepsilon^2 \lambda_{i_1}(p - i_m + 1), \text{ and} \tag{57}
$$
$$
\mathrm{Tr}\left(\boldsymbol{V}'\boldsymbol{\Sigma}\boldsymbol{V}\boldsymbol{T}_p\boldsymbol{D}^{-2}\right) = \varepsilon^2 \lambda_{i_1}(p - i_1 + 1) + \varepsilon^2 \lambda_{i_k}(p - i_m + 1), \tag{58}
$$

in which, $m = \max\{k - 1, 2\}$. This means that If the selected cycle is just a transposition $(i_1 i_2)$ then $i_m = i_2$. But if for the selected cycle $(i_1 i_2 \cdots i_k)$, $k$ is greater than 2 then $i_m = i_{k-1}$. Using above equations, eq. (31) yields

$$
\begin{aligned}
L(\boldsymbol{A}+\boldsymbol{V},\boldsymbol{B}+\boldsymbol{W}) - L(\boldsymbol{A},\boldsymbol{B}) = {}& \mathrm{Tr}\left(\boldsymbol{V}'\boldsymbol{V}\boldsymbol{\Pi}'\boldsymbol{\Lambda}_{\mathbb{I}_p}\boldsymbol{\Pi}\boldsymbol{T}_p\boldsymbol{D}^{-2}\right) - \mathrm{Tr}\left(\boldsymbol{V}'\boldsymbol{\Sigma}\boldsymbol{V}\boldsymbol{T}_p\boldsymbol{D}^{-2}\right) + O(\varepsilon^3) \\
= {}& \varepsilon^2 \lambda_{i_k}(p - i_1 + 1) + \varepsilon^2 \lambda_{i_1}(p - i_m + 1) \\
& -\varepsilon^2 \lambda_{i_1}(p - i_1 + 1) - \varepsilon^2 \lambda_{i_k}(p - i_m + 1) + O(\varepsilon^3) \\
= {}& -\varepsilon^2 i_1 \lambda_{i_k} - \varepsilon^2 i_m \lambda_{i_1} + \varepsilon^2 i_1 \lambda_{i_1} + \varepsilon^2 i_m \lambda_{i_k} \\
= {}& -\varepsilon^2 \left((\lambda_{i_1} - \lambda_{i_k})(i_m - i_1)\right) + O(\varepsilon^3). \tag{59}
\end{aligned}
$$

By the above definition of $i_m$, we have $i_m - i_1 > 0$ and since $i_1 < i_k$, $\lambda_{i_1} - \lambda_{i_k} > 0$. Hence, the first term in the above equation is negative and as $\varepsilon \to 0$, we have $L(\boldsymbol{A}+\boldsymbol{V},\boldsymbol{B}+\boldsymbol{W}) - L(\boldsymbol{A},\boldsymbol{B}) < 0$. Therefore, any any $(\boldsymbol{A},\boldsymbol{B}) = (\boldsymbol{U}_{\mathbb{I}_p}\boldsymbol{\Pi}\boldsymbol{D}, \hat{\boldsymbol{B}}(\boldsymbol{U}_{\mathbb{I}_p}\boldsymbol{\Pi}\boldsymbol{D}))$ with $\boldsymbol{\Pi} \ne \boldsymbol{I}_p$ is a saddle point.

**The Sufficient condition:**

From Lemma 1 we know that the loss $L(\boldsymbol{A}, \boldsymbol{B})$ can be written in the form of eq. (17). Use this equation to evaluate loss at $(\boldsymbol{A}^*, \boldsymbol{B}^*) = \left(\boldsymbol{U}_{\mathbb{N}_p} \boldsymbol{D}_p, \boldsymbol{D}_p^{-1} \boldsymbol{U}'_{\mathbb{N}_p} \boldsymbol{\Sigma}_{yx} \boldsymbol{\Sigma}_{xx}^{-1}\right)$ as follows

$$L(\boldsymbol{A}^*, \boldsymbol{B}^*) = p \operatorname{Tr}(\boldsymbol{\Sigma}_{yy}) - 2 \operatorname{Tr}\left(\boldsymbol{A}^* \boldsymbol{T}_p \boldsymbol{B}^* \boldsymbol{\Sigma}_{xy}\right) + \operatorname{Tr}\left(\boldsymbol{B}^{*'}\left(\boldsymbol{S}_p \circ \left(\boldsymbol{A}^{*'} \boldsymbol{A}^*\right)\right) \boldsymbol{B}^* \boldsymbol{\Sigma}_{xx}\right) \implies$$

$$L(\boldsymbol{A}^*, \boldsymbol{B}^*) = p \operatorname{Tr}(\boldsymbol{\Sigma}_{yy}) - 2 \operatorname{Tr}\left(\boldsymbol{U}_{\mathbb{N}_p} \boldsymbol{D}_p \boldsymbol{T}_p \boldsymbol{D}_p^{-1} \boldsymbol{U}'_{\mathbb{N}_p} \underbrace{\boldsymbol{\Sigma}_{yx} \boldsymbol{\Sigma}_{xx}^{-1} \boldsymbol{\Sigma}_{xy}}\right)$$

$$+ \operatorname{Tr}\left(\left(\boldsymbol{S}_p \circ \left(\boldsymbol{D}_p \underbrace{\boldsymbol{U}'_{\mathbb{N}_p} \boldsymbol{U}_{\mathbb{N}_p}} \boldsymbol{D}_p\right)\right) \boldsymbol{D}_p^{-1} \boldsymbol{U}'_{\mathbb{N}_p} \underbrace{\boldsymbol{\Sigma}_{yx} \boldsymbol{\Sigma}_{xx}^{-1} \boldsymbol{\Sigma}_{xx} \boldsymbol{\Sigma}_{xx}^{-1} \boldsymbol{\Sigma}_{xy}} \boldsymbol{U}_{\mathbb{N}_p} \boldsymbol{D}_p^{-1}\right) \implies$$

$$L(\boldsymbol{A}^*, \boldsymbol{B}^*) = p \operatorname{Tr}(\boldsymbol{\Sigma}_{yy}) - 2 \operatorname{Tr}\left(\boldsymbol{T}_p \underbrace{\boldsymbol{D}_p \boldsymbol{D}_p^{-1}} \underbrace{\boldsymbol{U}'_{\mathbb{N}_p} \boldsymbol{\Sigma} \boldsymbol{U}_{\mathbb{N}_p}}\right)$$

$$+ \operatorname{Tr}\left(\left(\underbrace{\boldsymbol{S}_p \circ (\boldsymbol{I}_p)}\right) \underbrace{\boldsymbol{D}_p \boldsymbol{D}_p^{-1}} \underbrace{\boldsymbol{U}'_{\mathbb{N}_p} \boldsymbol{\Sigma} \boldsymbol{U}_{\mathbb{N}_p}} \underbrace{\boldsymbol{D}_p^{-1} \boldsymbol{D}_p}\right) \implies$$

$$L(\boldsymbol{A}^*, \boldsymbol{B}^*) = p \operatorname{Tr}(\boldsymbol{\Sigma}_{yy}) - 2 \operatorname{Tr}\left(\boldsymbol{T}_p \boldsymbol{\Lambda}_{\mathbb{N}_p}\right) + \operatorname{Tr}\left(\boldsymbol{T}_p \boldsymbol{\Lambda}_{\mathbb{N}_p}\right) \implies$$

$$L(\boldsymbol{A}^*, \boldsymbol{B}^*) = p \operatorname{Tr}(\boldsymbol{\Sigma}_{yy}) - \operatorname{Tr}\left(\boldsymbol{T}_p \boldsymbol{\Lambda}_{\mathbb{N}_p}\right) = p \operatorname{Tr}(\boldsymbol{\Sigma}_{yy}) - \sum_{i=1}^{p} (p - i + 1) \lambda_i,$$

which is eq. (20), as claimed. Notice that the above value is independent of the diagonal matrix $\boldsymbol{D}_p$. From the necessary condition we know that any critical point not in the form of $(\boldsymbol{A}^*, \boldsymbol{B}^*)$ is a saddle point. Hence, due to the convexity of the loss at least one $(\boldsymbol{A}^*, \boldsymbol{B}^*)$ is a global minimum but since the value of the loss at $(\boldsymbol{A}^*, \boldsymbol{B}^*)$ is independent of $\boldsymbol{D}_p$ all these critical points yield the same value for the loss. Therefore, any critical point in the form of $(\boldsymbol{A}^*, \boldsymbol{B}^*)$ is a local and global minima. $\quad\square$

### A.7.1 SUPPLEMENTARY DETAILS OF THE PROOF OF THEOREM 2

To verify eq. (57), eq. (58), and eq. (59) in the proof of Theorem 2, we want to replace $\boldsymbol{V}$ and $\boldsymbol{W}$ in eq. (31) of Lemma 4 with $\boldsymbol{V} = \varepsilon(\boldsymbol{U}_{i_1;i_1} - \boldsymbol{U}_{i_k;i_k})\tilde{\boldsymbol{\Pi}} \boldsymbol{D}$ and $\boldsymbol{W} = \bar{\boldsymbol{W}}$ and simplify. eq. (31) is

$$L(\boldsymbol{A} + \boldsymbol{V}, \boldsymbol{B} + \boldsymbol{W}) - L(\boldsymbol{A}, \boldsymbol{B}) = \operatorname{Tr}\left(\boldsymbol{V}'\boldsymbol{V}\boldsymbol{\Pi}'\boldsymbol{\Lambda}_{\mathbb{I}_p}\boldsymbol{\Pi}\boldsymbol{T}_p\boldsymbol{D}^{-2}\right) - \operatorname{Tr}\left(\boldsymbol{V}'\boldsymbol{\Sigma}\boldsymbol{V}\boldsymbol{T}_p\boldsymbol{D}^{-2}\right)$$

$$+ 2 \operatorname{Tr}\left(\boldsymbol{V}'\boldsymbol{U}_{\mathbb{I}_p}\boldsymbol{\Pi}\boldsymbol{D}\left(\boldsymbol{S}_p \circ \left(\boldsymbol{D}^{-1}\boldsymbol{\Pi}'\boldsymbol{U}'_{\mathbb{I}_p}\boldsymbol{\Sigma}\boldsymbol{V}\boldsymbol{D}^{-2}\right)\right)\right)$$

$$+ 2 \operatorname{Tr}\left(\boldsymbol{V}'\boldsymbol{U}_{\mathbb{I}_p}\boldsymbol{\Pi}\boldsymbol{D}\left(\boldsymbol{S}_p \circ \left(\boldsymbol{D}^{-2}\boldsymbol{V}'\boldsymbol{\Sigma}\boldsymbol{U}_{\mathbb{I}_p}\boldsymbol{\Pi}\boldsymbol{D}^{-1}\right)\right)\right)$$

$$+ O(\varepsilon^3).$$

We investigate each term on the RHS separately. but before note that

$$\boldsymbol{E}_i \tilde{\boldsymbol{\Pi}} \boldsymbol{T}_p \tilde{\boldsymbol{\Pi}}' = \left(\tilde{\boldsymbol{\Pi}} \boldsymbol{T}_p \tilde{\boldsymbol{\Pi}}'\right)_{i,i} \boldsymbol{E}_i = (\boldsymbol{T}_p)_{\tilde{\sigma}^{-1}(i), \tilde{\sigma}^{-1}(i)} \boldsymbol{E}_i = (p - \tilde{\sigma}^{-1}(i) + 1) \boldsymbol{E}_i, \tag{60}$$

where, $\tilde{\sigma}$ and its function inverse $\tilde{\sigma}^{-1}$ are permutations corresponding to $\tilde{\boldsymbol{\Pi}}$ and $\tilde{\boldsymbol{\Pi}}'$ respectively. $\tilde{\boldsymbol{\Pi}} \boldsymbol{T}_p \tilde{\boldsymbol{\Pi}}'$ is a diagonal matrix where diagonal elements of $\boldsymbol{T}_p$ are ordered based on $\tilde{\sigma}^{-1}$. Moreover, recall that we decomposed the permutation matrix $\boldsymbol{\Pi}$ in $\boldsymbol{A}$ with a cycle $(i_1 i_2 \cdots i_k)$ as $\boldsymbol{\Pi} = \boldsymbol{\Pi}_{(i_1 i_k)} \underbrace{\boldsymbol{\Pi}_{(i_k i_2)} \cdots \boldsymbol{\Pi}_{(i_k i_{k-1})} \hat{\boldsymbol{\Pi}}} = \boldsymbol{\Pi}_{(i_1 i_k)} \hat{\boldsymbol{\Pi}}$, where $i_1, i_2, \cdots i_k$ are fixed points of $\hat{\boldsymbol{\Pi}}$. Therefore, with $\tilde{\sigma}$ being the permutation corresponding to $\tilde{\boldsymbol{\Pi}}$ we have

$$\tilde{\sigma}(i_1) = i_1 \implies \tilde{\sigma}^{-1}(i_1) = i_1, \text{ and} \tag{61}$$

$$\tilde{\sigma}(i_{k-1}) = i_m \implies \tilde{\sigma}^{-1}(i_k) = i_m, \tag{62}$$

where, $m = \max\{k - 1, 2\}$. This means that If the selected cycle is just a transposition $(i_1 i_2)$ then $i_m = i_2$. But if for the selected cycle $(i_1 i_2 \cdots i_k)$, $k$ is greater than 2 then $i_m = i_{k-1}$.

For the first term we have

$$\boldsymbol{V}'\boldsymbol{V} = \varepsilon^2 \boldsymbol{D}\tilde{\boldsymbol{\Pi}}'(\boldsymbol{U}'_{i_1;i_1} - \boldsymbol{U}'_{i_k;i_k})(\boldsymbol{U}_{i_1;i_1} - \boldsymbol{U}_{i_k;i_k})\tilde{\boldsymbol{\Pi}}\boldsymbol{D} \xrightarrow{\boldsymbol{U}'_{i_1;i_1}\boldsymbol{U}_{i_k;i_k} = 0}$$

$$\boldsymbol{V'V} = \varepsilon^2 \boldsymbol{D}\tilde{\boldsymbol{\Pi}}'(\boldsymbol{U}'_{i_1;i_1}\boldsymbol{U}_{i_1;i_1} + \boldsymbol{U}'_{i_k;i_k}\boldsymbol{U}_{i_k;i_k})\tilde{\boldsymbol{\Pi}}\boldsymbol{D} \xrightarrow[\boldsymbol{U}'_{i_k;i_k}\boldsymbol{U}_{i_k;i_k}=\boldsymbol{E}_{i_k}]{\boldsymbol{U}'_{i_1;i_1}\boldsymbol{U}_{i_1;i_1}=\boldsymbol{E}_{i_1}}$$

$$\boldsymbol{V'V} = \varepsilon^2 \boldsymbol{D}\tilde{\boldsymbol{\Pi}}'(\boldsymbol{E}_{i_1} + \boldsymbol{E}_{i_k})\tilde{\boldsymbol{\Pi}}\boldsymbol{D} \xrightarrow{\tilde{\boldsymbol{\Pi}}'(\boldsymbol{E}_{i_1}+\boldsymbol{E}_{i_k})\tilde{\boldsymbol{\Pi}} \text{ is diagonal}}$$

$$\boldsymbol{V'V} = \varepsilon^2 \tilde{\boldsymbol{\Pi}}'(\boldsymbol{E}_{i_1} + \boldsymbol{E}_{i_k})\tilde{\boldsymbol{\Pi}}\boldsymbol{D}^2 \implies$$

$$\mathrm{Tr}\left(\boldsymbol{V'V}\boldsymbol{\Pi}'\boldsymbol{\Lambda}_{\mathbb{N}_p}\boldsymbol{\Pi}\boldsymbol{T}_p\boldsymbol{D}^{-2}\right) = \mathrm{Tr}\left(\widetilde{\boldsymbol{V'V}}\,\boldsymbol{D}^{-2}\tilde{\boldsymbol{\Pi}}'\boldsymbol{\Pi}_{(i_1 i_k)}\boldsymbol{\Lambda}_{\mathbb{N}_p}\boldsymbol{\Pi}_{(i_1 i_k)}\tilde{\boldsymbol{\Pi}}\boldsymbol{T}_p\right)$$

$$= \mathrm{Tr}\left(\varepsilon^2\tilde{\boldsymbol{\Pi}}'(\boldsymbol{E}_{i_1} + \boldsymbol{E}_{i_k})\underbrace{\tilde{\boldsymbol{\Pi}}\boldsymbol{D}^2\boldsymbol{D}^{-2}\tilde{\boldsymbol{\Pi}}'}_{\boldsymbol{I}_p}\boldsymbol{\Pi}_{(i_1 i_k)}\boldsymbol{\Lambda}_{\mathbb{N}_p}\boldsymbol{\Pi}_{(i_1 i_k)}\tilde{\boldsymbol{\Pi}}\boldsymbol{T}_p\right)$$

$$= \varepsilon^2\,\mathrm{Tr}\left((\boldsymbol{E}_{i_1} + \boldsymbol{E}_{i_k})\boldsymbol{\Pi}_{(i_1 i_k)}\boldsymbol{\Lambda}_{\mathbb{N}_p}\boldsymbol{\Pi}_{(i_1 i_k)}\tilde{\boldsymbol{\Pi}}\boldsymbol{T}_p\tilde{\boldsymbol{\Pi}}'\right)$$

$$= \varepsilon^2\,\mathrm{Tr}\left(\lambda_{i_k}\boldsymbol{E}_{i_1}\tilde{\boldsymbol{\Pi}}\boldsymbol{T}_p\tilde{\boldsymbol{\Pi}}' + \lambda_{i_1}\boldsymbol{E}_{i_k}\tilde{\boldsymbol{\Pi}}\boldsymbol{T}_p\tilde{\boldsymbol{\Pi}}'\right) \xrightarrow{\text{eq. (60)}}$$

$$\mathrm{Tr}\left(\boldsymbol{V'V}\boldsymbol{\Pi}'\boldsymbol{\Lambda}_{\mathbb{N}_p}\boldsymbol{\Pi}\boldsymbol{T}_p\boldsymbol{D}^{-2}\right) = \varepsilon^2\lambda_{i_k}(p - \tilde{\sigma}^{-1}(i_1) + 1)\boldsymbol{E}_{i_1} + \varepsilon^2\lambda_{i_1}(p - \tilde{\sigma}^{-1}(i_k) + 1)\boldsymbol{E}_{i_k} \xrightarrow[\text{eq. (62)}]{\text{eq. (61)}}$$

$$\mathrm{Tr}\left(\boldsymbol{V'V}\boldsymbol{\Pi}'\boldsymbol{\Lambda}_{\mathbb{N}_p}\boldsymbol{\Pi}\boldsymbol{T}_p\boldsymbol{D}^{-2}\right) = \varepsilon^2\lambda_{i_k}(p - i_1 + 1)\boldsymbol{E}_{i_1} + \varepsilon^2\lambda_{i_1}(p - i_m + 1)\boldsymbol{E}_{i_k},$$

which is eq. (57) as claimed.

For the second term we have

$$\boldsymbol{V'\Sigma V} = \varepsilon^2 \boldsymbol{D}\tilde{\boldsymbol{\Pi}}'(\boldsymbol{U}'_{i_1;i_1} - \boldsymbol{U}'_{i_k;i_k})\boldsymbol{U}\boldsymbol{\Lambda}\boldsymbol{U}'(\boldsymbol{U}_{i_1;i_1} - \boldsymbol{U}_{i_k;i_k})\tilde{\boldsymbol{\Pi}}\boldsymbol{D}$$

$$= \varepsilon^2 \boldsymbol{D}\tilde{\boldsymbol{\Pi}}'(\underbrace{\boldsymbol{U}'_{i_1;i_1}\boldsymbol{U}\boldsymbol{\Lambda}\boldsymbol{U}'\boldsymbol{U}_{i_1;i_1}}_{\lambda_{i_1}\boldsymbol{E}_{i_1}} - \underbrace{\boldsymbol{U}'_{i_1;i_1}\boldsymbol{U}\boldsymbol{\Lambda}\boldsymbol{U}'\boldsymbol{U}_{i_k;i_k}}_{0}$$

$$- \underbrace{\boldsymbol{U}'_{i_k;i_k}\boldsymbol{U}\boldsymbol{\Lambda}\boldsymbol{U}'\boldsymbol{U}_{i_1;i_1}}_{0} + \underbrace{\boldsymbol{U}'_{i_k;i_k}\boldsymbol{U}\boldsymbol{\Lambda}\boldsymbol{U}'\boldsymbol{U}_{i_k;i_k}}_{\lambda_{i_k}\boldsymbol{E}_{i_k}})\tilde{\boldsymbol{\Pi}}\boldsymbol{D}$$

$$= \varepsilon^2\tilde{\boldsymbol{\Pi}}'(\lambda_{i_1}\boldsymbol{E}_{i_1} + \lambda_{i_k}\boldsymbol{E}_{i_k})\tilde{\boldsymbol{\Pi}}\boldsymbol{D}^2 \implies$$

$$\mathrm{Tr}\left(\boldsymbol{V'\Sigma V}\boldsymbol{T}_p\boldsymbol{D}^{-2}\right) = \mathrm{Tr}\left(\varepsilon^2\tilde{\boldsymbol{\Pi}}'(\lambda_{i_1}\boldsymbol{E}_{i_1} + \lambda_{i_k}\boldsymbol{E}_{i_k})\tilde{\boldsymbol{\Pi}}\boldsymbol{D}^2\boldsymbol{T}_p\boldsymbol{D}^{-2}\right)$$

$$= \varepsilon^2\,\mathrm{Tr}\left(\lambda_{i_1}\boldsymbol{E}_{i_1}\tilde{\boldsymbol{\Pi}}\boldsymbol{T}_p\tilde{\boldsymbol{\Pi}}' + \lambda_{i_k}\boldsymbol{E}_{i_k}\tilde{\boldsymbol{\Pi}}\boldsymbol{T}_p\tilde{\boldsymbol{\Pi}}'\right) \xrightarrow{\text{eq. (60)}}$$

$$\mathrm{Tr}\left(\boldsymbol{V'\Sigma V}\boldsymbol{T}_p\boldsymbol{D}^{-2}\right) = \varepsilon^2\lambda_{i_1}(p - \tilde{\sigma}^{-1}(i_1) + 1) + \varepsilon^2\lambda_{i_k}(p - \tilde{\sigma}^{-1}(i_k) + 1) \xrightarrow[\text{eq. (62)}]{\text{eq. (61)}}$$

$$\mathrm{Tr}\left(\boldsymbol{V'\Sigma V}\boldsymbol{T}_p\boldsymbol{D}^{-2}\right) = \varepsilon^2\lambda_{i_1}(p - i_1 + 1) + \varepsilon^2\lambda_{i_k}(p - i_m + 1),$$

which is eq. (58) as claimed.

Finally, we have to show that the third and the forth terms of the eq. (31) are canceled. First, observe that

$$\mathrm{Tr}\left(\boldsymbol{V'}\boldsymbol{U}_{\mathbb{N}_p}\boldsymbol{\Pi}\boldsymbol{D}\left(\boldsymbol{S}_p \circ \left(\boldsymbol{D}^{-1}\boldsymbol{\Pi}'\boldsymbol{U}'_{\mathbb{N}_p}\boldsymbol{\Sigma}\boldsymbol{V}\boldsymbol{D}^{-2}\right)\right)\right) =$$

$$\mathrm{Tr}\left(\varepsilon\boldsymbol{D}\tilde{\boldsymbol{\Pi}}'(\boldsymbol{U}'_{i_1;i_1} - \boldsymbol{U}'_{i_k;i_k})\boldsymbol{U}_{\mathbb{N}_p}\boldsymbol{\Pi}\left(\boldsymbol{S}_p \circ \left(\boldsymbol{\Pi}'\boldsymbol{U}'_{\mathbb{N}_p}\boldsymbol{\Sigma}\boldsymbol{V}\boldsymbol{D}^{-2}\right)\right)\right) =$$

$$\varepsilon\,\mathrm{Tr}\left(\tilde{\boldsymbol{\Pi}}'(\boldsymbol{E}_{i_1} - \boldsymbol{E}_{i_k})\boldsymbol{\Pi}\left(\boldsymbol{S}_p \circ \left(\boldsymbol{\Pi}'\boldsymbol{U}'_{\mathbb{N}_p}\boldsymbol{\Sigma}\boldsymbol{V}\boldsymbol{D}^{-2}\right)\right)\boldsymbol{D}\right) =$$

$$\varepsilon^2\,\mathrm{Tr}\left(\tilde{\boldsymbol{\Pi}}'(\boldsymbol{E}_{i_1} - \boldsymbol{E}_{i_k})\boldsymbol{\Pi}\left(\boldsymbol{S}_p \circ \left(\boldsymbol{\Pi}'(\lambda_{i_1}\boldsymbol{E}_{i_1} - \lambda_{i_k}\boldsymbol{E}_{i_k})\tilde{\boldsymbol{\Pi}}\right)\right)\right) =$$

$$\varepsilon^2\,\mathrm{Tr}\left((\boldsymbol{E}_{i_1} - \boldsymbol{E}_{i_k})\left(\left(\boldsymbol{\Pi}\boldsymbol{S}_p\tilde{\boldsymbol{\Pi}}'\right) \circ \left(\boldsymbol{\Pi}\boldsymbol{\Pi}'(\lambda_{i_1}\boldsymbol{E}_{i_1} - \lambda_{i_k}\boldsymbol{E}_{i_k})\tilde{\boldsymbol{\Pi}}\tilde{\boldsymbol{\Pi}}'\right)\right)\right) =$$

$$\varepsilon^2\,\mathrm{Tr}\left(\left(\boldsymbol{\Pi}\boldsymbol{S}_p\tilde{\boldsymbol{\Pi}}'\right) \circ ((\boldsymbol{E}_{i_1} - \boldsymbol{E}_{i_k})(\lambda_{i_1}\boldsymbol{E}_{i_1} - \lambda_{i_k}\boldsymbol{E}_{i_k}))\right) =$$

$$\varepsilon^2\,\mathrm{Tr}\left(\left(\boldsymbol{\Pi}\boldsymbol{S}_p\tilde{\boldsymbol{\Pi}}'\right) \circ (\lambda_{i_1}\boldsymbol{E}_{i_1} + \lambda_{i_k}\boldsymbol{E}_{i_k})\right), \text{ and}$$

$$\mathrm{Tr}\left(\boldsymbol{V'}\boldsymbol{U}_{\mathbb{N}_p}\boldsymbol{\Pi}\boldsymbol{D}\left(\boldsymbol{S}_p \circ \left(\boldsymbol{D}^{-2}\boldsymbol{V'}\boldsymbol{\Sigma}\boldsymbol{U}_{\mathbb{N}_p}\boldsymbol{\Pi}\boldsymbol{D}^{-1}\right)\right)\right) =$$

$$\mathrm{Tr}\left(\varepsilon \boldsymbol{D}\tilde{\boldsymbol{\Pi}}'(\boldsymbol{U}'_{i_1;i_1} - \boldsymbol{U}'_{i_k;i_k})\boldsymbol{U}_{\mathbb{N}_p}\boldsymbol{\Pi}\left(\boldsymbol{S}_p \circ \left(\boldsymbol{D}^{-1}\boldsymbol{V}'\boldsymbol{\Sigma}\boldsymbol{U}_{\mathbb{N}_p}\boldsymbol{\Pi}\boldsymbol{D}^{-1}\right)\right)\right) =$$

$$\varepsilon \,\mathrm{Tr}\left(\tilde{\boldsymbol{\Pi}}'(\boldsymbol{E}_{i_1} - \boldsymbol{E}_{i_k})\boldsymbol{\Pi}\left(\boldsymbol{S}_p \circ \left(\boldsymbol{D}^{-1}\boldsymbol{V}'\boldsymbol{\Sigma}\boldsymbol{U}_{\mathbb{N}_p}\boldsymbol{\Pi}\right)\right)\right) =$$

$$\varepsilon^2\,\mathrm{Tr}\left((\boldsymbol{E}_{i_1} - \boldsymbol{E}_{i_k})\boldsymbol{\Pi}\left(\boldsymbol{S}_p \circ \left(\tilde{\boldsymbol{\Pi}}'(\lambda_{i_1}\boldsymbol{E}_{i_1} - \lambda_{i_k}\boldsymbol{E}_{i_k})\boldsymbol{\Pi}\right)\right)\tilde{\boldsymbol{\Pi}}'\right) =$$

$$\varepsilon^2\,\mathrm{Tr}\left((\boldsymbol{E}_{i_1} - \boldsymbol{E}_{i_k})\left(\left(\boldsymbol{\Pi}\boldsymbol{S}_p\tilde{\boldsymbol{\Pi}}'\right) \circ \left(\boldsymbol{\Pi}\tilde{\boldsymbol{\Pi}}'(\lambda_{i_1}\boldsymbol{E}_{i_1} - \lambda_{i_k}\boldsymbol{E}_{i_k})\boldsymbol{\Pi}\tilde{\boldsymbol{\Pi}}'\right)\right)\right) =$$

$$\varepsilon^2\,\mathrm{Tr}\left((\boldsymbol{E}_{i_1} - \boldsymbol{E}_{i_k})\left(\left(\boldsymbol{\Pi}\boldsymbol{S}_p\tilde{\boldsymbol{\Pi}}'\right) \circ \left(\boldsymbol{\Pi}_{(i_1 i_k)}(\lambda_{i_1}\boldsymbol{E}_{i_1} - \lambda_{i_k}\boldsymbol{E}_{i_k})\boldsymbol{\Pi}_{(i_1 i_k)}\right)\right)\right) =$$

$$\varepsilon^2\,\mathrm{Tr}\left((\boldsymbol{E}_{i_1} - \boldsymbol{E}_{i_k})\left(\left(\boldsymbol{\Pi}\boldsymbol{S}_p\tilde{\boldsymbol{\Pi}}'\right) \circ \left((\lambda_{i_1}\boldsymbol{E}_{i_k} - \lambda_{i_k}\boldsymbol{E}_{i_1})\right)\right)\right) =$$

$$\varepsilon^2\,\mathrm{Tr}\left(\left(\boldsymbol{\Pi}\boldsymbol{S}_p\tilde{\boldsymbol{\Pi}}'\right) \circ \left((\boldsymbol{E}_{i_1} - \boldsymbol{E}_{i_k})(\lambda_{i_1}\boldsymbol{E}_{i_k} - \lambda_{i_k}\boldsymbol{E}_{i_1})\right)\right) =$$

$$-\varepsilon^2\,\mathrm{Tr}\left(\left(\boldsymbol{\Pi}\boldsymbol{S}_p\tilde{\boldsymbol{\Pi}}'\right) \circ (\lambda_{i_1}\boldsymbol{E}_{i_k} + \lambda_{i_k}\boldsymbol{E}_{i_1})\right) =$$

$$-\varepsilon^2\,\mathrm{Tr}\left(\left(\boldsymbol{\Pi}\boldsymbol{S}_p\tilde{\boldsymbol{\Pi}}'\right) \circ (\lambda_{i_1}\boldsymbol{E}_{i_k} + \lambda_{i_k}\boldsymbol{E}_{i_1})\right).$$

Now, note that in both cases the matrices that are multiplied elementwise with $\boldsymbol{\Pi}\boldsymbol{S}_p\tilde{\boldsymbol{\Pi}}'$ are diagonal and hence, we only need to look at diagonal elements of $\boldsymbol{\Pi}\boldsymbol{S}_p\tilde{\boldsymbol{\Pi}}'$. Moreover,

$$\boldsymbol{\Pi}\boldsymbol{S}_p\tilde{\boldsymbol{\Pi}}' = \boldsymbol{\Pi}_{(i_1 i_k)}\boldsymbol{\Pi}_{(i_k i_2)}\cdots\boldsymbol{\Pi}_{(i_k i_{k-1})}\hat{\boldsymbol{\Pi}}\boldsymbol{S}_p\hat{\boldsymbol{\Pi}}'\boldsymbol{\Pi}_{(i_k i_{k-1})}\cdots\boldsymbol{\Pi}_{(i_k i_2)},$$

where, $i_1 \cdots i_k$ are fixed points of permutation corresponding to $\hat{\boldsymbol{\Pi}}$ so $\hat{\boldsymbol{\Pi}}\boldsymbol{S}_p\hat{\boldsymbol{\Pi}}'$ has the same values at diagonal positions $i_1$ and $i_k$ as the original matrix $\boldsymbol{S}_p$. The only permutation that is only on the left side is $\boldsymbol{\Pi}_{(i_1 i_k)}$ which exchanges the $i_1$ and $i_k$ rows of $\boldsymbol{S}_p$. Since $\boldsymbol{S}_p$ is such that the elements at each row before the diagonal element are the same and $i_k > i_1$, we have the $i_1$ and $i_k$ diagonal elements of $\boldsymbol{\Pi}\boldsymbol{S}_p\tilde{\boldsymbol{\Pi}}'$ have the same value. Let that value be denoted as $s$. Then the sum of the above two equations yields $m(\lambda_{i_1} + \lambda_{i_k}) - m(\lambda_{i_1} + \lambda_{i_k}) = 0$, as claimed.

# B    DERIVATIVES OF THE LOSS FUNCTION

## B.1    FIRST AND SECOND ORDER FRÉCHET DERIVATIVE

In order to derive and analyze the critical points of the cost function which is a real-valued function of matrices we use the first and second order Fréchet derivatives as described in chapter 4 of Zeidler (1995). For a function $f : \mathbb{R}^{n \times m} \to \mathbb{R}$ the first order Fréchet derivative at the point $\boldsymbol{A} \in \mathbb{R}^{n \times m}$ is a linear functional $df(\boldsymbol{A}) : \mathbb{R}^{n \times m} \to \mathbb{R}$ such that

$$\lim_{\boldsymbol{V} \to 0} \frac{|f(\boldsymbol{A} + \boldsymbol{V}) - f(\boldsymbol{A}) - df(\boldsymbol{A})\boldsymbol{V}|}{\|\boldsymbol{V}\|_F} = 0,$$

where we used the shorthand $df(\boldsymbol{A})\boldsymbol{V} \equiv (df(\boldsymbol{A}))(\boldsymbol{V})$. Similarly, the 2nd derivative is a bilinear functional $d^2 f(\boldsymbol{A}) : \mathbb{R}^{n \times m} \times \mathbb{R}^{n \times m} \to \mathbb{R}$ such that

$$\lim_{\boldsymbol{V} \to 0} \frac{|df(\boldsymbol{A} + \boldsymbol{V})K - df(\boldsymbol{A})K - d^2 f(\boldsymbol{A})\boldsymbol{V}K|}{\|\boldsymbol{V}\|_F} = 0,$$

for all $\|K\|_F \leq 1$, where again $d^2 f(\boldsymbol{A})\boldsymbol{V}K \equiv (d^2 f(\boldsymbol{A}))(\boldsymbol{V}, K)$. The generalized Taylor formula then becomes:

$$f(\boldsymbol{A} + \boldsymbol{V}) = f(\boldsymbol{A}) + df(\boldsymbol{A})\boldsymbol{V} + \frac{1}{2}d^2 f(\boldsymbol{A})\boldsymbol{V}^2 + o(\|\boldsymbol{V}\|^2),$$

Moreover, we derive functions $\nabla f : \mathbb{R}^{n \times m} \to \mathbb{R}^{n \times m}$ and $\boldsymbol{H}(\boldsymbol{A}) : \mathbb{R}^{n \times m} \to \mathbb{R}^{n \times m}$ such that $df(\boldsymbol{A})\boldsymbol{V} = \langle \nabla f(\boldsymbol{A}), \boldsymbol{V} \rangle_F$ and $d^2 f(\boldsymbol{A})\boldsymbol{V}^2 = \langle \boldsymbol{H}(\boldsymbol{A})\boldsymbol{V}, \boldsymbol{V} \rangle_F$, where again $\boldsymbol{H}(\boldsymbol{A})\boldsymbol{V} \equiv \boldsymbol{H}(\boldsymbol{A})(\boldsymbol{V})$. Then clearly, $\boldsymbol{A} \in \mathbb{R}^{n \times m}$ is a critical point of $f$ iff $\nabla f(\boldsymbol{A}) = 0$ and for such $\boldsymbol{A}$s the sign of the bilinear form $\langle \boldsymbol{H}(\boldsymbol{A})\boldsymbol{V}, \boldsymbol{V} \rangle$ over *directions* $\boldsymbol{V}$ determines the type of the critical point.

Extending the generalized Taylor theorem of Zeidler (1995), the second order Taylor expansion for the loss $L(\boldsymbol{A}, \boldsymbol{B})$ is then given by

$$\begin{aligned} L(\boldsymbol{A} + \boldsymbol{V}, \boldsymbol{B} + \boldsymbol{W}) - L(\boldsymbol{A}, \boldsymbol{B}) =& d_{\boldsymbol{A}}L(\boldsymbol{A}, \boldsymbol{B})\boldsymbol{V} + d_{\boldsymbol{B}}L(\boldsymbol{A}, \boldsymbol{B})\boldsymbol{W} + \frac{1}{2}d_{\boldsymbol{A}}^2 L(\boldsymbol{A}, \boldsymbol{B})\boldsymbol{V}^2 \\ &+ d_{\boldsymbol{AB}}L(\boldsymbol{A}, \boldsymbol{B})\boldsymbol{V}\boldsymbol{W} + \frac{1}{2}d_{\boldsymbol{B}}^2 L(\boldsymbol{A}, \boldsymbol{B})\boldsymbol{W}^2 + R_{\boldsymbol{V}, \boldsymbol{W}}(\boldsymbol{A}, \boldsymbol{B}), \end{aligned} \tag{63}$$

where, if $\|\boldsymbol{V}\|_F, \|\boldsymbol{W}\|_F = O(\varepsilon)$ then $\|R(\boldsymbol{V}, \boldsymbol{W})\| = O(\varepsilon^3)$. Clearly, as at critical points where $d_{\boldsymbol{A}}L(\boldsymbol{A}, \boldsymbol{B})\boldsymbol{V} + d_{\boldsymbol{B}}L(\boldsymbol{A}, \boldsymbol{B})\boldsymbol{W} = 0$, as $\varepsilon \to 0$ we have $R_{\boldsymbol{V}, \boldsymbol{W}}(\boldsymbol{A}, \boldsymbol{B}) \to 0$ and the sign of the sum of the second order partial Fréchet derivatives determines the type of the critical point very much similar to second partial derivative test for two variable functions. However, here for local minima we have to show the sign is positive in all directions and for saddle points have to show the sign is positive in some directions and negative at least in on direction. Finally, note that the smoothness of the loss entails that Fréchet derivative and directional derivative (Gateaux) both exist and (foregoing some subtleties in definition) are the same.

## B.2    FIRST AND SECOND ORDER DERIVATIVE OF THE LOSS WRT TO $\boldsymbol{B}$

**Lemma 5.** *The first and second (partial Fréchet ) derivative of the loss $L(\boldsymbol{A}, \boldsymbol{B})$ wrt to $\boldsymbol{B}$ is derived as follows.*

$$d_{\boldsymbol{B}}L(\boldsymbol{A}, \boldsymbol{B})\boldsymbol{W} = -2\operatorname{Tr}\left(\boldsymbol{W}'\left(\boldsymbol{T}_p \boldsymbol{A}' \boldsymbol{\Sigma}_{yx} - (\boldsymbol{S}_p \circ (\boldsymbol{A}' \boldsymbol{A}))\, \boldsymbol{B} \boldsymbol{\Sigma}_{xx}\right)\right) \tag{64}$$

$$= -2\langle \boldsymbol{T}_p \boldsymbol{A}' \boldsymbol{\Sigma}_{yx} - (\boldsymbol{S}_p \circ (\boldsymbol{A}' \boldsymbol{A}))\, \boldsymbol{B} \boldsymbol{\Sigma}_{xx}, \boldsymbol{W} \rangle_F. \tag{65}$$

$$d_{\boldsymbol{B}^2}^2 L(\boldsymbol{A}, \boldsymbol{B})\boldsymbol{W}^2 = 2\langle (\boldsymbol{S}_p \circ (\boldsymbol{A}' \boldsymbol{A}))\, \boldsymbol{W} \boldsymbol{\Sigma}_{xx}, \boldsymbol{W} \rangle_F = 2\operatorname{Tr}\left(\boldsymbol{W}'\left(\boldsymbol{S}_p \circ (\boldsymbol{A}' \boldsymbol{A})\right)\boldsymbol{W} \boldsymbol{\Sigma}_{xx}\right). \tag{66}$$

*Proof.* Directly compute

$$L(\boldsymbol{A}, \boldsymbol{B} + \boldsymbol{W}) = \sum_{i=1}^{p} \|\boldsymbol{Y} - \boldsymbol{A}\boldsymbol{I}_{i;p}(\boldsymbol{B} + \boldsymbol{W})\boldsymbol{X}\|_F^2$$

$$= \sum_{i=1}^{p} \langle \boldsymbol{Y} - \boldsymbol{A}\boldsymbol{I}_{i;p}(\boldsymbol{B} + \boldsymbol{W})\boldsymbol{X}, \boldsymbol{Y} - \boldsymbol{A}\boldsymbol{I}_{i;p}(\boldsymbol{B} + \boldsymbol{W})\boldsymbol{X} \rangle_F$$

$$= \sum_{i=1}^{p} \langle \boldsymbol{Y} - \boldsymbol{A}\boldsymbol{I}_{i;p}\boldsymbol{B}\boldsymbol{X}, \boldsymbol{Y} - \boldsymbol{A}\boldsymbol{I}_{i;p}\boldsymbol{B}\boldsymbol{X} \rangle_F + \sum_{i=1}^{p} \langle \boldsymbol{Y} - \boldsymbol{A}\boldsymbol{I}_{i;p}\boldsymbol{B}\boldsymbol{X}, -\boldsymbol{A}\boldsymbol{I}_{i;p}\boldsymbol{W}\boldsymbol{X} \rangle_F$$

$$+ \sum_{i=1}^{p} \langle -\boldsymbol{A}\boldsymbol{I}_{i;p}\boldsymbol{W}\boldsymbol{X}, \boldsymbol{Y} - \boldsymbol{A}\boldsymbol{I}_{i;p}\boldsymbol{B}\boldsymbol{X} \rangle_F + \sum_{i=1}^{p} \langle -\boldsymbol{A}\boldsymbol{I}_{i;p}\boldsymbol{W}\boldsymbol{X}, -\boldsymbol{A}\boldsymbol{I}_{i;p}\boldsymbol{W}\boldsymbol{X} \rangle_F$$

$$= L(\boldsymbol{A}, \boldsymbol{B}) - \sum_{i=1}^{p} 2\langle \boldsymbol{Y} - \boldsymbol{A}\boldsymbol{I}_{i;p}\boldsymbol{B}\boldsymbol{X}, \boldsymbol{A}\boldsymbol{I}_{i;p}\boldsymbol{W}\boldsymbol{X} \rangle + O(\|\boldsymbol{W}\|_F^2) \implies$$

$$L(\boldsymbol{A}, \boldsymbol{B} + \boldsymbol{W}) - L(\boldsymbol{A}, \boldsymbol{B}) = -2\sum_{i=1}^{p} \langle \boldsymbol{Y} - \boldsymbol{A}\boldsymbol{I}_{i;p}\boldsymbol{B}\boldsymbol{X}, \boldsymbol{A}\boldsymbol{I}_{i;p}\boldsymbol{W}\boldsymbol{X} \rangle_F + O(\|\boldsymbol{W}\|_F^2) \overset{\boldsymbol{W} \to 0}{\implies}$$

$$d_{\boldsymbol{B}}L(\boldsymbol{A}, \boldsymbol{B})\boldsymbol{W} = -2\sum_{i=1}^{p} \mathrm{Tr}(\boldsymbol{X}'\boldsymbol{W}'\boldsymbol{I}_{i;p}\boldsymbol{A}'(\boldsymbol{Y} - \boldsymbol{A}\boldsymbol{I}_{i;p}\boldsymbol{B}\boldsymbol{X}))$$

$$= -2\,\mathrm{Tr}\left(\boldsymbol{W}'\left(\left(\sum_{i=1}^{p} \boldsymbol{I}_{i;p}\right)\boldsymbol{A}'\boldsymbol{Y}\boldsymbol{X}' - \left(\sum_{i=1}^{p} \boldsymbol{I}_{i;p}\boldsymbol{A}'\boldsymbol{A}\boldsymbol{I}_{i;p}\right)\boldsymbol{B}\boldsymbol{X}\boldsymbol{X}'\right)\right)$$

$$= -2\,\mathrm{Tr}\left(\boldsymbol{W}'\left(\boldsymbol{T}_p\boldsymbol{A}'\boldsymbol{Y}\boldsymbol{X}' - (\boldsymbol{S}_p \circ (\boldsymbol{A}'\boldsymbol{A}))\boldsymbol{B}\boldsymbol{X}\boldsymbol{X}'\right)\right),$$

which can be written as the given form. For the second derivative wrt $\boldsymbol{B}$ we have

$$d_{\boldsymbol{B}}L(\boldsymbol{A}, \boldsymbol{B})\boldsymbol{W} = -2\langle \boldsymbol{T}_p\boldsymbol{A}'\boldsymbol{\Sigma}_{yx} - (\boldsymbol{S}_p \circ (\boldsymbol{A}'\boldsymbol{A}))\boldsymbol{B}\boldsymbol{\Sigma}_{xx}, \boldsymbol{W} \rangle_F \implies$$

$$d_{\boldsymbol{B}}L(\boldsymbol{A}, \boldsymbol{B} + \bar{\boldsymbol{W}})\boldsymbol{W} = -2\langle \boldsymbol{T}_p\boldsymbol{A}'\boldsymbol{\Sigma}_{yx} - (\boldsymbol{S}_p \circ (\boldsymbol{A}'\boldsymbol{A}))(\boldsymbol{B} + \bar{\boldsymbol{W}})\boldsymbol{\Sigma}_{xx}, \boldsymbol{W} \rangle_F$$

$$= -2\langle \boldsymbol{T}_p\boldsymbol{A}'\boldsymbol{\Sigma}_{yx} - (\boldsymbol{S}_p \circ (\boldsymbol{A}'\boldsymbol{A}))\boldsymbol{B}\boldsymbol{\Sigma}_{xx}, \boldsymbol{W} \rangle_F$$

$$+ 2\langle (\boldsymbol{S}_p \circ (\boldsymbol{A}'\boldsymbol{A}))\bar{\boldsymbol{W}}\boldsymbol{\Sigma}_{xx}, \boldsymbol{W} \rangle_F \implies$$

$$d_{\boldsymbol{B}}L(\boldsymbol{A}, \boldsymbol{B} + \bar{\boldsymbol{W}})\boldsymbol{W} - d_{\boldsymbol{B}}L(\boldsymbol{A}, \boldsymbol{B})\boldsymbol{W} = 2\langle (\boldsymbol{S}_p \circ (\boldsymbol{A}'\boldsymbol{A}))\bar{\boldsymbol{W}}\boldsymbol{\Sigma}_{xx}, \boldsymbol{W} \rangle_F,$$

which by having $\bar{\boldsymbol{W}} \to 0$ results in the second order partial derivative. $\square$

### B.3 FIRST AND SECOND ORDER DERIVATIVE OF THE LOSS WRT TO $\boldsymbol{A}$

**Lemma 6.** *The first and second (partial Fréchet) derivative of the loss $L(\boldsymbol{A}, \boldsymbol{B})$ wrt to $\boldsymbol{A}$ is derived as follows.*

$$d_{\boldsymbol{A}}L(\boldsymbol{A}, \boldsymbol{B})\boldsymbol{V} = -2\langle \boldsymbol{\Sigma}_{yx}\boldsymbol{B}'\boldsymbol{T}_p - \boldsymbol{A}(\boldsymbol{S}_p \circ (\boldsymbol{B}\boldsymbol{\Sigma}_{xx}\boldsymbol{B}')), \boldsymbol{V} \rangle_F, \quad (67)$$

$$d_{\boldsymbol{A}\boldsymbol{B}}^2 L(\boldsymbol{A}, \boldsymbol{B})\boldsymbol{V}\boldsymbol{W} = -2\langle \boldsymbol{\Sigma}_{yx}\boldsymbol{W}'\boldsymbol{T}_p - \boldsymbol{A}(\boldsymbol{S}_p \circ (\boldsymbol{B}\boldsymbol{\Sigma}_{xx}\boldsymbol{W}')) - \boldsymbol{A}(\boldsymbol{S}_p \circ (\boldsymbol{W}\boldsymbol{\Sigma}_{xx}\boldsymbol{B}')), \boldsymbol{V} \rangle_F, \quad (68)$$

$$d_{\boldsymbol{A}^2}^2 L(\boldsymbol{A}, \boldsymbol{B})\boldsymbol{V}^2 = 2\langle \boldsymbol{V}(\boldsymbol{S}_p \circ (\boldsymbol{B}\boldsymbol{\Sigma}_{xx}\boldsymbol{B}')), \boldsymbol{V} \rangle_F. \quad (69)$$

*Proof.* Directly compute

$$L(\boldsymbol{A} + \boldsymbol{V}, \boldsymbol{B}) = \sum_{i=1}^{p} \langle \boldsymbol{Y} - (\boldsymbol{A} + \boldsymbol{V})\boldsymbol{I}_{i;p}\boldsymbol{B}\boldsymbol{X}, \boldsymbol{Y} - (\boldsymbol{A} + \boldsymbol{V})\boldsymbol{I}_{i;p}\boldsymbol{B}\boldsymbol{X} \rangle_F$$

$$= \sum_{i=1}^{p} \langle \boldsymbol{Y} - \boldsymbol{A}\boldsymbol{I}_{i;p}\boldsymbol{B}\boldsymbol{X}, \boldsymbol{Y} - \boldsymbol{A}\boldsymbol{I}_{i;p}\boldsymbol{B}\boldsymbol{X} \rangle_F - \sum_{i=1}^{p} \langle \boldsymbol{Y} - \boldsymbol{A}\boldsymbol{I}_{i;p}\boldsymbol{B}\boldsymbol{X}, \boldsymbol{V}\boldsymbol{I}_{i;p}\boldsymbol{B}\boldsymbol{X} \rangle_F$$

$$+ \sum_{i=1}^{p} \langle -\boldsymbol{V}\boldsymbol{I}_{i;p}\boldsymbol{B}\boldsymbol{X}, \boldsymbol{Y} - \boldsymbol{A}\boldsymbol{I}_{i;p}\boldsymbol{B}\boldsymbol{X} \rangle_F + \sum_{i=1}^{p} \langle -\boldsymbol{V}\boldsymbol{I}_{i;p}\boldsymbol{B}\boldsymbol{X}, -\boldsymbol{V}\boldsymbol{I}_{i;p}\boldsymbol{B}\boldsymbol{X} \rangle_F$$

$$= L(\boldsymbol{A}, \boldsymbol{B}) - \sum_{i=1}^{p} 2\langle \boldsymbol{Y} - \boldsymbol{A}\boldsymbol{I}_{i;p}\boldsymbol{B}\boldsymbol{X}, \boldsymbol{V}\boldsymbol{I}_{i;p}\boldsymbol{B}\boldsymbol{X} \rangle_F + \sum_{i=1}^{p} \langle \boldsymbol{V}\boldsymbol{I}_{i;p}\boldsymbol{B}\boldsymbol{X}, \boldsymbol{V}\boldsymbol{I}_{i;p}\boldsymbol{B}\boldsymbol{X} \rangle_F$$

$$L(\boldsymbol{A} + \boldsymbol{V}, \boldsymbol{B}) - L(\boldsymbol{A}, \boldsymbol{B}) = -\sum_{i=1}^{p} 2\langle \boldsymbol{Y} - \boldsymbol{A}\boldsymbol{I}_{i;p}\boldsymbol{B}\boldsymbol{X}, \boldsymbol{V}\boldsymbol{I}_{i;p}\boldsymbol{B}\boldsymbol{X}\rangle_F + O(\|\boldsymbol{V}\|_F^2) \stackrel{\boldsymbol{V}\to 0}{\Longrightarrow}$$

$$d_{\boldsymbol{A}}L(\boldsymbol{A}, \boldsymbol{B})\boldsymbol{V} = -\sum_{i=1}^{p} 2\langle \boldsymbol{Y} - \boldsymbol{A}\boldsymbol{I}_{i;p}\boldsymbol{B}\boldsymbol{X}, \boldsymbol{V}\boldsymbol{I}_{i;p}\boldsymbol{B}\boldsymbol{X}\rangle_F$$

$$= -2\operatorname{Tr}(\boldsymbol{V}'(\boldsymbol{\Sigma}_{yx}\boldsymbol{B}'\sum_{i=1}^{p}\boldsymbol{I}_{i;p} - \boldsymbol{A}\sum_{i=1}^{p}\boldsymbol{I}_{i;p}\boldsymbol{B}\boldsymbol{\Sigma}_{xx}\boldsymbol{B}'\boldsymbol{I}_{i;p})) \implies$$

$$d_{\boldsymbol{A}}L(\boldsymbol{A}, \boldsymbol{B})\boldsymbol{V} = -2\langle \boldsymbol{\Sigma}_{yx}\boldsymbol{B}'\boldsymbol{T}_p - \boldsymbol{A}\left(\boldsymbol{S}_p \circ (\boldsymbol{B}\boldsymbol{\Sigma}_{xx}\boldsymbol{B}')\right), \boldsymbol{V}\rangle_F \implies$$

$$d_{\boldsymbol{A}}L(\boldsymbol{A} + \bar{\boldsymbol{V}}, \boldsymbol{B})\boldsymbol{V} = -2\langle \boldsymbol{\Sigma}_{yx}\boldsymbol{B}'\boldsymbol{T}_p - (\boldsymbol{A} + \bar{\boldsymbol{V}})\left(\boldsymbol{S}_p \circ (\boldsymbol{B}\boldsymbol{\Sigma}_{xx}\boldsymbol{B}')\right), \boldsymbol{V}\rangle_F$$

$$d_{\boldsymbol{A}}L(\boldsymbol{A} + \bar{\boldsymbol{V}}, \boldsymbol{B})\boldsymbol{V} - d_{\boldsymbol{A}}L(\boldsymbol{A}, \boldsymbol{B})\boldsymbol{V} = 2\langle \bar{\boldsymbol{V}}\left(\boldsymbol{S}_p \circ (\boldsymbol{B}\boldsymbol{\Sigma}_{xx}\boldsymbol{B}')\right), \boldsymbol{V}\rangle_F \stackrel{\bar{\boldsymbol{V}}\to 0}{\Longrightarrow}$$

$$d^2_{\boldsymbol{A}^2}L(\boldsymbol{A}, \boldsymbol{B})(\boldsymbol{V}, \bar{\boldsymbol{V}}) = 2\langle \bar{\boldsymbol{V}}\left(\boldsymbol{S}_p \circ (\boldsymbol{B}\boldsymbol{\Sigma}_{xx}\boldsymbol{B}')\right), \boldsymbol{V}\rangle_F \implies$$

$$d^2_{\boldsymbol{A}^2}L(\boldsymbol{A}, \boldsymbol{B})\boldsymbol{V}^2 = 2\langle \boldsymbol{V}\left(\boldsymbol{S}_p \circ (\boldsymbol{B}\boldsymbol{\Sigma}_{xx}\boldsymbol{B}')\right), \boldsymbol{V}\rangle_F$$

$$\begin{aligned}
d_{\boldsymbol{A}}L(\boldsymbol{A}, \boldsymbol{B} + \boldsymbol{W})\boldsymbol{V} = &-2\langle \boldsymbol{\Sigma}_{yx}(\boldsymbol{B} + \boldsymbol{W})'\boldsymbol{T}_p, \boldsymbol{V}\rangle_F \\
&-2\langle -\boldsymbol{A}\left(\boldsymbol{S}_p \circ ((\boldsymbol{B} + \boldsymbol{W})\boldsymbol{\Sigma}_{xx}(\boldsymbol{B} + \boldsymbol{W})')\right), \boldsymbol{V}\rangle_F \\
&-2\langle \boldsymbol{\Sigma}_{yx}\boldsymbol{B}'\boldsymbol{T}_p - \boldsymbol{A}\left(\boldsymbol{S}_p \circ (\boldsymbol{B}\boldsymbol{\Sigma}_{xx}\boldsymbol{B}')\right), \boldsymbol{V}\rangle_F \\
= &d_{\boldsymbol{A}}L(\boldsymbol{A}, \boldsymbol{B})\boldsymbol{V} - 2\langle \boldsymbol{\Sigma}_{yx}\boldsymbol{W}'\boldsymbol{T}_p, \boldsymbol{V}\rangle_F \\
&-2\langle -\boldsymbol{A}\left(\boldsymbol{S}_p \circ (\boldsymbol{B}\boldsymbol{\Sigma}_{xx}\boldsymbol{W}')\right) - \boldsymbol{A}\left(\boldsymbol{S}_p \circ (\boldsymbol{W}\boldsymbol{\Sigma}_{xx}\boldsymbol{B}')\right), \boldsymbol{V}\rangle_F + O(\|\boldsymbol{W}\|_F^2) \implies
\end{aligned}$$

$$\begin{aligned}
d_{\boldsymbol{A}}L(\boldsymbol{A}, \boldsymbol{B} + \boldsymbol{W})\boldsymbol{V} - d_{\boldsymbol{A}}L(\boldsymbol{A}, \boldsymbol{B})\boldsymbol{V} = &-2\langle \boldsymbol{\Sigma}_{yx}\boldsymbol{W}'\boldsymbol{T}_p, \boldsymbol{V}\rangle_F \\
&-2\langle -\boldsymbol{A}\left(\boldsymbol{S}_p \circ (\boldsymbol{B}\boldsymbol{\Sigma}_{xx}\boldsymbol{W}')\right) - \boldsymbol{A}\left(\boldsymbol{S}_p \circ (\boldsymbol{W}\boldsymbol{\Sigma}_{xx}\boldsymbol{B}')\right), \boldsymbol{V}\rangle_F \\
&+ O(\|\boldsymbol{W}\|_F^2) \stackrel{\boldsymbol{W}\to 0}{\Longrightarrow}
\end{aligned}$$

$$d^2_{\boldsymbol{A}\boldsymbol{B}}L(\boldsymbol{A}, \boldsymbol{B})\boldsymbol{V}\boldsymbol{W} = -2\langle \boldsymbol{\Sigma}_{yx}\boldsymbol{W}'\boldsymbol{T}_p - \boldsymbol{A}\left(\boldsymbol{S}_p \circ (\boldsymbol{B}\boldsymbol{\Sigma}_{xx}\boldsymbol{W}')\right) - \boldsymbol{A}\left(\boldsymbol{S}_p \circ (\boldsymbol{W}\boldsymbol{\Sigma}_{xx}\boldsymbol{B}')\right), \boldsymbol{V}\rangle_F.$$

$$\square$$

