# OpenReview forum: "Neural Networks for Principal Component Analysis: A New Loss Function Provably Yields Ordered Exact Eigenvectors "
_ICLR.cc/2020/Conference — Reject_

### Official Review · AnonReviewer2 · 2019-10-21
**Official Blind Review #2**

**Rating:** 6

**Review:**

This paper proposes a new loss function to compute the exact ordered eigenvectors of a dataset. The loss is motivated from the idea of computing the eigenvectors sequentially. However doing so would be computationally expensive, and the authors show that the loss function they propose (sum of sequential losses) has the same order (constant less than 7) of computational complexity as using the squared loss. A proof of the correctness of the algorithm is given, along with experiments to verify its performance.

The loss function proposed in the paper is useful, and the decomposition in Lemma 1 shows that it is not computationally expensive. While the writing of the proofs of the theorems is clear, I find it hard to understand the flow of the paper. It would help if the authors could summarize the argument of the proof at the start of Sec 4. Along a similar vein, it would also help if the authors could describe in words the significance / claim of every theorem.

The repetition in stating the theorems can be avoided. The main result (Theorem 2) is stated twice.

**Experience Assessment:**

I do not know much about this area.

**Review Assessment: Checking Correctness Of Derivations And Theory:**

I assessed the sensibility of the derivations and theory.

**Review Assessment: Checking Correctness Of Experiments:**

I carefully checked the experiments.

**Review Assessment: Thoroughness In Paper Reading:**

I read the paper at least twice and used my best judgement in assessing the paper.

---

> ### Author Response · Authors · 2019-11-13
> **Response to Reviewer #2**
>
> Thanks. We have added one paragraph at the start of section 4 that provides an overview of the arguments of the proofs. For the other point, please refer to remarks 2, 3 (new), 4, and 5 which hopefully clarifies the significance of the theorems.

---

### Official Review · AnonReviewer1 · 2019-10-23
**Official Blind Review #1**

**Rating:** 6

**Review:**

This paper proposes a new loss function for performing principal component analysis (PCA) using linear autoencoders (LAEs). With this new loss function, the decoder weights of LAEs can eventually converge to the exact ordered unnormalized eigenvectors of the sample covariance matrix. The main contribution is to add the identifiability of principal components in PCA using LAEs and. Two empirical experiments were done to show the effectiveness of proposed loss function on one synthetic dataset and the MNIST dataset.
Overall, this paper provides a nontrivial contribution for performing principal component analysis (PCA) using linear autoencoders (LAEs), with this new novel loss function. This paper is well presented.
There are some issues to be addressed:
1. The output matrix is constrained to be the same size of the input, which is scarcely seen in practical applications.
2. Literature on (denoising) auto-encoder can be reviewed more thoroughly.
3. Comparison with state-of-the-art auto-encoder can be provided to demonstrate the effectiveness of the proposed algorithm.
4. It is better to explain the meaning of each variable when it first appears, e.g., , the projection matrices A and B, and Variable A* in theorem 2.
5. In the experiment part, in both the Synthetic Data or MNIST, the size of each data set is relatively small. It's better to add experimental results on big data sets with larger dimension.
6. In order to better show the effectiveness of the new loss function, you can add some comparative test for different choice of compressed dimension p.
7. There are some typos, such as ‘faila’ in the second line of the second paragraph in the INTRODUCTION.


**Experience Assessment:**

I have read many papers in this area.

**Review Assessment: Checking Correctness Of Derivations And Theory:**

I assessed the sensibility of the derivations and theory.

**Review Assessment: Checking Correctness Of Experiments:**

I assessed the sensibility of the experiments.

**Review Assessment: Thoroughness In Paper Reading:**

I read the paper at least twice and used my best judgement in assessing the paper.

---

> ### Author Response · Authors · 2019-11-13
> **Response to Reviewer #1**
>
> This reviewer's comments were especially valuable in helping to establish where some assumptions were in fact not needed.
>
> 1. This is a very good point. After careful examination, in the case that input and output have different dimensions, say ${Y}\in \mathbb{R}^{n\times m}$ and ${X}\in \mathbb{R}^{n'\times m}$, all the claims actually still hold and the given loss can be used as a linear least square regressor. In the writing we assumed the same dimension since the focus was on low rank decomposition where ${Y}={X}$. We have added a remark (6) in the paper to be explicit about this fact. The reason for the validity of the claims for the case $n'\neq n$ as explained in the remark is as follows:
>
> The given loss function structurally is very similar to MSE loss and can be represented as a sum of Frobenius norms on the space of $n\times m$ matrices. In this case the covariance matrix $ {\Sigma}={\Sigma}_{yx} {\Sigma}_{xx}^{-1} {\Sigma}_{xy}$ is still $n\times n$. Clearly, for under-constrained systems with $n<n'$ the full rank assumption of $ {\Sigma}$ holds. For the overdetermined case, where $n'>n$ the second and third assumptions in Assumption 1 can be relaxed: we only require ${\Sigma}_{xx}$ to be full rank since this is the only matrix that is inverted in the theorems. Note that if $p>\min(n',n)$ then ${\Lambda}_{\mathbb{I}_p}$: the $p\times p$ diagonal matrix of eigenvalues of ${\Sigma}$ for a $p$-index-set $\mathbb{I}_p$ bounds to have some zeros and will be say rank $r<p$, which in turn, results in an encoder with rank $r$. However, the Theorem 1 is proved for encoder of any rank $r\leq p$. Finally, following Theorem 2 then the first $r$ columns of the encoder ${A}$ converges to ordered eigenvectors of ${\Sigma}$ while the $p-r$ remaining columns span the kernel (sub)space of ${\Sigma}$.
>
> 2 and 3- we have updated the introduction.
>
> 4-Fixed
>
> 5- Dealing with large datasets is a leading edge of our algorithm when the whole data is too large to fit in memory. We don't expect the performance to be different if we switch to a larger dataset since our algorithm allows processing the data in batches, in which case the algorithm will yield the result that converges to the desired real ordered eigenvectors as well.
>
> 6- We conducted extra experiments on MNIST dataset with compressed dimension p being 1,5,10,20,50 and 100. The settings of the other parameters is the same as the ones shown in our paper. The results are as follows: reconstruction error is 2.857e6, 2.113e6, 1.619e6, 1.127e6, 5.546e5, 2.700e5, respectively, and the total running time on average (with one GeForce GTX 1080 Ti Graphics Card) is 0.253 seconds, 7.062 seconds, 26.855 seconds, 4 minutes 18.408 seconds, 17 minutes 10.213 seconds, 35 minutes 31.986 seconds, respectively.

---

### Official Review · AnonReviewer3 · 2019-10-24
**Official Blind Review #3**

**Rating:** 3

**Review:**

This paper proposes and analyzes a new loss function for linear autoencoders (LAEs) whose minima directly recover the principal components of the data. The core idea is to simultaneously solve a set of MSE LAE problems with tied weights and increasingly stringent masks on the encoder/decoder matrices. My intuition is that the weights that touch every subproblem are the most motivated to find the largest principal component, the weights that touch all but one find the next largest, and so forth; I found this idea clever and elegant.

That said, I lean towards rejection, because the paper does not do a very good job of demonstrating the practical or theoretical utility of this approach. As I see it, there are two main claims that one could make to motivate this work:
1. This is a practical algorithm for doing PCA.
2. This is a step towards better understanding (and perhaps improving) nonlinear autoencoders, which do things that PCA can't.
Claim (2) might be compelling, but the authors do not make it, and it isn't self evident.

I do not find claim (1) convincing on the basis of the evidence presented. PCA is an extremely well studied problem, with lots of good solutions such as randomized SVD (Halko et al., 2009). A possible advantage of using LAEs to address the PCA problem is that they play nicely with SGD, but again, the claim that the SGD-LAE approach is superior to, say, randomized SVD on a data subsample requires evidence. Also, even if one buys the claim that LAEs are a good way to solve PCA, one can always recover the eigenvectors/eigenvalues by a final decomposition step; the authors claim that an advantage of their approach is that it does not require such "bells and whistles", but this seems like a pretty minor consideration; implementing the proposed loss function seems at least as complicated as making a call to an SVD solver, and it's hard for me to imagine a situation where the cost of that final SVD isn't dominated by the cost of solving the MSE LAE problem.

In summary, I think this paper proposes a clever and elegant solution to a problem that doesn't seem to be very important. I can't recommend acceptance unless the authors can come up with a stronger argument for why it's not just interesting but also useful.

**Experience Assessment:**

I have read many papers in this area.

**Review Assessment: Checking Correctness Of Derivations And Theory:**

I assessed the sensibility of the derivations and theory.

**Review Assessment: Checking Correctness Of Experiments:**

I carefully checked the experiments.

**Review Assessment: Thoroughness In Paper Reading:**

I read the paper at least twice and used my best judgement in assessing the paper.

---

> ### Author Response · Authors · 2019-11-13
> **Response to Reviewer #3- Part 2/2**
>
> (2) Practical implications:
> The question of whether randomized SVD outperforms SGB-based methods, or visa versa, remains an open one, and is a likely data-dependent question, as factors such as size and sparsity are important.  There have been several developments by others who themselves have outlined the benefits of SGD-based PCA/SVD (for instance, [6] and also cf. their quote from [3]). Chief among the compellingly reasons is that, in recent years, we have seen unprecedented gains in the performance of very large SGD optimizations, with autoencoders in particular successfully handling larger numbers of high-dimensional training data (e.g., images). The loss function we offer is attractive in terms of parallelizability and distributability, and does not prescribe any single specific algorithm or implementation, so stands to continue to benefit from the arms race between SGD and its competitors.
>
> Finally, the submission's focus has been on rigorously establishing theoretical properties. It is not our current focus of interest, for instance, to conduct a size analysis, as this is better deferred to some a treatment with a clear and specific characterization of problem instances of particular interest. In contrast, our own research directions involve us seeking to generalize the theory to tensors and tensor rank decomposition.
>
> Next, we offer a response to the suggestion that one can always recover the eigenvectors/eigenvalues by a final decomposition step and the SVD would be dominated by MSE LAE costs anyway. We clarify our position in two parts:
>
> (1) The exact cost of a post hoc processing step to perform the SVD will depend on the density and size of the data. It isn't hard to imagine circumstances (e.g., with large, dense inputs) in which even on the reduced output, this cubic step dominates and is prohibitive.
>
> (2) More importantly, this single loss function (without an additional post hoc processing step) fits seamlessly into optimization pipelines (where SGD is but one instance). The result is that the loss allows for PCA/SVD computation
> as a single optimization layer, akin to an instance of a fully differentiable building block in a NN pipeline [7], potentially as part of a much larger network.
>
> In light of the importance of (2), we intend to make this benefit much more explicit in the paper's introduction.
>
> [1] Baldi, Pierre, and Kurt Hornik. "Neural networks and principal component analysis: Learning from examples without local minima." Neural networks 2.1 (1989): 53-58.
>
> [2] Zhou, Y., and Y. Liang. "Critical points of linear neural networks: Analytical forms and landscape properties." Proc. Sixth International Conference on Learning Representations (ICLR). 2018.
>
> [3] Kunin, Daniel, et al. "Loss Landscapes of Regularized Linear Autoencoders." International Conference on Machine Learning. 2019.
>
> [4] Pretorius, Arnu, Steve Kroon, and Herman Kamper. "Learning Dynamics of Linear Denoising Autoencoders." International Conference on Machine Learning. 2018.
>
> [5] Frye, Charles G., et al. "Numerically Recovering the Critical Points of a Deep Linear Autoencoder." arXiv preprint arXiv:1901.10603 (2019).
>
> [6] Plaut, Elad. "From principal subspaces to principal components with linear autoencoders." arXiv preprint arXiv:1804.10253 (2018)
>
> [7] Amos, Brandon, and J. Zico Kolter. "Optnet: Differentiable optimization as a layer in neural networks." Proceedings of the 34th International Conference on Machine Learning-Volume 70. JMLR. org, 2017.

---

> ### Author Response · Authors · 2019-11-13
> **Response to Reviewer #3- Part 1/2**
>
> This review has been extremely useful---responding to it has broadened our understanding of our submission and enabled us to identify several connections that were heretofore less clear to us.
>
> We would love to hear your feedback on the following discussions inspired by your review, and we will be more than happy to incorporate them into the revised paper if you are in favor of so.
>
> Before addressing the reviewer's objections, we consider two key points worthy of emphasizing, that we will use throughout this response:
>
> (i) Corollary 1 and Remark (5) which follows it: Based on the corollary any critical point of our loss $L$ is a critical point of the original MSE loss but not vice versa. In light of Theorem 2 this means that $L$ eliminates those undesirable global minima of the original loss (i.e., exactly those which suffer from the invariance).
>
> Above describes advantage owing to the difference from the original loss, but there is also further profit gained from their similarities:
>
> (ii) Consider the side by side comparison of our loss and MSE loss, along with their respective gradients, provided on pages 4 and 5 of the paper. Any gradient written for the original loss can be turned simply into the gradient for $L$ by just two component-wise matrix products with constant matrices. Moreover, by Lemma 1 the complexity of evaluating $L$ itself is of the same order as MSE loss too. Given the many repeated terms in the formulas, a careful implementation will eliminate much of the added complexity.
>
> Built on the above two points, the clarification for the two claims mentioned in the review are then as follows.
>
> (1) Theoretical contribution:
> We believe point (i) alone is a substantial and important theoretical contribution: Analyzing the loss surface for various architectures of linear/non-linear neural networks is a highly active and prominent area of research. Many of these works (e.g. [2, 3, 4, 5]) start by citing the seminal results of [1] for shallow LAEs before extending it to more complex networks. However, most work retains the original MSE loss, and they prove the same critical point characterization of [1] for their specific architecture of interest. Most notably [2] extends the results of [1] to deep linear networks and shallow RELU networks. First, the submission is unique in going after a loss with better loss surface properties. In addition, secondly, given that the set of critical points of $L$  is a subset of critical points of MSE loss, many of the mentioned results likely extend. In light of the removal of undesirable global minima through $L$, examining more complex networks is certainly a very promising direction.

---

### Decision · Program_Chairs · 2019-12-19

**Decision:**

Reject

**Comment:**

Quoting from R3: "This paper proposes and analyzes a new loss function for linear autoencoders (LAEs) whose minima directly recover the principal components of the data. The core idea is to simultaneously solve a set of MSE LAE problems with tied weights and increasingly stringent masks on the encoder/decoder matrices."  With two weak acceptance recommendations and a recommendation for rejection, this paper is borderline in terms of its scores.

The approach and idea are interesting.  The main shortcoming of the paper, as highlighted by the reviewers, is that the approach and theoretical analysis are not properly motivated to solve an actual problem faced in real-world data.  The approach does not provide a better algorithm for recovering the eigenvectors of the data, nor is it proposed as part of a learning framework to solve a real-world problem.  Experiments are shown on synthetic data and MNIST.  As a stand-alone theoretical result, it leaves open questions as to the proposed utility.